# Implicit Bias of Mirror Flow for Shallow Neural Networks in Univariate Regression

**Shuang Liang**
UCLA
liangshuang@g.ucla.edu

**Guido Montúfar**
UCLA & MPI MIS
montufar@math.ucla.edu

## Abstract

We examine the implicit bias of mirror flow in least squares error regression with wide and shallow neural networks. For a broad class of potential functions, we show that mirror flow exhibits lazy training and has the same implicit bias as ordinary gradient flow when the network width tends to infinity. For univariate ReLU networks, we characterize this bias through a variational problem in function space. Our analysis includes prior results for ordinary gradient flow as a special case and lifts limitations which required either an intractable adjustment of the training data or networks with skip connections. We further introduce *scaled potentials* and show that for these, mirror flow still exhibits lazy training but is not in the kernel regime. For univariate networks with absolute value activations, we show that mirror flow with scaled potentials induces a rich class of biases, which generally cannot be captured by an RKHS norm. A takeaway is that whereas the parameter initialization determines how strongly the curvature of the learned function is penalized at different locations of the input space, the scaled potential determines how the different magnitudes of the curvature are penalized.

## 1 Introduction

The implicit bias of a parameter optimization procedure refers to the phenomenon where, among the many candidate parameter values that might minimize the training loss, the optimization procedure is biased towards selecting one with certain particular properties. This plays an important role in explaining how overparameterized neural networks, even when trained without explicit regularization, can still learn suitable functions that perform well on new data (Neyshabur et al., 2015; Zhang et al., 2017). As such, the concrete characterization of the implicit biases of parameter optimization in neural networks and how they affect the solution functions is one of the key concerns in deep learning theory and has been subject of intense study in recent years (see, e.g., Ji & Telgarsky, 2019; Lyu & Li, 2020; Williams et al., 2019; Chizat & Bach, 2020; Sahs et al., 2022; Frei et al., 2023; Jin & Montúfar, 2023). In this work we consider an important class of parameter optimization procedures that have remained relatively underexplored, namely mirror descent as applied to solving regression problems with overparametrized neural networks.

Mirror descent is a broad class of first-order optimization algorithms that generalize ordinary gradient descent (Nemirovskij & Yudin, 1983). It can be viewed as using a general distance-like function, defined by the choice of a convex potential function, instead of the usual Euclidean squared distance to determine the update direction in the search space (Beck & Teboulle, 2003). The choice of the geometry of the parameter space of a learning system is crucial, as it can affect the speed of convergence and the implicit bias of parameter optimization (Neyshabur et al., 2017). The implicit bias of mirror descent has been studied theoretically for linear models (Gunasekar et al., 2018a; Sun et al., 2023; Pesme et al., 2024), matrix sensing (Wu & Rebeschini, 2021), and also for nonlinear models at least in a local sense (Azizan et al., 2022). However, we are not aware of any works characterizing the implicit bias of mirror decent for neural networks describing more than the possible properties of the parameters also the nature of the returned solution functions. This is the gap that we target in this article. We focus on the particular setting of mirror flow applied to least squares error regression with shallow and infinitely wide neural networks.

We show for networks with standard parametrization that, under a broad class of potential functions, mirror flow displays lazy training, meaning that even though the loss converges to zero the parameters

remain nearly unchanged. We show in turn that mirror flow has the same implicit bias as gradient flow when the network width tends to infinity. This is surprising because, in contrast, for finite-dimensional parameter models mirror flow has been observed to exhibit a different implicit bias than gradient descent (see, e.g., Gunasekar et al., 2018a; Sun et al., 2023). We then proceed to characterize the particular solution function that is returned by mirror flow for wide ReLU networks, by expressing the difference between the network functions before and after training, $g(x) = f_{\text{trained}}(x) - f_{\text{initial}}(x)$, as the solution to a variational problem. By examining the function representation cost, we show that mirror flow is biased towards minimizing a weighted $L^2$ norm of the second-order derivative of the difference function, $\int p^{-1}(x)(g''(x))^2 \mathrm{d}x$, where the weight $p^{-1}$ is determined by the network parameter initialization, and two additional terms that control the first-order derivative. This includes as a special case a previous result for gradient flow and ReLU activations by Jin & Montúfar (2023). Our analysis also removes a key limitation of that result, which required either an intractable linear adjustment of the training data or otherwise adding skip connections to the network. We observe that other previous works on the implicit bias of gradient descent suggested that shallow ReLU networks in the kernel regime could be well-approximated by a cubic spline interpolator of the training data (Williams et al., 2019; Heiss et al., 2019; Sahs et al., 2022) but overlooked the need of linearly adjusting the training data or adding skip connections. We illustrate that even a slight modification of the training data by applying a constant shift to all training labels can lead to significantly different solutions upon training which are biased not only in terms of curvature but also slopes.

To explore training algorithms capable of implementing different forms of regularization depending on the parameter geometry, we introduce mirror descent with *scaled potentials*. These are separable potentials $\Phi(\theta) = \sum_k \phi(\theta_k)$ whose input is scaled by the network width $n$ as $\Phi(n\theta)$. We show that for networks with standard parametrization, mirror flow with scaled potentials also exhibits lazy training but is not in the kernel regime. In this case the training dynamics can become fundamentally different from that of gradient flow, for which the lazy and kernel regimes fully overlap (see, e.g., Lee et al., 2019). For networks with absolute value activations, we show that mirror flow with scaled potentials induces a rich class of implicit regularizations. The difference function minimizes a functional of the form $\int D_\phi(g''(x)p^{-1}(x), 0)p(x)dx$, where $D_\phi$ denotes the Bregman divergence induced by $\phi$. Notably, for non-homogeneous potential functions $\phi$, the resulting regularization is not homogeneous and therefore it cannot be expressed by an RKHS norm. Overall our results show that, whereas the choice of parameter initialization distribution determines how strongly the curvature is penalized at different locations in the input domain via $p$, the choice of the potential function $\phi$ determines how the different magnitudes of the curvature are penalized.

## 1.1 CONTRIBUTIONS

The goal of this article is to provide insights into the implicit bias of mirror flow for shallow and wide neural networks with standard parametrization. Our contributions can be summarized as follows.

- We show for a broad class of potential functions that, mirror flow, when applied to networks with general input dimension, displays lazy training. We show that in this case the implicit bias of mirror flow is equivalent to that of gradient flow when the number of hidden units tends to infinity.

- We characterize the implicit bias of mirror flow for univariate networks in function space. Concretely, we express the difference of the trained and initialized network function as the solution to a variational minimization problem in function space (Theorem 2).

- We introduce a class of scaled potentials for which we show that mirror flow, applied to networks with general input dimension, falls in the lazy regime but not in the kernel regime. We characterize the corresponding implicit bias for infinitely wide networks and observe it is strongly dependent on the potential and thus different from ordinary gradient flow.

- Finally, we characterize the implicit bias of mirror flow with scaled potentials in function space for univariate networks with absolute value activation function (Theorem 9). Our results show that the choice of the potential biases the magnitude of the curvature of the solution function.

## 1.2 RELATED WORKS

**Implicit bias of mirror descent** The implicit bias of mirror descent has been studied for underdetermined linear models. In classification problems with linearly separable data, Sun et al. (2022) show that mirror descent with homogeneous potentials converges in direction to a max-margin

classifier with respect to a norm defined by the potential. Pesme et al. (2024) extend this work to non-homogeneous potentials. For regression, Gunasekar et al. (2018a); Azizan & Hassibi (2019) show that mirror descent is biased towards the minimizer of the loss that is closest to the initialization with respect to the Bregman divergence defined by the potential. Azizan et al. (2022) extend this result to nonlinear models, albeit placing a strong assumption that the initial parameter is sufficiently close to a solution. For matrix sensing, Wu & Rebeschini (2021) show mirror descent with spectral hypentropy potential tends to recover low-rank matrices. Li et al. (2022) connect gradient flow with commuting reparameterization to mirror flow, and show that both have identical trajectories and implicit biases. This connection is further studied by Chou et al. (2023) specifically for linear models.

**Implicit bias of gradient descent for overparametrized networks**  A seminal line of works in theoretical deep learning investigates the training dynamics of overparameterized neural networks in the so-called lazy regime (Du et al., 2019; Lee et al., 2019; Lai et al., 2023). Gradient descent training in this regime minimizes the training loss but, remarkably, the network parameters remain nearly unchanged. In turn, the network becomes equivalent to a kernel model (Jacot et al., 2018). A particular consequence is that in the lazy regime the implicit bias of gradient descent can be described as the minimization of an RKHS norm (Zhang et al., 2020). Williams et al. (2019); Sahs et al. (2022) derived the explicit form of this RKHS norm for univariate shallow ReLU networks. However, Williams et al. (2019) implicitly constrain the computation to a subspace in the RKHS consisting of functions with specific boundary conditions, and Sahs et al. (2022) placed an incorrect assumption on the behavior of the initial breakpoints of the network in the infinite width limit. Jin & Montúfar (2023) obtained a function description of the implicit bias of gradient descent for univariate and multivariate shallow ReLU networks where they explicitly characterize the effect of parameter initialization. However, their result requires either networks with skip connections or an intractable linear data adjustment. The latter, noted by the authors, is also equivalent to computing the RKHS norm in a function subspace. Beside the kernel regime, we observe that a complementary line of works studies the dynamics and implicit bias in the active regime, also known as the rich or adaptive regime, where the hidden features change during training and the networks do not behave like kernel models (Mei et al., 2019; Williams et al., 2019; Chizat & Bach, 2020; Jacot et al., 2021; Mulayoff et al., 2021; Shevchenko et al., 2022; Boursier et al., 2022; Chistikov et al., 2023; Qiao et al., 2024).

**Representation cost**  For a parameterized model and a given cost function, e.g., a norm, on the parameter space, the representation cost of a target function refers to the minimal cost required to implement that function. Existing work has revealed a close connection between representation cost and the implicit bias of gradient-based optimization methods, as the latter can often be described by specific parameter norm minimization problems. A number of works have studied the representation cost for neural networks with varying architectures (Gunasekar et al., 2018b; Savarese et al., 2019; Ongie et al., 2020; Boursier & Flammarion, 2023; Jacot, 2023; Parkinson et al., 2024). Among these, Savarese et al. (2019); Ongie et al. (2020); Boursier & Flammarion (2023) have focused on infinitely wide and shallow ReLU networks, similar to our work. While these works focused on the $L^1$ norm on the output weights as the cost function, we considered the $L^2$ norm and encompasses a broad family of cost functions induced by Bregman divergence. Moreover, instead of studying representation cost in a static and optimization-free manner, we provide detailed analyses of the optimization dynamics of mirror flow and characterize the implicit bias with concrete convergence guarantees.

## 2 PROBLEM SETUP

We focus on fully-connected two-layer neural networks with $d$ input units, $n$ hidden units and one output unit, defined by:

$$f(x, \theta) = \sum_{k=1}^{n} a_k \sigma(w_k^T x - b_k) + d, \tag{1}$$

where $\sigma(z)$ is a nonlinear activation function. We write $\theta = \text{vec}(\boldsymbol{W}, \boldsymbol{b}, \boldsymbol{a}, d) \in \mathbb{R}^p$, with $p = n(d + 2) + 1$, for the vector of all parameters, comprising the input weights $\boldsymbol{W} = (w_1, \ldots, w_n)^T \in \mathbb{R}^{n \times d}$, input biases $\boldsymbol{b} = (b_1, \ldots, b_n)^T \in \mathbb{R}^n$, output weights $\boldsymbol{a} = (a_1, \ldots, a_n)^T \in \mathbb{R}^n$, and output bias $d \in \mathbb{R}$. We assume the parameters of the network are initialized by independent samples of mutually independent random variables $\mathcal{W}, \mathcal{B}, \mathcal{A}$ and $\mathcal{D}$ as follows:

$$w_k \sim \mathcal{W} = \text{Unif}(\mathbb{S}^{d-1}), \ b_k \sim \mathcal{B}, \ a_k \sim \frac{1}{\sqrt{n}} \mathcal{A}, \ k = 1, \ldots, n; \quad d \sim \mathcal{D}. \tag{2}$$

We assume $\mathcal{W}$ is uniform on the unit sphere in $\mathbb{R}^d$, and $\mathcal{B}$, $\mathcal{A}$, $\mathcal{D}$ are sub-Gaussian random variables. Further, assume that $\mathcal{B}$ has a continuous density function $p_{\mathcal{B}}$ and that the random vector $(\mathcal{W}, \mathcal{B})$ is symmetric, i.e., $(\mathcal{W}, \mathcal{B})$ and $(-\mathcal{W}, -\mathcal{B})$ have the same distribution. We use $\mu$ for the distribution of $(\mathcal{W}, \mathcal{B})$, and $\hat{\theta} = \text{vec}(\hat{\boldsymbol{W}}, \hat{\boldsymbol{b}}, \hat{\boldsymbol{a}}, \hat{d})$ for the initial parameter vector.

In Appendix C, we further discuss our settings on the network parametrization and initialization. Specifically, we compare the parametrization in (1), which is known as the standard parametrization, to the *NTK parametrization* (Jacot et al., 2018), and discuss the effect of the *Anti-Symmetrical Initialization* (Zhang et al., 2020), which is a common technique to ensure zero function output at initialization, on the implicit bias of mirror descent training.

Consider a finite training data set $\{(x_i, y_i)\}_{i=1}^m \subset \mathbb{R}^d \times \mathbb{R}$. Consider the mean squared error $L(\theta) = \frac{1}{2m} \sum_{i=1}^m (f(x_i, \theta) - y_i)^2$. Given a strictly convex and twice differentiable *potential function* $\Phi$ on the parameter space, the *Bregman divergence* (Bregman, 1967) with respect to $\Phi$ is defined as $D_\Phi(\theta, \theta') = \Phi(\theta) - \Phi(\theta') - \langle \nabla \Phi(\theta'), \theta - \theta' \rangle$. The mirror descent parameter update (Nemirovskij & Yudin, 1983; Beck & Teboulle, 2003) for minimizing the loss $L(\theta)$ is defined by:

$$\theta(t+1) = \arg\min_{\theta'} \eta \langle \theta', \nabla L(\theta(t)) \rangle + D_\Phi(\theta', \theta(t)), \tag{3}$$

where the constant $\eta > 0$ is the learning rate. When the learning rate takes infinitesimal values (we consider $\eta = \frac{\eta_0}{n}$ for some constant $\eta_0 > 0$ and $n \to \infty$), (3) can be characterized by the *mirror flow*:

$$\frac{\mathrm{d}}{\mathrm{d}t} \theta(t) = -\eta (\nabla^2 \Phi(\theta(t)))^{-1} \nabla L(\theta(t)). \tag{4}$$

The mirror flow (4) can be interpreted as a Riemannian gradient flow with metric tensor $\nabla^2 \Phi$. For convenience, we provide in Appendix F a brief introduction to the Riemannian geometry of the parameter space endowed with metric $\nabla^2 \Phi$.

The dynamics for the predictions can be given as follows. Let $\boldsymbol{y} = [y_1, \ldots, y_m]^T \in \mathbb{R}^m$ denote the vector of the training labels and $\boldsymbol{f}(t)$ denote the vector of the network prediction on the training input data points at time $t$, i.e., $\boldsymbol{f}(t) = [f(x_1, \theta(t)), \ldots, f(x_m, \theta(t))]^T \in \mathbb{R}^m$. By the chain rule,

$$\frac{\mathrm{d}}{\mathrm{d}t} \boldsymbol{f}(t) = -n\eta H(t)(\boldsymbol{f}(t) - \boldsymbol{y}),$$
$$\text{where } H(t) \triangleq n^{-1} J_\theta \boldsymbol{f}(t) (\nabla^2 \Phi(\theta(t)))^{-1} J_\theta(\boldsymbol{f}(t))^T. \tag{5}$$

Here $J_\theta \boldsymbol{f}(t) = [\nabla_\theta f(x_1, \theta(t)), \ldots, \nabla_\theta f(x_m, \theta(t))]^T \in \mathbb{R}^{m \times p}$ denotes the Jacobian matrix of the network output $\boldsymbol{f}(t)$ with respect to the network parameter $\theta$. We see in (5) that different choices of $\Phi$ induce different modifications of the Gram matrix $J_\theta \boldsymbol{f}(t)(J_\theta \boldsymbol{f}(t))^T$.

To fix a few notations, we write $[n] = \{1, 2, \ldots, n\}$ for $n \in \mathbb{N}$. We use $C^k(\Omega)$ for the space of $k$ times continuously differentiable real-valued functions on $\Omega$ and $C(\Omega)$ for the space of continuous real-valued functions on $\Omega$. We use $\text{supp}(\cdot)$ to denote the support set of a function or a distribution. We say that a sequence of events $\{\mathcal{E}_n\}_{n \in \mathbb{N}}$ holds with high probability if $P(\mathcal{E}_n) \to 1$ as $n \to \infty$.

## 3 MAIN RESULTS

We present our main theoretical results, followed by an overview of the proof techniques.

### 3.1 IMPLICIT BIAS OF MIRROR FLOW WITH UNSCALED POTENTIALS

We consider the following assumptions on the potential function.

**Assumption 1.** *The potential function* $\Phi\colon \mathbb{R}^p \to \mathbb{R}$ *satisfies: (i)* $\Phi$ *can be written in the form* $\Phi(\theta) = \sum_{k=1}^p \phi(\theta_k - \hat{\theta}_k)$, *where* $\phi$ *is a real-valued function on* $\mathbb{R}$*; (ii)* $\Phi$ *is twice continuously differentiable; (iii)* $\nabla^2 \Phi(\theta)$ *is positive definite for all* $\theta \in \mathbb{R}^p$*.*

Potentials that satisfy (i) are referred to as *separable* in the literature, as they operate independently on each coordinate of the input. We consider separable potentials since they have a natural definition as $n$ grows and can be conveniently extended to infinitely dimensional spaces (see details in Section 3.3). Items (ii) and (iii) are common in the mirror descent literature (Sun et al., 2023; Pesme et al., 2024). Assumption 1 is quite mild and covers a broad class of potentials including: (1) $\phi(x) = x^2$

(which recovers ordinary gradient descent); (2) $\phi(x) = \cosh(x)$; and (3) the *hypentropy function* $\phi(x) = \rho_\beta(x) = x \operatorname{arcsinh}(x/\beta) - \sqrt{x^2 + \beta^2}$ for $\beta > 0$ (Ghai et al., 2020). In particular and significantly, we do not require homogeneity. Assumption 1 does not cover the unnormalized negative entropy $\phi(x) = x \log x - x$ and the corresponding natural gradient descent (NGD). However, since the second derivative of the hypentropy function, $1/\sqrt{x^2 + \beta^2}$, converges to that of the negative entropy, $1/x$, as $\beta \to 0^+$, we can still obtain insights into NGD by applying our results to hypentropy potentials. We refer to potentials that satisfy Assumption 1 as *unscaled potentials* in order to distinguish them from the *scaled potentials* which we examine in Section 3.2.

The following theorem characterizes the implicit bias of mirror flow for least squares regression with wide neural networks with ReLU activation $\sigma(s) = \max\{0, s\}$.

**Theorem 2** (**Implicit bias of mirror flow for wide ReLU network**). *Consider a two layer ReLU network* (1) *with $d \geq 1$ input units and $n$ hidden units, where we assume $n$ is sufficiently large. Consider parameter initialization* (2) *and, specifically, let $p_\mathcal{B}(b)$ denote the density function for random variable $\mathcal{B}$. Consider any finite training data set $\{(x_i, y_i)\}_{i=1}^m$ that satisfies $x_i \neq x_j$ when $i \neq j$ and $\{\|x_i\|_2\}_{i=1}^m \subset \operatorname{supp}(p_\mathcal{B})$, and consider mirror flow* (4) *with a potential function $\Phi$ satisfying Assumption 1 and learning rate $\eta = \Theta(1/n)$. Then, there exist constants $C_1, C_2 > 0$ such that with high probability over the random parameter initialization, for any $t \geq 0$,*

$$\|\theta(t) - \hat{\theta}\|_\infty \leq C_1 n^{-1/2}, \quad \lim_{n \to \infty} \|H(t) - H(0)\|_2 = 0, \quad \|\boldsymbol{f}(t) - \boldsymbol{y}\|_2 \leq e^{-\eta_0 C_2 t} \|\boldsymbol{f}(0) - \boldsymbol{y}\|_2. \quad (6)$$

*Moreover, letting $\theta(\infty) = \lim_{t \to \infty} \theta(t)$, we have for any given $x \in \mathbb{R}^d$, that $\lim_{n \to \infty} |f(x, \theta(\infty)) - f(x, \theta_{\mathrm{GF}}(\infty))| = 0$, where $\theta_{\mathrm{GF}}(\infty)$ denotes the limiting point of gradient flow, i.e., mirror flow* (4) *with $\Phi(\theta) = \|\theta\|_2^2$, on the same training data and initial parameter.*

*Assuming univariate input data, i.e., $d = 1$, we have for any given $x \in \mathbb{R}$, that $\lim_{n \to \infty} |f(x, \theta(\infty)) - f(x, \hat{\theta}) - \bar{h}(x)| = 0$, where $\bar{h}(\cdot)$ is the solution to the following variational problem:*

$$\min_{h \in \mathcal{F}_{\mathrm{ReLU}}} \mathcal{G}_1(h) + \mathcal{G}_2(h) + \mathcal{G}_3(h) \quad \text{s.t.} \ h(x_i) = y_i - f(x_i, \hat{\theta}), \ \forall i \in [m],$$

$$\text{where} \begin{cases} \mathcal{G}_1(h) = \displaystyle\int_{\operatorname{supp}(p_\mathcal{B})} \frac{(h''(x))^2}{p_\mathcal{B}(x)} \mathrm{d}x \\ \mathcal{G}_2(h) = (h'(+\infty) + h'(-\infty))^2 \\ \mathcal{G}_3(h) = \dfrac{1}{\mathbb{E}[\mathcal{B}^2]} \Big( \displaystyle\int_{\operatorname{supp}(p_\mathcal{B})} h''(x)|x|\mathrm{d}x - 2h(0) \Big)^2. \end{cases} \quad (7)$$

*Here $\mathcal{F}_{\mathrm{ReLU}} = \{h(x) = \int \alpha(w, b)[wx - b]_+ \mathrm{d}\mu \colon \alpha \in C(\operatorname{supp}(\mu)), \alpha \text{ is uniformly continuous}\}$, $h'(+\infty) = \lim_{x \to +\infty} h'(x)$, and $h'(-\infty) = \lim_{x \to -\infty} h'(x)$. In addition, if $\mathcal{B}$ is supported on $[-B, B]$ for some $B > 0$, then $\mathcal{G}_3(h) = \frac{1}{\mathbb{E}[\mathcal{B}^2]}(B(h'(+\infty) - h'(-\infty)) - (h(B) + h(-B)))^2$.*

To interpret the result, (6) means that mirror flow exhibits lazy training and the network behaves like a kernel model with kernel $H(0)$. The theorem indicates that mirror flow has the same implicit bias as ordinary gradient flow, and the reason behind this is that, loosely speaking, since the parameters stay nearly unchanged during training, the effect of the potential reduces to the effect of a quadratic form, i.e., $\Phi(\theta) \simeq \|\theta\|_2^2$. Further below we discuss scaled potentials breaking outside this regime.

To interpret the implicit bias described in (7), we discuss the effect of the different functionals on the solution function: (1) $\mathcal{G}_1$ serves as a curvature penalty, which can be viewed as the $p_\mathcal{B}^{-1}$-weighted $L^2$ norm of the second derivative of $h$. Note in our setting $p_\mathcal{B}$ corresponds to the density of $\mathcal{B}/\mathcal{W}$, which is the breakpoint of a ReLU in the network. (2) $\mathcal{G}_2$ favors opposite slopes at positive and negative infinity. (3) The effect of $\mathcal{G}_3$ is less clear under general sub-Gaussian initializations. In the case where $\mathcal{B}$ is compactly supported, $\mathcal{G}_3$ can be interpreted as follows: if $h$ is generally above the $x$-axis (specifically, $h(B) + h(-B) > 0$), then $\mathcal{G}_3$ encourages $h'(+\infty) > h'(-\infty)$ and thus favors a U-shaped function; if on the contrary $h$ is below the $x$-axis, $\mathcal{G}_3$ favors an inverted U-shaped function.

The functional $\mathcal{G}_1$ is familiar from the literature on space adaptive interpolating splines and in particular natural cubic splines when $p_\mathcal{B}$ is constant (see, e.g., Ahlberg et al., 2016). However, with $\mathcal{G}_2$ and $\mathcal{G}_3$, the solution to (7) has regularized boundary conditions. In Figure 1, we illustrate numerically that gradient flow returns natural cubic splines only on very specific training data.

**Remark 3** (Relaxed potential assumption). Assumption 1 requires the potential $\Phi$ to be "centered" at the initial parameter $\hat{\theta}$. In Appendix A.4, we show Theorem 2 also holds for potentials of the form $\Phi(\theta) = \sum_k \phi(\theta_k)$, given the initial input biases are sampled from a bounded distribution.

**Remark 4** (Gradient flow with reparametrization). As observed by Li et al. (2022), mirror flow is equivalent to gradient flow with commuting reparameterization. In Appendix D, we use our results to characterize the implicit bias of gradient flow with particular reparametrizations.

**Remark 5** (Absolute value activations). For networks with absolute value activation $\sigma(s) = |s|$, a similar result to Theorem 2 holds, which we formally present in Theorem 24. The only change is that $\mathcal{G}_2$ and $\mathcal{G}_2$ disappear in the objective functional.

**Remark 6** (Skip connections). For networks with skip connections, i.e., $f_{\text{skip}}(x, (\theta, u, v)) = f(x, \theta) + ux + v$ with $u, v \in \mathbb{R}$, the skip connections can modify the value of $\mathcal{G}_2(f_{\text{skip}})$ and $\mathcal{G}_3(f_{\text{skip}})$ without affecting $\mathcal{G}_1(f_{\text{skip}})$. Therefore, if we assume that the skip connections have a much faster training dynamics than the other parameters[1], then the solution function is biased towards solving $\min_{h \in \mathcal{F}_{\text{ReLU}}, u, v \in \mathbb{R}} \mathcal{G}_1(h)$ subject to $h(x_i) + ux_i + v = y_i$ for $i \in [m]$. By this we recover the result obtained by Jin & Montúfar (2023) for univariate networks with skip connections.

**Remark 7** (Natural gradient descent). Theorem 2 holds for hypentropy potential $\rho_\beta$ for any $\beta > 0$, and in particular, in the limit as $\beta \to 0^+$. This is consistent with a result by Zhang et al. (2019) stating NGD converges to the same point as ordinary gradient descent.

## 3.2 Implicit bias of mirror flow with scaled potentials

We introduce scaled potentials and show that these induce a rich class of implicit biases.

**Assumption 8.** *Assume the potential function $\Phi \colon \mathbb{R}^p \to \mathbb{R}$ satisfies: (i) $\Phi$ can be written in the form of $\Phi(\theta) = \frac{1}{n^2} \sum_{k=1}^p \phi\big(n(\theta_k - \hat{\theta}_k)\big)$, where $\phi$ is a real-valued function on $\mathbb{R}$; (ii) $\phi$ takes the form of $\phi(x) = \frac{1}{1+\omega}\big(\psi(x) + \omega x^2\big)$, where $\omega > 0$ and $\psi \in C^3(\mathbb{R})$ is a convex function on $\mathbb{R}$.*

In Assumption 8, the factor $1/n^2$ prevents the largest eigenvalue of the Hessian $\nabla^2 \Phi$ from diverging as $n \to \infty$. The form of $\phi$ can be interpreted as a convex combination of an arbitrary smooth, convex function $\psi$ and the quadratic function $x^2$. A further discussion is presented in Section 3.3.

The following result characterizes the implicit bias of mirror flow with scaled potentials for wide networks with absolute value activations. We consider absolute value activations because, compared to ReLU they yield a smaller function space that makes the problem more tractable.

**Theorem 9** (**Implicit bias of scaled mirror flow for wide absolute value network**). *Consider a two layer network (1) with absolute value activation, $d \geq 1$ input units and $n$ hidden units, where we assume $n$ is sufficiently large. Consider parameter initialization (2) and, specifically, let $p_{\mathcal{B}}(b)$ denote the density function for random variable $\mathcal{B}$. Additionally, assume $\mathcal{B}$ is compactly supported. Given any finite training data set $\{(x_i, y_i)\}_{i=1}^m$ with $x_i \neq x_j$ when $i \neq j$ and $\{\|x_i\|_2\}_{i=1}^m \subset \text{supp}(p_{\mathcal{B}})$, consider mirror flow (4) with a potential satisfying Assumptions 8 and learning rate $\eta = \Theta(1/n)$. Specifically, assume the potential is induced by a univariate function $\phi(z) = \frac{1}{1+\omega}(\psi(z) + \omega z^2)$. There exists $\omega_0 > 0$ (depending on $\psi$, the training data and the initialization distribution) such that for any $\omega > \omega_0$, there exist constants $C_1, C_2 > 0$ such that with high probability over the random parameter initialization, for any $t \geq 0$,*

$$\|\theta(t) - \hat{\theta}\|_\infty \leq C_1 n^{-1}, \quad \|\boldsymbol{f}(t) - \boldsymbol{y}\|_2 \leq e^{-\eta_0 C_2 t}\|\boldsymbol{f}(0) - \boldsymbol{y}\|_2 \tag{8}$$

*Moreover, assuming univariate input data, i.e., $d = 1$, and letting $\theta(\infty) = \lim_{t \to \infty} \theta(t)$, we have for any given $x \in \mathbb{R}$ that $\lim_{n \to \infty} |f(x, \theta(\infty)) - f(x, \hat{\theta}) - \bar{h}(x)| = 0$, where $\bar{h}(\cdot)$ is the solution to the following variational problem:*

$$\min_{h \in \mathcal{F}_{\text{Abs}}} \int_{\text{supp}(p_{\mathcal{B}})} D_\phi\Big(\frac{h''(x)}{2p_{\mathcal{B}}(x)}, 0\Big) p_{\mathcal{B}}(x)\mathrm{d}x \quad \text{s.t. } h(x_i) = y_i - f(x_i, \hat{\theta}), \forall i \in [m]. \tag{9}$$

*Here $D_\phi$ denotes the Bregman divergence on $\mathbb{R}$ induced by the univariate function $\phi$, and $\mathcal{F}_{\text{Abs}} = \{h(x) = \int \alpha(w, b)|wx - b|\mathrm{d}\mu \colon \alpha \in C(\text{supp}(\mu)), \alpha \text{ is even and uniformly continuous}\}$.*

---

[1]This is can achieved by setting the learning rate for skip connections significantly larger than the learning rate of the other network parameters.

Note, given any non-homogeneous function $\phi$, the objective functional in (9) is not homogeneous and therefore it cannot be expressed by any function norm (including RKHS norm). Also note, even though mirror flow displays lazy training, the kernel matrix $H(t)$ does not necessarily stay close to $H(0)$ due to the scaling factor $n$ inside the potential. This implies that training may not be in the kernel regime (see numerical illustration in Figure 5).

The implicit bias described in (9) can be interpreted as follows: the parameter initialization distribution determines $p_{\mathcal{B}}$ and thus the strength of the penalization of the second derivative of the learned function at different locations of the input space; on the other hand, the potential function $\phi$ determines the strength of the penalization of the different magnitudes of the second derivative. In particular, if we choose $\phi = \frac{1}{2}(|x|^p + x^2)$, then a smaller $p$ allows relatively extreme magnitudes for the second derivative, whereas a larger $p$ allows for more moderate values. This is illustrated in Figure 2.

Theorem 9 relies on the assumption that $\omega$ in $\phi = \frac{1}{1+\omega}(\psi + \omega x^2)$ is larger than a threshold $\omega_0$. In Proposition 30, we give a closed-form formula for this threshold. Further, numerical experiments in Section 4 indicate the theorem holds even when $\omega$ takes relatively small values, for instance, $\omega = 1$.

**Remark 10** (Scaled natural gradient descent). In Theorem 38, we show Theorem 9 also holds for scaled hypentropy potentials $\Phi(\theta) = \frac{1}{n^2}\rho_\beta(n(\theta_k - \hat{\theta}_k))$ when $\beta$ is sufficiently large, and it therefore provides insights into scaled NGD. Empirically we observe that as $\beta \to 0^+$, networks trained with scaled hypentropy converge to the solution to problem (9) with $\phi(x) = |x|^3$ (see Figure 7).

### 3.3 GENERAL FRAMEWORK FOR THE PROOF OF THE MAIN RESULTS

We present a general framework to derive our main results, which builds on the strategy of Jin & Montúfar (2023) with a few key differences. First, we examine the training dynamics of mirror flow, for which the analysis is more intricate than that of gradient flow due to the need to account for the evolution of the inverse Hessian of the potential throughout training. In particular, in the scaled potential case, we carefully analyze how the Hessian's properties, such as the decay rate of its smallest eigenvalue, affect the parameter trajectories. Second, we solve the minimal representation cost problem to derive the function description of the implicit bias. This allows us to remove a previous requirement to linearly adjust the data when the network does not include skip connections. The detailed proofs for Theorems 2 and 9 are presented in Appendix A and Appendix B, respectively.

**Linearization** The first step of the proof is to show that, under mirror flow, functions obtained by training the network are close to those obtained by training only the output weights. Let $\tilde{\theta}(t) = \text{vec}(\boldsymbol{W}(0), \boldsymbol{b}(0), \tilde{\boldsymbol{a}}(t), d(0))$ denote the parameter under the update rule where $\boldsymbol{W}, \boldsymbol{b}, d$ are kept at their initial values and

$$\frac{\mathrm{d}}{\mathrm{dt}}\tilde{\boldsymbol{a}}(t) = -\eta(\nabla^2\Phi(\tilde{\boldsymbol{a}}(t)))\nabla_{\tilde{\boldsymbol{a}}}L(\tilde{\theta}(t)), \quad \tilde{\boldsymbol{a}}(0) = \boldsymbol{a}(0). \tag{10}$$

Here, we write $\Phi(\tilde{\boldsymbol{a}})$ for the potential function applied to the output weights. For unscaled potentials, $\Phi(\tilde{\boldsymbol{a}}) = \sum_{k=1}^n \phi(\tilde{a}_k - \hat{a}_k)$; for scaled potentials, $\Phi(\tilde{\boldsymbol{a}}) = n^{-2}\sum_{k=1}^n \phi(n(\tilde{a}_k - \hat{a}_k))$. Let $\tilde{\boldsymbol{f}}(t) = [f(x_1, \tilde{\theta}(t)), \ldots, f(x_m, \tilde{\theta}(t))]^T \in \mathbb{R}^m$. The evolution of $\tilde{\boldsymbol{f}}(t)$ can be given by

$$\frac{\mathrm{d}}{\mathrm{dt}}\tilde{\boldsymbol{f}}(t) = -n\eta\tilde{H}(t)(\tilde{\boldsymbol{f}}(t) - \boldsymbol{y}), \quad \tilde{\boldsymbol{f}}(0) = \boldsymbol{f}(0),$$
$$\text{where } \tilde{H}(t) \triangleq n^{-1}J_{\tilde{\boldsymbol{a}}}\tilde{\boldsymbol{f}}(t)(\nabla^2\Phi(\tilde{\boldsymbol{a}}(t)))^{-1}J_{\tilde{\boldsymbol{a}}}\tilde{\boldsymbol{f}}(t)^T. \tag{11}$$

We aim to show that for any fixed $x \in \mathbb{R}^d$, $\lim_{n\to\infty}\sup_{t\geq 0}|f(x, \theta(t)) - f(x, \tilde{\theta}(t))| = 0$. For unscaled potentials, we adapt an established approach from gradient flow analysis (Du et al., 2019; Lee et al., 2019). Specifically, we show $\|\theta(t) - \hat{\theta}\|_\infty$ and $\|\tilde{\theta}(t) - \hat{\theta}\|_\infty$ are bounded by $O(n^{-1/2})$ (Proposition 14). Then we bound $\|H(t) - \tilde{H}(t)\|_2$ with the continuity of the Hessian map $\nabla^2\Phi(\cdot)$ in Assumption 1, which gives the desired bound on $|f(x, \theta(t)) - f(x, \tilde{\theta}(t))|$ (Proposition 16). For scaled potentials, we use a similar strategy to show $\|\theta(t) - \hat{\theta}\|_\infty$ and $\|\tilde{\theta}(t) - \hat{\theta}\|_\infty$ are bounded by $O(n^{-1})$ (Proposition 27). Then we show $\|n\theta(t) - n\tilde{\theta}(t)\|_\infty$ can be bounded when the smallest and largest eigenvalues of $\nabla^2\Phi(\theta)$ have mild decay and growth rates, respectively, as $\|\theta\|_2$ increases, which allows us to control $|f(x, \theta(t)) - f(x, \tilde{\theta}(t))|$ (Proposition 29).

**Implicit bias in parameter space** The second step is to characterize the implicit bias for wide networks in their parameter space. In view of the previous step, it suffices to characterize the bias for networks where only the output weights are trained. Since the network is linear with respect to the output weights, we can use the following result on the implicit bias of mirror flow for linear models.

**Theorem 11** (Gunasekar et al., 2018a). *If mirror flow* (10) *converges to zero loss, then the final parameter is the solution to following constrained optimization problem:*

$$\min_{\tilde{\boldsymbol{a}} \in \mathbb{R}^n} D_\Phi(\tilde{\boldsymbol{a}}, \hat{\boldsymbol{a}}) \quad \text{s.t. } f(x_i, \tilde{\theta}) = y_i, \ \forall i \in [m]. \tag{12}$$

We reformulate (12) in a way that allows us to consider the infinite width limit, i.e., $n \to \infty$. Let $\mu_n$ denote the empirical distribution on $\mathbb{R}^d \times \mathbb{R}$ induced by the sampled input weights and biases $\{(\hat{w}_k, \hat{b}_k)\}_{k=1}^n$.[2] Consider a function $\alpha \colon \operatorname{supp}(\mu) \to \mathbb{R}$ whose value encodes the difference of the output weight from its initialization for a hidden unit given by the argument, i.e., $\alpha(\hat{w}_k, \hat{b}_k) = n(\tilde{a}_k - \hat{a}_k), \ \forall k \in [n]$. Then the difference of the network output at a given $x \in \mathbb{R}^d$ from its initialization is $g_n(x, \alpha) = f(x, \tilde{\theta}) - f(x, \hat{\theta}) = \int \alpha(w, b)\sigma(w^T x - b)\mathrm{d}\mu_n$. Noticing $\mu_n$ converges weakly to $\mu$, we write the difference between the infinitely wide networks before and after training as:

$$g(x, \alpha) = \int \alpha(w, b)\sigma(w^T x - b)\mathrm{d}\mu, \quad \alpha \in \Theta. \tag{13}$$

Here $\Theta \subset C(\operatorname{supp}(\mu))$ is chosen as the parameter space for the model (13). Our goal is to identify a cost functional on the parameter space $\mathcal{C}_\Phi \colon \Theta \to \mathbb{R}$ such that, as the width of the network goes to infinity, the solution to (12) converges to the solution to the following problem:[3]

$$\min_{\alpha \in \Theta} \mathcal{C}_\Phi(\alpha) \quad \text{s.t. } g(x_i, \alpha) = y_i - f(x_i, \hat{\theta}), \ \forall i \in [m]. \tag{14}$$

For unscaled potentials, noticing the final output weight returned by mirror flow, i.e., the minimizer of (12), has $O(n^{-1/2})$ bounded difference from its initialization, we have the following approximation for the objective function of (12) when $n$ tends to infinity:[4]

$$D_\Phi(\tilde{\boldsymbol{a}}, \hat{\boldsymbol{a}}) \stackrel{n \to \infty}{=} \sum_{k=1}^n \frac{\phi''(0)}{2}(\tilde{a}_k - \hat{a}_k)^2 = \frac{\phi''(0)}{2n} \int (\alpha(w, b))^2 \mathrm{d}\mu_n, \tag{15}$$

where we consider the Taylor approximation of $\phi$ at the origin. Based on this, for unscaled potentials we show that in the infinite width limit, problem (12) can be described by problem (14) with $\mathcal{C}_\Phi(\alpha) = \int (\alpha(w, b))^2 \mathrm{d}\mu$, regardless of the choice of $\Phi$ (Proposition 17).

In the case of scaled potentials, the objective function in (12) can be reformulated as

$$D_\Phi(\tilde{\boldsymbol{a}}, \hat{\boldsymbol{a}}) = \frac{1}{n^2} \sum_{k=1}^n \Big(\phi(n(\tilde{a}_k - \hat{a}_k)) - \phi(0) - n\phi'(0)(\tilde{a}_k - \hat{a}_k)\Big) = \frac{1}{n} \int D_\phi(\alpha(w, b), 0) \, \mathrm{d}\mu_n. \tag{16}$$

Based on this, for scaled potentials we show that, as $n$ tends to infinity, problem (12) can be described by problem (14) with $\mathcal{C}_\Phi(\alpha) = \int D_\phi(\alpha(w, b), 0)\mathrm{d}\mu$ (Proposition 32).

**Function space description of the implicit bias** The last step of our proof is to translate the implicit bias in parameter space, as described by problem (14), to the function space of univariate networks, $\mathcal{F} = \{h(x) = \int \alpha(w, b)\sigma(wx - b)\mathrm{d}\mu \colon \alpha \in \Theta\}$. We do so by "pulling back" the cost functional $\mathcal{C}_\Phi$ from the parameter space to the function space, via the map $\mathcal{R}_\Phi \colon \mathcal{F} \to \Theta$ defined by

$$\mathcal{R}_\Phi(h) = \underset{\alpha \in \Theta}{\arg\min} \, \mathcal{C}_\Phi(\alpha) \quad \text{s.t. } g(x, \alpha) = h(x), \ \forall x \in \mathbb{R}. \tag{17}$$

The optimization problem in (17) is known as the *minimal representation cost problem*, as it seeks the parameter $\alpha$ with the minimal cost $\mathcal{C}_\Phi(\alpha)$ such that $g(\cdot, \alpha)$ represents a given function $h$. In the following result, we show that for general cost functionals, the map $\mathcal{R}_\Phi$ allows one to translate problem (14) to a variational problem on the function space. The proof is presented in Appendix A.3.

---

[2] $\mu_n(A) = \frac{1}{n} \sum_{k=1}^n \mathbb{I}_A((\hat{w}_k, \hat{b}_k))$ for any measurable set $A \subset \mathbb{R}^d \times \mathbb{R}$, where $\mathbb{I}_A$ is the indicator of $A$.

[3] The convergence is in the following sense: let $\bar{\boldsymbol{a}}$ be the solution to (12) and $\bar{\alpha}_n \in \Theta$ be a continuous function satisfying $\bar{\alpha}_n(\hat{w}_k, \hat{b}_k) = n(\bar{a}_k - \hat{a}_k)$ for all $k \in [n]$. Let $\bar{\alpha}$ be the solution to (14). Then for any given $x \in \mathbb{R}^d$, $g_n(x, \bar{\alpha}_n)$ converges in probability to $g(x, \bar{\alpha})$ as $n \to \infty$.

[4] Here we write $x_n \stackrel{n \to \infty}{=} y_n$ to signify that $\lim_{n \to \infty} x_n - y_n = 0$.

**Proposition 12.** *Assume that $\Theta$ is a vector subspace in $C(\mathrm{supp}(\mu))$ and $C_\Phi$ is strictly convex functional on $\Theta$. Consider the map $\mathcal{R}_\Phi$ defined in (17). If $\bar{\alpha}$ is the solution to problem (14), then $g(\cdot, \bar{\alpha})$ solves the following variational problem:*

$$\min_{h \in \mathcal{F}} \mathcal{C}_\Phi \circ \mathcal{R}_\Phi(h) \quad \text{s.t. } h(x_i) = y_i - f(x_i, \hat{\theta}), \ \forall i \in [m]. \tag{18}$$

With Proposition 12, to obtain function description of the implicit bias, it suffices to characterize the exact form of the map $\mathcal{R}_\Phi$, which is equivalent to solving the minimal representation cost problem. For unscaled potentials and ReLU networks, we set $\Theta_{\mathrm{ReLU}}$ as the space of uniformly continuous functions on $\mathrm{supp}(\mu)$ and our approach builds on the following key observations:

(1) every $\alpha \in \Theta_{\mathrm{ReLU}}$ has a unique even-odd decomposition, $\alpha = \alpha^+ + \alpha^-$ where $\alpha^+ \in \Theta_{\mathrm{Even}} = \{\alpha \in \Theta_{\mathrm{ReLU}} \colon \alpha \text{ is even}\}$ and $\alpha^- \in \Theta_{\mathrm{Odd}} = \{\alpha \in \Theta_{\mathrm{ReLU}} \colon \alpha \text{ is odd}\};$[5]
(2) under the cost functional $\mathcal{C}_\Phi(\alpha) = \int (\alpha(w, b))^2 \mathrm{d}\mu$, the Pythagorean-type equality holds: $\mathcal{C}_\Phi(\alpha) = \mathcal{C}_\Phi(\alpha^+) + \mathcal{C}_\Phi(\alpha^-), \forall \alpha \in \Theta_{\mathrm{ReLU}}$.
(3) the even-odd decomposition in the parameter space is equivalent to decomposing any infinitely wide ReLU network as the sum of an infinitely wide network with absolute value activation and an affine function (this has been previously observed by Ongie et al., 2020):

$$\int \alpha(w, b)[wx - b]_+ \mathrm{d}\mu = \int \alpha^+(w, b)\frac{|wx - b|}{2}\mathrm{d}\mu + \int \alpha^-(w, b)\frac{wx - b}{2}\mathrm{d}\mu. \tag{19}$$

Based on these, we solve the minimal representation cost problem (17) for $h \in \mathcal{F}_{\mathrm{ReLU}}$ by decomposing $h$ as in (19) and then solving two simpler problems on $\mathcal{F}_{\mathrm{Abs}}$ and $\mathcal{F}_{\mathrm{Affine}}$ (Proposition 21), where $\mathcal{F}_{\mathrm{Affine}}$ is the space of affine functions on $\mathbb{R}$. We show $\mathcal{C}_\Phi \circ \mathcal{R}_\Phi = \mathcal{G}_1$ on $\mathcal{F}_{\mathrm{Abs}}$ and $\mathcal{C}_\Phi \circ \mathcal{R}_\Phi = \mathcal{G}_2 + \mathcal{G}_3$ on $\mathcal{F}_{\mathrm{Affine}}$ and thus the implicit bias on $\mathcal{F}_{\mathrm{ReLU}}$ can be described by $\mathcal{G}_1 + \mathcal{G}_2 + \mathcal{G}_3$ (Proposition 22).

For the case of scaled potentials, where the cost functional is $\mathcal{C}_\Phi(\alpha) = \int D_\phi(\alpha(w, b), 0)\mathrm{d}\mu$, the decomposition technique discussed above no longer applies since the Pythagorean equality generally does not holds. Therefore, solving the minimal representation cost problem remains challenging (see more details in Appendix E.1). To mitigate this, we restricted our attention to networks with absolute value activation functions, and solve problem (17) on the subspace $\mathcal{F}_{\mathrm{Abs}}$ (Proposition 36).

## 4 EXPERIMENTS

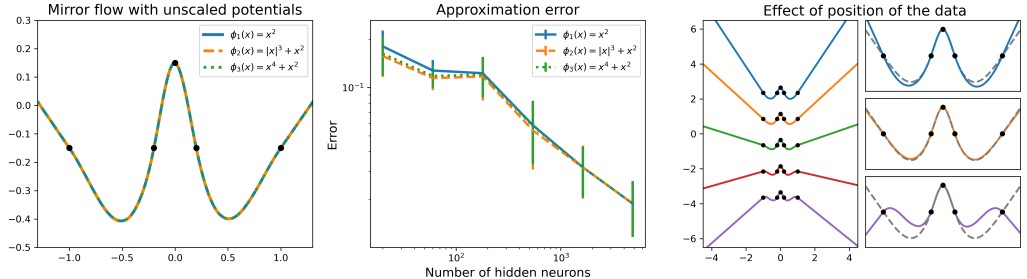

Figure 1: Illustration of Theorem 2. Left: ReLU networks with 4860 hidden neurons, uniformly initialized input biases and zero initialized output weights and biases, trained with mirror flow on a common data set using unscaled potentials: $\phi_1 = x^2$, $\phi_2 = |x|^3 + x^2$, and $\phi_3 = x^4 + x^2$. Middle: $L^\infty$-error between the solution to the variational problem and the networks trained using mirror descent with $\phi_1$, $\phi_2$ and $\phi_3$, against the network width. Right: ReLU networks trained with gradient descent on five different data sets, each obtained by translating the same data set along the $y$-axis.

We numerically illustrate Theorem 2 in Figure 1. In agreement with the theory, the left panel shows that mirror flow with different unscaled potentials have the same implicit bias. The middle panel verifies that the solution to the variational problem captures the networks obtained by mirror descent training, with decreasing error as the number of neurons increases. Notice the errors across different

---

[5]Recall a function $\alpha$ on $\{-1, 1\} \times [-B_b, B_b]$ is called an even function if $\alpha(w, b) = \alpha(-w, -b)$ for all $(w, b)$ in its domain; and an odd function if $\alpha(w, b) = -\alpha(-w, -b)$ for all $(w, b)$ in its domain.

potentials are nearly equal. In fact, we find that the parameter trajectories for different potentials also overlap (see Figure 3). The right panel demonstrates that applying vertical translations to the training data results in much different training outcomes. Only on specific training data, the trained network is equal to the natural cubic spline interpolation of the data (shown as a dashed gray curve). Meanwhile, networks have opposite slopes at positive and negative infinity, which illustrates the effect of functional $\mathcal{G}_2$, and that when the data lies above the $x$-axis, networks have a U-shape and when the data is below the $x$-axis, they have an inverted U-shape, which illustrates the effect of $\mathcal{G}_3$.

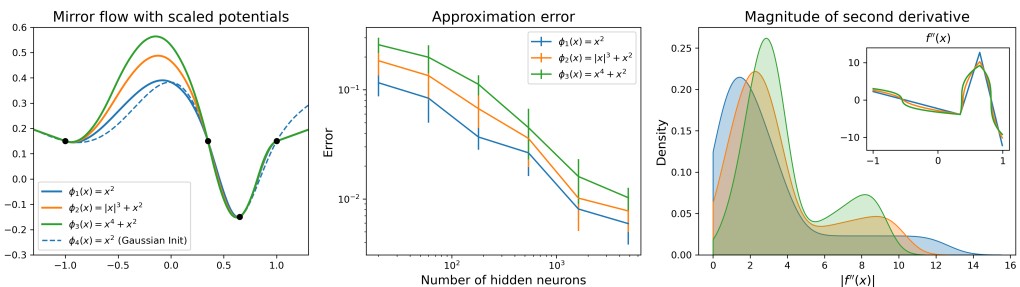

Figure 2: Illustration of Theorem 9. Left: absolute value networks with $4860$ hidden neurons, uniformly initialized input biases and zero initialized output weights and biases, trained with mirror descent on a common data set using scaled potentials: $\phi_1 = x^2$, $\phi_2 = |x|^3 + x^2$, and $\phi_3 = x^4 + x^2$. For comparison we also plot a network with Gaussian initialized input biases, trained with gradient descent. Middle: $L^\infty$-error between the solution to the variational problems and networks trained using mirror descent for scaled potentials $\phi_1$, $\phi_2$ and $\phi_3$, against the network width. Right: the distribution of the magnitude of the second derivative of the solutions to the variational problems for $\phi_1$ (blue), $\phi_2$ (orange), and $\phi_3$ (green). The inset shows the second derivatives over the input domain.

We illustrate Theorem 9 numerically in Figure 2. The left panel illustrates the effect of the scaled potentials and parameter initialization on the implicit bias of mirror descent. When fixing the potential, a network with Gaussian initialization (dashed blue curve) learns a function whose curvature is more concentrated around the origin, compared to a network with uniform initialization (solid blue curve). When fixing the initialization, different potentials result in functions with different curvature magnitudes (solid curves in different colors). The middle plot verifies that the solution to the variational problem captures the trained network solutions as the network width increases. The right panel further illustrates how the potentials in the left panel lead to functions with different curvature magnitudes. We see $\phi_1$ allows occurrences of more extreme small and large magnitudes, whereas $\phi_3$ allows the second derivative to concentrate more on moderate values.

## 5  CONCLUSION

We obtained a function space description of the implicit bias of mirror descent in wide neural networks. In the infinite width limit, the implicit bias becomes independent of the potential and thus equivalent to that of gradient descent. On the other hand, for scaled potentials we found that the implicit bias strongly depends on the potential and in general cannot be captured as a minimum norm in a RKHS. A takeaway message is that the parameter initialization distribution determines the strength of the curvature penalty over different locations of the input space and the potential determines the strength of the penalty over different magnitudes of the curvature.

**Limitations and future work**  Our variational characterization of the implicit bias only applies to univariate, shallow networks with unit-norm initialized input weights. Characterizing the implicit bias for deep nonlinear networks remains an open challenge even for ordinary gradient descent. Extending the results to multivariate networks with general input weights initialization requires stronger theoretical tools (see Appendix E.2 for details). We considered separable potentials which allow for a convenient definition in the infinite width limit and we did not cover potentials defined on a proper subset of the parameter space, such as the negative entropy. Extending our results to more general potentials could be an interesting direction for future work. Other future research directions include deriving closed-form solutions for the variational problems and investigating the rate of convergence as the number of neurons increases.

**Reproducibility statement**  Code to reproduce our experiments is made available at `https://github.com/shuangliang15/implicit-bias-mirror-descent`.

ACKNOWLEDGMENT

This project has been supported by NSF CCF-2212520. GM has also been supported by NSF DMS-2145630, DFG SPP 2298 project 464109215, and BMBF in DAAD project 57616814.

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

APPENDIX

The appendix is organized into the following sections.

- Appendix A: Proof of Theorem 2
- Appendix B: Proof of Theorem 9
- Appendix C: Discussion on parametrization and initialization
- Appendix D: Gradient flow with reparametrization
- Appendix E: Minimal representation cost problem in general settings
- Appendix F: Riemannian geometry of the parameter space $(\Theta, \nabla^2 \Phi)$
- Appendix G: Experiment details
- Appendix H: Additional numerical experiments

## A  PROOF OF THEOREM 2

In this section, we present the proof of Theorem 2. We follow the strategy that we introduced in Section 3.3 and break down the main theorem into several intermediate results. Specifically, in Proposition 14 we show mirror flow training converges to zero training error and is in the lazy regime. In Proposition 17, we show mirror flow with unscaled potentials has the same implicit bias as ordinary gradient flow as the network width tends to infinity. In Proposition 22, we obtain the function space description of the implicit bias.

We use $\text{plim}_{n\to\infty} X_n = X$ or $X_n \xrightarrow{P} X$ to denote that random variables $X_n$ converge to $X$ in probability as $n \to \infty$. We use the notations $O_p$ and $o_p$ to denote the standard mathematical orders in probability, i.e., $X_n = O_p(a_n)$ if for every $\varepsilon > 0$, there exist $M_\varepsilon, N_\varepsilon > 0$ such that $P(|X_n| \le M_\varepsilon a_n) > 1 - \varepsilon$ for every $n > N_\varepsilon$; and $X_n = o_p(a_n)$ if for every $\varepsilon > 0$, there exists $n_\varepsilon$ such that $P(|X_n| < \varepsilon a_n) > 1 - \varepsilon$ for every $n > n_\varepsilon$.

### A.1  LINEARIZATION

We show that under mirror flow with unscaled potentials, the network obtained by training all parameters can be well-approximated by that obtained by training only the output weights.

We first show that the initial kernel matrices $H(0)$, $\tilde{H}(0)$ converge to a positive definite matrix as the width of the network increases.

**Lemma 13.** *Consider network (1) with ReLU or absolute value activation function and initialization (2). Assume that the potential function satisfies Assumption 1, and that the training data $\{x_i\}_{i\in[m]}$ satisfies $x_i \ne x_j$ when $i \ne j$, and $\{\|x_i\|_2\}_{i=1}^m \subset \text{supp}(p_\mathcal{B})$. Then there exists a constant $\lambda_0 > 0$, which is determined by the activation function, the training data, and the initialization distribution, such that $\text{plim}_{n\to\infty} \lambda_{\min}(H(0)) = \text{plim}_{n\to\infty} \lambda_{\min}(\tilde{H}(0)) = (\phi''(0))^{-1}\lambda_0$.*

**Proof.** [Proof of Lemma 13] Note for $i, j \in [m]$, we have

$$H_{ij}(0) = \frac{1}{n}\sum_{k=1}^{3n+1} \nabla_{\theta_k} f(x_i; \hat{\theta}) \frac{1}{\phi''(0)} \nabla_{\theta_k} f(x_j; \hat{\theta})$$

$$= \frac{1}{n\phi''(0)}\sum_{k=1}^{n} \sigma(\hat{w}_k^T x_i - \hat{b}_k)\sigma(\hat{w}_k^T x_j - \hat{b}_k) + \frac{1}{n\phi''(0)}$$

$$+ \frac{1}{n\phi''(0)}\sum_{k=1}^{n} \hat{a}_k^2 \sigma'(\hat{w}_k^T x_i - \hat{b}_k)\sigma'(\hat{w}_k^T x_j - \hat{b}_k)(x_i^T x_j + 1).$$

Noticing $\{a_k\}_{k\in[n]}$ are independent samples from the random variable $\frac{1}{\sqrt{n}}\mathcal{A}$ and by sub-Gaussian concentration bound, we have that $\max_{k\in[n]} |\hat{a}_k| \le C\sqrt{(\log n)/n}$ holds with high probability for some constant $C > 0$. It then follows that:

$$\text{plim}_{n\to\infty} H_{ij}(0) = \frac{1}{\phi''(0)}\mathbb{E}\left[\sigma(\mathcal{W}^T x_i - \mathcal{B})\sigma(\mathcal{W}^T x_j - \mathcal{B})\right].$$

Define $H^\infty = (\phi''(0))^{-1}G$ where $G_{ij} = \mathbb{E}[\sigma(\mathcal{W}^T x_i - \mathcal{B})\sigma(\mathcal{W}^T x_j - \mathcal{B})]$. Then we have $\text{plim}_{n\to\infty} H(0) = H^\infty$. In the meantime, we have

$$\tilde{H}_{ij}(0) = \frac{1}{n}\sum_{k=1}^n \nabla_{a_k} f(x_i; \hat{\theta})\frac{1}{\phi''(\hat{a}_k)}\nabla_{a_k} f(x_j; \hat{\theta}) \xrightarrow{P} H_{ij}^\infty,$$

which means $\tilde{H}(0)$ also converges to $H^\infty$ in probability. Now it suffices to show $G$ is positive definite and let $\lambda_0 = \lambda_{\min}(G)$. Consider the Hilbert space of squared integrable functions:

$$\mathcal{H} = \{h\colon \text{supp}(\mu) \to \mathbb{R}\colon \mathbb{E}[(h(\mathcal{W},\mathcal{B}))^2] < \infty\}$$

with inner-product $\langle h, h'\rangle_{\mathcal{H}} = \mathbb{E}[h(\mathcal{W},\mathcal{B})h'(\mathcal{W},\mathcal{B})]$. For each $x_i \in \mathbb{R}$, define $h_i(w,b) = \sigma(w^T x_i - b)$. Since $(\mathcal{W},\mathcal{B})$ is sub-Gaussian and $h_i$ is uniformly continuous, $h_i \in \mathcal{H}$. Note $G$ is the Gram matrix of $\{h_i\}_{i=1}^m$. Hence, $G \succ 0$ is equivalent to $\{h_i\}_{i=1}^m$ are linearly independent in $\mathcal{H}$.

Assume $\sum_{i=1}^m c_i h_i = 0$ in $\mathcal{H}$, which means $\sum_{i=1}^m c_i h_i(w,b) = 0$ almost everywhere. Without loss of generality, assume $\|x_1\|_2 \geq \|x_2\|_2 \geq \ldots \geq \|x_m\| \geq 0$. We aim to show $c_1 = 0$. Notice that, for any $i \in [m]$, $h_i$ is not differentiable at $(w,b)$ if and only if $w^T x_i - b = 0$. In particular, $h_1$ is not differentiable at $(x_1/\|x_1\|_2, \|x_1\|_2)$. Now for all $j > 1$, by noticing that $\|x_j\| \leq \|x_1\|$ and $x_1 \neq x_j$ we have,

$$(\frac{x_1}{\|x_1\|_2})^T x_j - \|x_1\|_2 = \|x_j\|\cos(\angle(x_1,x_j)) - \|x_1\|_2 < 0$$

where $\angle(x_1,x_j)$ denotes the angle between $x_1$ and $x_j$. This implies that $h_j$ is differentiable at $(x_1/\|x_1\|_2, \|x_1\|_2)$ and so is $\sum_{j\neq 1} c_j h_j$. Therefore, to make $\sum_i c_i h_i$ differentiable everywhere, we have $c_1 = 0$. With the same strategy, we can show $c_j = 0$ for $j = 2, 3, \ldots, m-1$, since $\|x_j\| > 0$. It then follows that all coefficients are zero, which completes the proof. $\qquad\square$

In the next proposition, we show that the model output $\boldsymbol{f}(t)$ converges exponentially fast to $\boldsymbol{y}$ and the parameters have $O_p(n^{-1/2})$ bounded $\ell_\infty$ distance from the initialzation.

**Proposition 14.** *Consider a single-hidden-layer univariate network* (1) *with ReLU or absolute value activation function and initialization* (2). *Assume the number of hidden neurons is sufficiently large. Given any finite training data set $\{(x_i, y_i)\}_{i=1}^m$ that satisfies $x_i \neq x_j$ when $i \neq j$, and $\{\|x_i\|_2\}_{i=1}^m \subset \text{supp}(p_{\mathcal{B}})$, consider mirror flow* (4) *with the potential satisfying Assumptions 1 and learning rate $\eta = \eta_0/n$ with $\eta_0 > 0$. Then there exist constants $C_1, C_2 > 0$ such that with high probability over initialization the following holds:*

$$\sup_{t\geq 0}\|\theta(t) - \hat{\theta}\|_\infty \leq C_1 n^{-1/2}, \quad \lim_{n\to\infty}\sup_{t\geq 0}\|H(t) - H(0)\|_2 = 0, \quad \inf_{t\geq 0}\lambda_{\min}(H(t)) > C_2.$$

*Moreover,*

$$\|\boldsymbol{f}(t) - \boldsymbol{y}\|_2 \leq \exp(-\eta_0 C_2 t)\|\boldsymbol{f}(0) - \boldsymbol{y}\|_2, \ \forall t \geq 0.$$

*Specifically, the constants $C_1$ and $C_2$ can be chosen as*

$$C_1 = \frac{8\phi''(0)\beta\sqrt{m}(B_x+1)K\|\boldsymbol{f}(0) - \boldsymbol{y}\|_2}{\lambda_0}, \quad C_2 = \frac{\lambda_0}{4\phi''(0)}, \qquad (20)$$

*where $\lambda_0 = \text{plim}_{n\to\infty}\lambda_{\min}(\frac{1}{n}J_\theta \boldsymbol{f}(0)J_\theta \boldsymbol{f}(0)^T)$, $\beta = \max_{z\in[-1,1]}(\phi''(z))^{-1}$, $m$ is the size of the training data set, $B_x = \max_{i\in[m]}\|x_i\|_2$, and $K$ is a constant determined by $\mathcal{B}$ and $\mathcal{A}$.*

**Proof.** [Proof of Proposition 14] To ease the notation, we let $J(t) = J_\theta \boldsymbol{f}(t)$ and $Q(t) = (\nabla^2\Phi(\theta(t)))^{-1}$. For $i \in [m]$ and $k \in [n]$, let $\sigma_{ik}(t) = \sigma(w_k(t)^T x_i - b_k(t))$ and $\sigma'_{ik}(t) = \sigma'(w_k(t)^T x_i - b_k(t))$.

By Lemma 13, for sufficiently large $n$ and with high probability, we have $\lambda_{\min}(H(0)) > \lambda_0/(2\phi''(0))$. According to Assumption 1, we have $0 < \beta < \infty$.

Consider

$$\tau(t) = \max\left\{\sqrt{n}\|\boldsymbol{a}(t) - \boldsymbol{a}(0)\|_\infty, n\max_{k\in[n]}\|w_k(t) - w_k(0)\|_\infty, n\|\boldsymbol{b}(t) - \boldsymbol{b}(0)\|_\infty, n|d(t) - d(0)|\right\},$$

and

$$S = \{t \in [0, \infty)\colon \tau(t) > C_1\}.$$

Let $t_1 = \inf S$. Clearly, $0 < t_1$. We now aim to prove that $t_1 = +\infty$. Assume, for the sake of contradiction, that $t_1 < +\infty$. Since $\tau(t)$ is continuous, we have $\tau(t_1) = C_1$.

In the following let $t$ denote any point in $[0, t_1]$. Noticing that $\tau(t) \leq C_1$ implies $\|\theta(t) - \theta(0)\|_\infty \leq C_1 n^{-1/2}$, we have for large enough $n$,

$$\|Q(t)\|_2 = \|\operatorname{Diag}(\frac{1}{\phi''(\theta_k(t) - \theta_k(0))})\|_2 = \max_{k \in [3n+1]} \frac{1}{\phi''(\theta_k(t) - \theta_k(0))} \leq \max_{z \in [-1,1]} \frac{1}{\phi''(z)} = \beta. \tag{21}$$

Next we bound the difference between $H(t)$ and $H(0)$. By concentration inequality for sub-Gaussian random variables (see e.g., Wainwright, 2019), there exists $K > 1$ such that with high probability,

$$\|\boldsymbol{b}(0)\|_\infty \leq \|\boldsymbol{b}(0)\|_2 \leq K\sqrt{n}, \quad \|\boldsymbol{a}(0)\|_\infty \leq \|\boldsymbol{a}(0)\|_2 \leq K.$$

Then we have for any $i, j \in [m], k \in [n]$,

$$|\sigma_{ik}(t)| \leq |w_k(t)^T x_i| + |b_k(t)| \leq (B_x + 1)2K\sqrt{n},$$

and

$$|\sigma_{ik}(t) - \sigma_{jk}(t)| \leq \|w_k(t) - w_k(0)\|_2 \|x_i\|_2 + |b_k(t) - b_k(0)| \leq \sqrt{d}(B_x + 1)C_1 n^{-1}.$$

For $i, j \in [m]$ we have

$$|H_{ij}(t) - H_{ij}(0)| \leq I_1 + I_2 + I_3 + I_4, \tag{22}$$

where

$$\begin{cases} I_1 = \dfrac{1}{n} \sum_{k=1}^{n} \left| \dfrac{\sigma_{ik}(t)\sigma_{jk}(t)}{\phi''(a_k(t) - a_k(0))} - \dfrac{\sigma_{ik}(0)\sigma_{jk}(0)}{\phi''(0)} \right|; \\[2ex] I_2 = \dfrac{1}{n} \sum_{k=1}^{n} \sum_{r=1}^{d} |x_{i,r} x_{j,r}| \left| \dfrac{a_k(t)^2 \sigma'_{ik}(t) \sigma'_{jk}(t)}{\phi''(w_{k,r}(t) - w_{k,r}(0))} - \dfrac{a_k(0)^2 \sigma'_{ik}(0) \sigma'_{jk}(0)}{\phi''(0)} \right|; \\[2ex] I_3 = \dfrac{1}{n} \sum_{k=1}^{n} \left| \dfrac{a_k(t)^2 \sigma'_{ik}(t) \sigma'_{jk}(t)}{\phi''(b_k(t) - b_k(0))} - \dfrac{a_k(0)^2 \sigma'_{ik}(0) \sigma'_{jk}(0)}{\phi''(0)} \right|; \\[2ex] I_4 = \dfrac{1}{n} \left| \dfrac{1}{\phi''(d(t) - d(0))} - \dfrac{1}{\phi''(0)} \right|. \end{cases} \tag{23}$$

Notice that according to Assumption (1), $\frac{1}{\phi''(\cdot)}$ is continuous and hence is uniformly continuous on $[-1, 1]$. Then by noticing $\|\theta(t) - \theta(0)\|_\infty = O(n^{-1/2})$, we have for large enough $n$,

$$\max_{k \in [3n+1]} \left| \frac{1}{\phi''(\theta_k(t) - \theta_k(0))} - \frac{1}{\phi''(0)} \right| = o_p(1). \tag{24}$$

It immediately follows that $I_4 = o_p(1)$.

For $I_1$, notice that, according to the law of large numbers,

$$\operatorname{plim}_{n \to \infty} \frac{1}{n} \sum_{k \in [n]} |\sigma_{ik}\sigma_{jk}| = E[\sigma(\mathcal{W}^T x_i - \mathcal{B})\sigma(\mathcal{W}^T x_j - \mathcal{B})].$$

Therefore, we have $\frac{1}{n} \sum_{k \in [n]} |\sigma_{ik}\sigma_{jk}| = O_p(1)$. Then we have

$$\begin{aligned} I_1 &\leq \frac{1}{n} \sum_{k=1}^{n} \Big( |\sigma_{ik}(t) - \sigma_{ik}(0)| |\sigma_{jk}(t)| \beta + |\sigma_{ik}(0)| |\sigma_{jk}(t) - \sigma_{jk}(0)| \beta \\ &\quad + |\sigma_{ik}(0)| |\sigma_{jk}(0)| \Big| \frac{1}{\phi''(a_k(t) - a_k(0))} - \frac{1}{\phi''(0)} \Big| \Big) \\ &\leq \frac{1}{n} \sum_{k \in [n]} \Big( O_p(n^{-1}) O_p(\sqrt{n}) \beta + O_p(n^{-1}) O_p(\sqrt{n}) \beta \Big) + o_p(1) \cdot \frac{1}{n} \sum_{k \in [n]} |\sigma_{ik}\sigma_{jk}| \\ &= o_p(1). \end{aligned}$$

For $I_2$, notice that $z \mapsto z^2$ is Lipschitz continuous on $[-2K, 2K]$. Therefore, we have

$$\max_{k \in [n]} |a_k(t)^2 - a_k(0)^2| \le O_p(1)\|\boldsymbol{a}(t) - \boldsymbol{a}(0)\|_\infty = o_p(1).$$

Then we have

$$
\begin{aligned}
I_2 &\le \frac{B_x^2}{n} \sum_{k,r} \Big( a_k(0)^2 \beta |\sigma'_{ik}(t)\sigma'_{jk}(t) - \sigma'_{ik}(0)\sigma'_{ik}(0)| \\
&\quad + a_k(0)^2 \Big| \frac{1}{\phi''(w_{k,r}(t) - w_{k,r}(0))} - \frac{1}{\phi''(0)} \Big| + |a_k(t)^2 - a_k(0)^2|\beta \Big) \\
&= \frac{B_x^2 d}{n} \Big( 2\beta\|\boldsymbol{a}(0)\|_2^2 + o_p(1)\|\boldsymbol{a}(0)\|_2^2 + \beta n \cdot o_p(1) \Big) \\
&= o_p(1).
\end{aligned}
$$

Using a technique similar to that applied for $I_2$ above, we have $I_3 = o_p(1)$. Therefore, with a union bound, we have

$$\|H(t) - H(0)\|_2 \le \|H(t) - H(0)\|_F \le \sum_{i,j \in [m]} |H_{ij}(t) - H_{ij}(0)| = o_p(1). \tag{25}$$

Then with large enough $n$ we can bound the least eigenvalue of $H(t)$ as

$$\lambda_{\min}(H(t)) \ge \lambda_{\min}(H(0)) - \|H(t) - H(0)\|_2 > \frac{\lambda_0}{4\phi''(0)}. \tag{26}$$

Therefore we have

$$\frac{\mathrm{d}}{\mathrm{d}t}\|\boldsymbol{f}(t) - \boldsymbol{y}\|_2^2 \le -\frac{\eta_0 \lambda_0}{2\phi''(0)}\|\boldsymbol{f}(t) - \boldsymbol{y}\|_2^2$$

and it follows that

$$\|\boldsymbol{f}(t) - \boldsymbol{y}\|_2 \le \exp(-\frac{\eta_0 \lambda_0}{4\phi''(0)}t)\|\boldsymbol{f}(0) - y\|_2. \tag{27}$$

Notice that

$$\|J_{a_k}\boldsymbol{f}(t)\|_2 \le \sqrt{m}\|J_{a_k}\boldsymbol{f}\|_\infty \le \sqrt{m}(B_x\|w_k(t)\|_2 + |b_k(t)|) \le 2K(B_x + 1)\sqrt{mn}. \tag{28}$$

Then with large enough $n$ and for all $k \in [n]$, we have

$$
\begin{aligned}
|\frac{\mathrm{d}}{\mathrm{d}t}a_k(t)| &\le \frac{\eta_0}{n}\|Q(t)\|_2\|J_{a_k}\boldsymbol{f}(t)\|_2\|\boldsymbol{f}(t) - \boldsymbol{y}\|_2 \\
&\le \frac{\eta_0}{n}\beta \cdot 2K(B_x + 1)\sqrt{mn} \cdot \exp(-\frac{\eta_0 \lambda_0}{4\phi''(0)}t)\|\boldsymbol{f}(0) - \boldsymbol{y}\|_2.
\end{aligned}
\tag{29}
$$

Hence,

$$|a_k(t) - a_k(0)| \le \int_0^t |\frac{\mathrm{d}}{\mathrm{d}t}a_k(s)|\mathrm{d}s < \frac{8\phi''(0)\beta\sqrt{m}(B_x + 1)K\|\boldsymbol{f}(0) - \boldsymbol{y}\|_2}{\lambda_0}n^{-1/2} = C_1 n^{-1/2}. \tag{30}$$

Note that with large enough $n$,

$$
\begin{cases}
\|J_{w_{k,r}}\boldsymbol{f}(t)\|_2 \le \sqrt{m}B_x|a_k(t)| \le 2K\sqrt{m}B_x, & \forall k \in [n], r \in [d]; \\
\|J_{b_k}\boldsymbol{f}(t)\|_2 \le \sqrt{m}|a_k(t)| \le 2K\sqrt{m}, & \forall k \in [n]; \\
\|J_d\boldsymbol{f}(t)\|_2 = \sqrt{m}.
\end{cases}
\tag{31}
$$

Comparing (31) to (28) and using a similar computations as in (29) and (30), we conclude that $\tau(t_1) < C_1$, which yields a contradiction. Thus $t_1 = \infty$ and $\tau(t) \le C_1$ for all $t \ge 0$. Meanwhile, notice that (26) and (27) hold for all $t \ge 0$, which gives all the claimed results. $\qquad\square$

We point out that Proposition 14 can be extended to the case where only output weights are trained, without requiring additional efforts. We present the results in the following proposition and omit the proof for brevity.

**Proposition 15.** *Assume the same assumptions in Proposition 14 and consider the mirror flow of training only the output weights of the networks, as described in* (10). *Let $C_1, C_2$ be the constants defined in* (20). *Then with high probability over initialization the following holds:*

$$\sup_{t \geq 0} \|\tilde{\theta}(t) - \hat{\theta}\|_\infty \leq C_1 n^{-1}, \quad \lim_{n \to \infty} \sup_{t \geq 0} \|\tilde{H}(t) - \tilde{H}(0)\|_2 = 0,$$

*and*

$$\|\tilde{\boldsymbol{f}}(t) - \boldsymbol{y}\|_2 \leq \exp(-\eta_0 C_2 t) \|\boldsymbol{f}(0) - \boldsymbol{y}\|_2, \ \forall t \geq 0.$$

Next we show that the network obtained by training only the output weights can be well-approximated by the network obtained by training all parameters.

**Proposition 16.** *Under the same assumption in Proposition 14, given any $x \in \mathbb{R}^d$, with high probability over initialization,*

$$\lim_{n \to \infty} \sup_{t \geq 0} |f(x, \theta(t)) - f(x, \tilde{\theta}(t))| = 0.$$

**Proof.** [Proof of Proposition 16] For $i \in [m]$ and $k \in [n]$, let $\sigma'_{ik}(t) = \sigma'(w_k(t)^T x_i - b_k(t))$. Under initialization 2, we have $\|\boldsymbol{a}(0)\|_2 = O_p(1)$ and thus

$$
\begin{aligned}
|H_{ij}(0) - \tilde{H}_{ij}(0)| &\leq \frac{1}{n\phi''(0)} \sum_{k \in [n]} |a_k(0)|^2 |\sigma'_{ik}(0)\sigma'_{jk}(0)| |x_i^T x_j + 1| + \frac{1}{n\phi''(0)} \\
&\leq \frac{B_x^2 + 1}{n\phi''(0)} \|\boldsymbol{a}(0)\|_2 + O_p(n^{-1}) \\
&= O_p(n^{-1}).
\end{aligned}
\tag{32}
$$

Note the size of $H(0)$ and $\tilde{H}(0)$ is independent of $n$. Therefore, $\|H(0) - \tilde{H}(0)\|_2 = o_p(1)$. According to Proposition 14 and Proposition 15, we have

$$\sup_{t \geq 0} \|H(t) - \tilde{H}(t)\|_2 \leq \sup_{t \geq 0} \|H(t) - H(0)\|_2 + \sup_{t \geq 0} \|\tilde{H}(t) - \tilde{H}(0)\|_2 + \|H(0) + \tilde{H}(0)\|_2 = o_p(1).$$

$$\tag{33}$$

In the following, we use the notation $\Delta = \sup_{t \geq 0} \|H(t) - \tilde{H}(t)\|_2$. Now we show $\boldsymbol{f}(t)$ remains close to $\tilde{\boldsymbol{f}}(t)$. Let $\boldsymbol{r}(t) = \boldsymbol{f}(t) - \tilde{\boldsymbol{f}}(t)$ and $C_2$ be the constant in Proposition 14. Consider the function $u(t) = \|\boldsymbol{r}(t)\|_2$ defined on $t \geq 0$. Clearly, $u(t)$ is continuous. Since $u(t) = 0$ if and only if $\boldsymbol{r}(t) = 0$, we have that $u(t)$ is differentiable whenever $u(t) \neq 0$. Note by Proposition 14, we have that, with high probability, $\lambda_{\min}(H(t)) \geq C_2$ for all $t \geq 0$. Then when $u(t) \neq 0$, we have

$$
\begin{aligned}
\frac{\mathrm{d}}{\mathrm{d}t}(e^{C_2\eta_0 t} u(t))^2 &= \frac{\mathrm{d}}{\mathrm{d}t}\|e^{C_2\eta_0 t}\boldsymbol{r}(t)\|_2^2 \\
&= 2e^{C_2\eta_0 t}\boldsymbol{r}(t)^T \frac{\mathrm{d}}{\mathrm{d}t}(e^{C_2\eta_0 t}\boldsymbol{r}(t)) \\
&= 2e^{2C_2\eta_0 t}\boldsymbol{r}(t)^T \left(C_2\eta_0\boldsymbol{r}(t) - \eta_0 H(t)(\boldsymbol{f}(t) - \boldsymbol{y}) + \eta_0 \tilde{H}(t)(\tilde{\boldsymbol{f}}(t) - \boldsymbol{y})\right) \\
&= 2\eta_0 e^{2C_2\eta_0 t}\left(\boldsymbol{r}(t)^T (\tilde{H}(t) - H(t))(\tilde{\boldsymbol{f}}(t) - \boldsymbol{y}) - \boldsymbol{r}(t)^T (H(t) - C_2 I_n)\boldsymbol{r}(t)\right) \\
&\leq 2\eta_0 e^{2C_2\eta_0 t}\boldsymbol{r}(t)^T (\tilde{H}(t) - H(t))(\tilde{\boldsymbol{f}}(t) - \boldsymbol{y}) \\
&\leq 2\eta_0 e^{2C_2\eta_0 t} u(t) \|\tilde{H}(t) - H(t)\|_2 \|\tilde{\boldsymbol{f}}(t) - \boldsymbol{y}\|_2.
\end{aligned}
\tag{34}
$$

Meanwhile, we have

$$\frac{\mathrm{d}}{\mathrm{d}t}(e^{C_2\eta_0 t} u(t))^2 = 2e^{C_2\eta_0 t} u(t) \frac{\mathrm{d}}{\mathrm{d}t}(e^{C_2\eta_0 t} u(t))$$

and hence by (34)

$$\frac{\mathrm{d}}{\mathrm{d}t}(e^{C_2\eta_0 t} u(t)) \leq \eta_0 e^{C_2\eta_0 t}\|\tilde{H}(t) - H(t)\|_2 \|\tilde{\boldsymbol{f}}(t) - \boldsymbol{y}\|_2.$$

Now for any $t \geq 0$, consider $t' = \inf\{s \in [0, t] \colon \boldsymbol{r}(s) \neq 0\}$. Then we have

$$e^{C_2\eta_0 t} u(t) \leq \int_{t'}^{t} \eta_0 e^{C_2\eta_0 s} \|\tilde{H}(s) - H(s)\|_2 \|\tilde{\boldsymbol{f}}(s) - \boldsymbol{y}\|_2 \mathrm{d}s.$$

Hence for all $t \geq 0$, we have

$$
\begin{aligned}
\|\boldsymbol{r}(t)\|_2 &\leq e^{-C_2\eta_0 t} \int_{t'}^{t} \eta_0 e^{C_2\eta_0 s} \|\tilde{H}(s) - H(s)\|_2 \|\tilde{\boldsymbol{f}}(s) - \boldsymbol{y}\|_2 \mathrm{d}s \\
&\leq \eta_0 e^{-C_2\eta_0 t} \Delta \int_{t'}^{t} \|\boldsymbol{f}(0) - \boldsymbol{y}\|_2 \mathrm{d}s \\
&\leq \eta_0 e^{-C_2\eta_0 t} \|\boldsymbol{f}(0) - \boldsymbol{y}\|_2 t \Delta.
\end{aligned}
\tag{35}
$$

Given $x \in \mathbb{R}^d$, by the chain rule we have:

$$
\begin{aligned}
\frac{\mathrm{d}}{\mathrm{dt}} f(x, \theta(t)) &= -\eta_0 H(t, x)(\boldsymbol{f}(t) - \boldsymbol{y}), \\
\text{where } H(t, x) &\triangleq \frac{1}{n} J_\theta f(x, \theta(t))(\nabla^2 \Phi(\theta(t)))^{-1} J_\theta \boldsymbol{f}(t)^T
\end{aligned}
\tag{36}
$$

and

$$
\begin{aligned}
\frac{\mathrm{d}}{\mathrm{dt}} f(x, \tilde{\theta}(t)) &= -\eta_0 \tilde{H}(t, x)(\tilde{\boldsymbol{f}}(t) - \boldsymbol{y}), \\
\text{where } \tilde{H}(t, x) &\triangleq \frac{1}{n} J_{\boldsymbol{a}} f(x, \hat{\theta})(\nabla^2 \Phi(\tilde{\boldsymbol{a}}(t)))^{-1} J_{\boldsymbol{a}} \boldsymbol{f}(0)^T.
\end{aligned}
\tag{37}
$$

Similar to (22) and (23), one can show $\sup_{t \geq 0} \|H(t, x) - H(0, x)\|_2 = o_p(1)$ and $\sup_{t \geq 0} \|\tilde{H}(t, x) - \tilde{H}(0, x)\|_2 = o_p(1)$. Meanwhile, similar to (32), one can show $\|H(0, x) - \tilde{H}(0, x)\|_2 = o_p(1)$. Therefore, we have

$$\Delta_x \triangleq \sup_{t \geq 0} \|H(t, x) - \tilde{H}(t, x)\|_2 = o_p(1). \tag{38}$$

Notice that for both ReLU and absolute value activation, we have $|\sigma(z)| \leq |z|$ and $|\sigma'(z)| \leq 1$. Then for large enough $n$ we have

$$
\begin{aligned}
\|J_\theta \boldsymbol{f}(0)\|_F^2 &= \sum_{i=1}^{m} \left( 1 + \sum_{k=1}^{n} |\sigma(w_k(0)^T x_i - b_k(0))|^2 \right. \\
&\qquad \left. + \sum_{k=1}^{n} |a_k(0)|^2 |\sigma'(w_k(0)^T x_i - b_k(0))|^2 (\|x_i\|_2^2 + 1) \right) \\
&\leq m + \sum_{i=1}^{m} \sum_{k=1}^{n} \left( 2\|w_k(0)\|_2^2 \|x_i\|_2^2 + 2|b_k(0)|^2 + |a_k(0)|^2 (\|x_i\|_2^2 + 1) \right) \\
&\leq m(1 + 2B_x^2 + 2\|\boldsymbol{b}(0)\|_2^2 + (B_x^2 + 1)\|\boldsymbol{a}(0)\|_2^2).
\end{aligned}
\tag{39}
$$

Using a concentration inequality for sub-Gaussian random variables, we have $\|\boldsymbol{b}(0)\|_2$ and $\|\sqrt{n}\boldsymbol{a}(0)\|_2$ can be bounded by $C\sqrt{n}$ with some constant $C > 0$. Therefore, by (39) we have $\|J_\theta \boldsymbol{f}(0)\|_2 = O_p(\sqrt{n})$. Note a similar bound applies to $\|J_\theta f(x, \hat{\theta})\|_2$. Then for large enough $n$ we have

$$\|H(0, x)\|_2 \leq \frac{1}{n} \|J_\theta f(x, \hat{\theta})\|_2 \cdot \frac{1}{\phi''(0)} \cdot \|J_\theta \boldsymbol{f}(0)\|_2 = O_p(1).$$

Then using (35) and (38) for $t \geq 0$ we have that

$$
\begin{aligned}
|f(x, \theta(t)) - f(x, \tilde{\theta}(t))| &\leq \int_0^t \left| \frac{\mathrm{d}}{\mathrm{d}s}(f(x, \theta(s)) - f(x, \tilde{\theta}(s))) \right| \mathrm{d}s \\
&\leq \int_0^t \left| \eta_0 H(s, x)(\boldsymbol{f}(s) - \boldsymbol{y}) - \eta_0 \tilde{H}(s, x)(\tilde{\boldsymbol{f}}(s) - \boldsymbol{y}) \right| \mathrm{d}s \\
&\leq \int_0^t \eta_0 (\|H(0, x)\|_2 + \|H(s, x) - H(0, x)\|_2) \|\boldsymbol{r}(s)\|_2 \mathrm{d}s \\
&\quad + \int_0^t \eta_0 \|H(s, x) - \tilde{H}(s, x)\|_2 \|\tilde{\boldsymbol{f}}(s) - \boldsymbol{y}\|_2 \mathrm{d}s \\
&\leq O_p(1)\Delta \int_0^t e^{-C_2 \eta_0 s} s \, \mathrm{d}s + \Delta_x O_p(1) \int_0^t e^{-C_2 \eta_0 s} \mathrm{d}s \\
&\leq O_p(1)\Delta + O_p(1)\Delta_x \\
&= o_p(1),
\end{aligned}
$$

which gives the claimed results. $\qquad\square$

## A.2 Implicit bias in parameter space

As we discussed in Section 3.3, the final parameter of the networks whose only output weights are trained is the solution to the following problem:

$$
\min_{\tilde{\boldsymbol{a}} \in \mathbb{R}^n} \ D_\Phi(\tilde{\boldsymbol{a}}, \tilde{\boldsymbol{a}}(0)) \quad \text{s.t.} \ \sum_{k \in [n]} \tilde{a}_k [w_k(0)^T x_i - b_k(0)]_+ = y_i, \ \forall i \in [m]. \tag{40}
$$

In the following result, we characterize the limit of the solution to (40) when $n$ tends to infinity.

**Proposition 17.** *Let $g_n$ and $g$ defined as follows:*

$$
\begin{cases}
g_n(x, \alpha) = \displaystyle\int \alpha(w, b)[w^T x - b]_+ \, \mathrm{d}\mu_n \\
g(x, \alpha) = \displaystyle\int \alpha(w, b)[w^T x - b]_+ \, \mathrm{d}\mu.
\end{cases} \tag{41}
$$

*Assume $\bar{\boldsymbol{a}}$ is the solution to (40) where the potential $\Phi$ satisfies Assumption 1. Let $\Theta_{\mathrm{ReLU}} = \{\alpha \in C(\mathrm{supp}(\mu)) : \alpha \text{ is uniformly continuous}\}$. Assume $\bar{\alpha}_n \in \Theta_{\mathrm{ReLU}}$ satisfies that $\bar{\alpha}_n(w_k(0), b_k(0)) = n(\bar{a}_k - a_k(0))$ for all $k \in [n]$. Assume $\bar{\alpha}$ is the solution of the following problem:*

$$
\min_{\alpha \in \Theta_{\mathrm{ReLU}}} \int (\alpha(w, b))^2 \mathrm{d}\mu, \quad \text{s.t.} \ g(x_i, \alpha) = y_i - f(x_i, \hat{\theta}), \ \forall i \in [m]. \tag{42}
$$

*Then given any $x \in \mathbb{R}^d$, with high probability over initialization, $\lim_{n \to \infty} |g_n(x, \bar{\alpha}_n) - g(x, \bar{\alpha})| = 0$.*

**Proof.** [Proof of Proposition 17] Recall $\{w_k(0), b_k(0)\}_{k=1}^n$ denote the sampled initial input weights and biases, and $\mu_n$ denotes the associated empirical measure. We use the notation $\|\cdot\|_n$ for the $L^2(\mu_n)$-seminorm in $\Theta_{\mathrm{ReLU}}$, i.e., for $\alpha \in \Theta_{\mathrm{ReLU}}$,

$$
\|\alpha\|_n^2 = \frac{1}{n} \sum_{k=1}^n \big(\alpha(w_k(0), b_k(0))\big)^2.
$$

We write $\Theta_{\mathrm{ReLU}}/_{\|\cdot\|_n}$ for the quotient space of $\Theta_{\mathrm{ReLU}}$ by the subspace of functions $\alpha$ with $\|\alpha\|_n = 0$. In the sequel, we will use the notation $\alpha$ to represent the corresponding element (equivalence class) in $\Theta_{\mathrm{ReLU}}/_{\|\cdot\|_n}$, if there is no ambiguity. Let $\langle \cdot, \cdot \rangle_n$ denote the inner-product in $\Theta_{\mathrm{ReLU}}/_{\|\cdot\|_n}$ defined as: for $\alpha, \alpha' \in \Theta_{\mathrm{ReLU}}/_{\|\cdot\|_n}$,

$$
\langle \alpha, \alpha' \rangle_n = \frac{1}{n} \sum_{i=1}^n \alpha\big(w_k(0), b_k(0)\big) \alpha'\big(w_k(0), b_k(0)\big).
$$

Note equipped with $\langle \cdot, \cdot \rangle_n$, $\Theta_{\mathrm{ReLU}}/_{\|\cdot\|_n}$ becomes a finite dimensional Hilbert space and is isometric to $\mathbb{R}^n$.

Notice that for any $\boldsymbol{a} \in \mathbb{R}^n$ and $\alpha$ that satisfies $\alpha(w_k(0), b_k(0)) = n(a_k - a_k(0))$ for $k \in [n]$, we have

$$D_\Phi(\boldsymbol{a}, \boldsymbol{a}(0)) = \sum_{k \in [n]} \left( \phi(a_k - a_k(0)) - \phi(0) - \phi'(0)(a_k - a_k(0)) \right)$$

$$= n \int \phi(\frac{\alpha(w, b)}{n}) - \phi(0) - \phi'(0)\frac{\alpha(w, b)}{n} \mathrm{d}\mu_n.$$

Define $\sigma_i(w, b) = [w^T x_i - b]_+ \in \Theta_{\mathrm{ReLU}}$. Since $\bar{\boldsymbol{a}}$ is the solution to (40), we have that $\bar{\alpha}_n$ solves

$$\min_{\alpha \in \Theta_{\mathrm{ReLU}}} L_n(\alpha) \quad \text{s.t. } \langle \alpha, \sigma_i \rangle_n = y_i - f(x_i, \hat{\theta}), \ i \in [m],$$
$$\text{where } L_n(\alpha) \triangleq n^2 \int \phi(\frac{\alpha(w, b)}{n}) - \phi(0) - \phi'(0)\frac{\alpha(w, b)}{n} \mathrm{d}\mu_n. \tag{43}$$

At the same time, let $\bar{\alpha}'_n$ be the solution to the following problem:

$$\min_{\alpha \in \Theta_{\mathrm{ReLU}}} L'_n(\alpha) \quad \text{s.t. } \langle \alpha, \sigma_i \rangle_n = y_i - f(x_i, \hat{\theta}), \ i \in [m],$$
$$\text{where } L'_n(\alpha) \triangleq n^2 \int \frac{\phi''(0)}{2}(\frac{\alpha(w, b)}{n})^2 \mathrm{d}\mu_n. \tag{44}$$

Note the objectives and constraints in problem (43) and problem (44) are essentially defined in $\Theta_{\mathrm{ReLU}}/_{\|\cdot\|_n}$. Specifically, for any $\alpha, \alpha' \in \Theta_{\mathrm{ReLU}}$, if $\|\alpha - \alpha'\|_n = 0$, then $\alpha$ and $\alpha'$ are equivalent with respect to feasibility and objective function value in both problems. Therefore, we shall address these two problems in $\Theta_{\mathrm{ReLU}}/_{\|\cdot\|_n}$. Note $L_n$ and $L'_n$ are strictly convex in $\Theta_{\mathrm{ReLU}}/_{\|\cdot\|_n}$. Hence, $\bar{\alpha}_n$ and $\bar{\alpha}'_n$ are unique in the $L^2(\mu_n)$ sense.

Consider the Taylor expansion with Peano remainder: $\phi(z) = \phi(0) + \phi'(0)z + \frac{\phi''(0)}{2}z^2 + R(z)z^2$ where $\lim_{z \to 0} R(z) = 0$. Then we have

$$|L_n(\alpha) - L'_n(\alpha)| = |\int R(\frac{\alpha(w, b)}{n})\alpha^2(w, b)\mathrm{d}\mu_n|$$

$$\leq \max_{k \in [n]} R\left(\frac{\alpha(w_k(0), b_k(0))}{n}\right) \|\alpha\|_n^2.$$

By Theorem 11 and Proposition 15, $\bar{\alpha}_n$ and $\bar{\alpha}'_n$ are determined by the limit points of mirror flow (10) with potential $\Phi(\theta) = \sum_k \phi(\theta_k - \hat{\theta}_k)$ and $\Phi(\theta) = \sum_k (\theta_k - \hat{\theta}_k)^2$, respectively. Specifically, we have that $\max_{k \in [n]} \bar{\alpha}_n(w_k(0), b_k(0))/n = \max_{k \in [n]} \bar{a}_k - a_k(0) = O_p(n^{-1/2})$ and that $\|\bar{\alpha}_n\|_n^2 = n\|\bar{\boldsymbol{a}} - \boldsymbol{a}(0)\|_2^2 = O_p(1)$. Similarly, we have $\max_{k \in [n]} \bar{\alpha}'_n(w_k(0), b_k(0))/n = O_p(n^{-1})$ and $\|\bar{\alpha}'_n\|_n^2 = O_p(1)$. Therefore, we have

$$|L_n(\bar{\alpha}_n) - L'_n(\bar{\alpha}_n)|, \ |L_n(\bar{\alpha}'_n) - L'_n(\bar{\alpha}'_n)| = o_p(1). \tag{45}$$

Using (45) and noting the optimality of $\bar{\alpha}_n$ and $\bar{\alpha}'_n$, we have

$$L'_n(\bar{\alpha}_n) \leq L_n(\bar{\alpha}_n) + o_p(1) \leq L_n(\bar{\alpha}'_n) + o_p(1) \leq L'_n(\bar{\alpha}'_n) + o_p(1). \tag{46}$$

Notice that, in the sense of Fréchet differentialbility, $L'_n$ is twice differentiable and $\phi''(0)$-strongly convex. Thus we have

$$L'_n(\bar{\alpha}_n) \geq L'_n(\bar{\alpha}'_n) + \langle \nabla L'_n(\bar{\alpha}'_n), \bar{\alpha}_n - \bar{\alpha}'_n \rangle_n + \frac{\phi''(0)}{2}\|\bar{\alpha}_n - \bar{\alpha}'_n\|_n^2, \tag{47}$$

where $\nabla L'_n$ denotes the Fréchet differential. Note $\nabla L'_n(\bar{\alpha}'_n) = \phi''(0)\bar{\alpha}'_n$. Then combining (46) and (47) gives

$$\langle \phi''(0)\bar{\alpha}'_n, \bar{\alpha}_n - \bar{\alpha}'_n \rangle_n + \frac{\phi''(0)}{4}\|\bar{\alpha}_n - \bar{\alpha}'_n\|_n^2 \leq o_p(1).$$

By the first-order optimality condition for problem (44), $\bar{\alpha}'_n$ can be written as linear combination of $\{\sigma_i\}_{i=1}^m$. On the other hand, by the constraints we have for all $i \in [m]$,

$$\langle \bar{\alpha}_n - \bar{\alpha}'_n, \sigma_i \rangle_n = y_i - f(x_i, \theta(0)) - (y_i - f(x_i, \theta(0))) = 0.$$

Therefore, $\langle \bar{\alpha}'_n, \bar{\alpha}_n - \bar{\alpha}'_n \rangle_n = 0$ and hence

$$\|\bar{\alpha}_n - \bar{\alpha}'_n\|_n = o_p(1).$$

For a given data $x \in \mathbb{R}^d$, notice that

$$\|\sigma(w^T x - b)\|_n^2 \leq \frac{1}{n} \sum_{k=1}^n |w_k(0)^T x - b_k(0)|^2 \leq \|x\| + \frac{1}{n}\|\boldsymbol{b}(0)\|_2^2 = O_p(1).$$

Therefore, we have

$$\begin{aligned}
\left| g_n(x, \bar{\alpha}_n) - g_n(x, \bar{\alpha}'_n) \right| &= \left| \langle \bar{\alpha}_n(w, b) - \bar{\alpha}'_n(w, b), \sigma(w^T x - b) \rangle_n \right| \\
&\leq \|\bar{\alpha}_n - \bar{\alpha}'_n\|_n \|\sigma(w^T x - b)\|_n \\
&= o_p(1).
\end{aligned} \tag{48}$$

Now it remains to show that $|g_n(x, \bar{\alpha}'_n) - g(x, \bar{\alpha})| = o_p(1)$. By the optimality condition for problem (44), $\bar{\alpha}'_n$ can be chosen as follows

$$\bar{\alpha}'_n(w, b) = \sum_{i=1}^m \lambda_i^{(n)} \sigma_i(w, b), \quad \forall (w, b) \in \mathrm{supp}(\mu),$$

where the Lagrangian multipliers $(\lambda_i^{(n)})_{i=1}^m$ are determined by the following random linear system

$$\Big\langle \sum_{i=1}^m \lambda_i^{(n)} \sigma_i, \ \sigma_j \Big\rangle_n = y_j, \quad j = 1, \ldots, m. \tag{49}$$

Note that $\bar{\alpha}'_n$ is uniformly continuous.

Similarly, by the optimality condition for problem (42), $\bar{\alpha}$ can be given by

$$\bar{\alpha}(w, b) = \sum_{i=1}^m \lambda_i \sigma_i(w, b),$$

where the Lagrangian multipliers $(\lambda_i)_{i=1}^m$ are determined by the following determinant linear system

$$\Big\langle \sum_{i=1}^m \lambda_i \sigma_i, \ \sigma_j \Big\rangle = y_j, \quad j = 1, \ldots, m. \tag{50}$$

Here $\langle \cdot, \cdot \rangle$ denotes the usual inner product in $L^2(\mu)$, i.e. $\langle f, g \rangle = \int f g \, \mathrm{d}\mu$. Note that $\{\sigma_i\}_{i \in [m]}$ are all continuous functions and that $\mu$ is compactly supported. Then by law of large number, we have $\langle \sigma_i, \sigma_j \rangle_n$ converges in probability to $\langle \sigma_i, \sigma_j \rangle$ for all $i, j \in [m]$, which means the coefficients in system (49) converges in probability to the coefficients in system (50). It then follows that

$$\|\lambda^{(n)} - \lambda\|_\infty = o_p(1). \tag{51}$$

Given a fixed $x \in \mathbb{R}^d$, recall that

$$\begin{cases} g_n(x, \bar{\alpha}) = \int \bar{\alpha}(w, b)[w^T x - b]_+ \mathrm{d}\mu_n; \\ g(x, \bar{\alpha}) = \int \bar{\alpha}(w, b)[w^T x - b]_+ \mathrm{d}\mu. \end{cases}$$

Noting that $\bar{\alpha}$ and the ReLU activation function is uniformly continuous and that $\mu$ is sub-Gaussian, by law of large number we have

$$|g_n(x, \bar{\alpha}) - g(x, \bar{\alpha})| = o_p(1). \tag{52}$$

On the other side, we have

$$|g_n(x, \bar{\alpha}) - g_n(x, \bar{\alpha}'_n)| \leq \int \left( \sum_{i=1}^m |\lambda_i - \lambda_i^{(n)}| \sigma_i(w, b) \right) [w^T x - b]_+ \mathrm{d}\mu_n$$

$$\leq \|\lambda - \lambda^{(n)}\|_\infty \int \left( \sum_{i=1}^m \sigma_i(w, b) \right) [w^T x - b]_+ \mathrm{d}\mu_n.$$

Again by law of large number,

$$\int [w^T x - b]_+ \sum_{j=1}^m \sigma_j(w, b) \, \mathrm{d}\mu_n \xrightarrow{P} \int [w^T x - b]_+ \sum_{j=1}^m \sigma_j(w, b) \, \mathrm{d}\mu.$$

Hence, with large enough $n$ and with high probability, $\int [w^T x - b]_+ \sum_{j=1}^m \sigma_j(w, b) \, \mathrm{d}\mu_n$ can be bounded by a finite number independent of $n$. Therefore, by (51) we have

$$|g_n(x, \bar{\alpha}) - g_n(x, \bar{\alpha}'_n)| = o_p(1).$$

Finally, according to (48) and (52), we obtain

$$|g(x, \bar{\alpha}) - g_n(x, \bar{\alpha}_n)| \leq |g(x, \bar{\alpha}) - g_n(x, \bar{\alpha})| + |g_n(x, \bar{\alpha}) - g_n(x, \bar{\alpha}'_n)| + |g_n(x, \bar{\alpha}'_n) - g_n(x, \bar{\alpha}_n)|$$
$$= o_p(1),$$

which concludes the proof. $\qquad\qquad\square$

Note the above proof can be directly applied to networks with absolute value activations. Then Proposition 17 for networks with absolute value activations is stated as follows.

**Proposition 18** (**Proposition 17 for networks absolute value activations**). *Let $g_n$ and $g$ defined as follows:*

$$\begin{cases} g_n(x, \alpha) = \int \alpha(w, b) |w^T x - b| \, \mathrm{d}\mu_n \\ g(x, \alpha) = \int \alpha(w, b) |w^T x - b| \, \mathrm{d}\mu. \end{cases} \tag{53}$$

*For a potential $\Phi$ that satisfies Assumption 1, assume $\bar{\boldsymbol{a}}$ is the solution to the following problem:*

$$\min_{\tilde{\boldsymbol{a}} \in \mathbb{R}^n} D_\Phi(\tilde{\boldsymbol{a}}, \tilde{\boldsymbol{a}}(0)) \quad \text{s.t.} \sum_{k \in [n]} \tilde{a}_k |w_k(0)^T x_i - b_k(0)| = y_i, \; \forall i \in [m].$$

*Let $\Theta_{\mathrm{Abs}} = \{ \alpha \in C(\mathrm{supp}(\mu)) \colon \alpha \text{ is even and uniformly continuous} \}$ and $\bar{\alpha}_n \in \Theta_{\mathrm{Abs}}$ be any continuous function that satisfies $\bar{\alpha}_n(w_k(0), b_k(0)) = n(\bar{a}_k - a_k(0))$. Assume $\bar{\alpha}$ is the solution of the following problem:*

$$\min_{\alpha \in \Theta_{\mathrm{Abs}}} \int (\alpha(w, b))^2 \mathrm{d}\mu, \quad \text{s.t.} \; g(x_i, \alpha) = y_i - f(x_i, \hat{\theta}), \; \forall i \in [m],$$

*Then given any $x \in \mathbb{R}^d$, with high probability over initialization, $\lim_{n \to \infty} |g_n(x, \bar{\alpha}_n) - g(x, \bar{\alpha})| = 0$.*

### A.3 FUNCTION DESCRIPTION OF THE IMPLICIT BIAS

Assume the input dimension is one, i.e., $d = 1$. Next we aim to characterize the solution to (42) in the function space:

$$\mathcal{F}_{\mathrm{ReLU}} = \left\{ g(x, \alpha) = \int \alpha(w, b) [wx - b]_+ \mathrm{d}\mu \colon \alpha \in \Theta_{\mathrm{ReLU}} \right\}.$$

Recall the following result from classical analysis.

**Lemma 19.** *Assume that (1) $f$ is a continuous function defined on a neighborhood $U$ of $x_0 \in \mathbb{R}$; (2) $f$ is continuously differentiable on $U - \{x_0\}$; and (3) $\lim_{x \to x_0^+} f'(x) = \lim_{x \to x_0^-} f'(x) = k$ for some constant $k \in \mathbb{R}$. Then $f$ is differentiable at $x_0$ and $f'(x_0) = k$.*

**Proof.** [Proof of Lemma 19] Consider $x > x_0$ and $x \in U$. Since $f$ is continuous on $[x_0, x]$ and continuously differentiable on $(x_0, x)$, by mean value theorem there exists $x' \in (x_0, x)$ such that

$$\frac{f(x) - f(x_0)}{x - x_0} = f'(x').$$

Considering the limit process $x \to x_0$, we know $f$ is right differentiable at $x_0$ and

$$f'_+(x_0) = \lim_{x' \to x_0^+} f'(x') = k.$$

Similarly, we know $f$ is left differentiable at $x_0$ as well and

$$f'_-(x_0) = f'_+(x_0) = k.$$

This implies $f$ is differentiable at $x_0$ and $f'(x_0) = k$. $\qquad\square$

In the following result, we characterize the analytical properties of functions in $\mathcal{F}_{\text{ReLU}}$.

**Proposition 20.** *Assume $h \in \mathcal{F}_{\text{ReLU}}$. Then (1) $h$ is continuously differentiable on $\mathbb{R}$; (2) $h'(-\infty) = \lim_{x \to -\infty} h'(x)$ and $h'(\infty) = \lim_{x \to +\infty} h'(x)$ are well-defined; (3) $h$ is twice continuously differentiable almost everywhere on $\mathbb{R}$; (4) $\text{supp}(h'') \subset \text{supp}(p_{\mathcal{B}})$; and (5) $\int_{\mathbb{R}} h''(b)|b|\mathrm{d}b$ is well-defined.*

**Proof.** [Proof of Lemma 20] Assume $h(x)$ takes the form of $h(x) = \int \alpha(w, b)[wx - b]_+\mathrm{d}\mu$. Notice that $\alpha$ and ReLU activation function are uniformly continuous and that $\mu$ is sub-Gaussian. Hence, $h(x)$ is well-defined for all $x \in \mathbb{R}$.

By our assumption $(\mathcal{W}, \mathcal{B})$ is symmetric and $\mathcal{B}$ has continuous density function. Hence, we have

$$p_{\mathcal{B}}(b) = p_{\mathcal{W}, \mathcal{B}}(1, b) + p_{\mathcal{W}, \mathcal{B}}(-1, b) = p_{\mathcal{W}, \mathcal{B}}(-1, -b) + p_{\mathcal{W}, \mathcal{B}}(1, -b) = p_{\mathcal{B}}(-b),$$

which implies $p_{\mathcal{B}}$ is an even function.

Since $p_{\mathcal{B}}$ is a continuous on $\mathbb{R}$, $(\text{supp}(p_{\mathcal{B}}))^{\text{o}}$, i.e., the interior of $\text{supp}(p_{\mathcal{B}})$, is an open set. Hence, $(\text{supp}(p_{\mathcal{B}}))^{\text{o}}$ can be represented by a countable union of disjoint open intervals. Without loss of generality, we assume $\text{supp}(p_{\mathcal{B}}) = [m, M]$ where $m, M \in \{\infty, -\infty\} \cup \mathbb{R}$. We have

$$h(x) = \frac{1}{2}\int_m^M \alpha(1, b)p_{\mathcal{B}}(b)[x - b]_+\mathrm{d}b + \frac{1}{2}\int_m^M \alpha(-1, -b)p_{\mathcal{B}}(b)[b - x]_+\mathrm{d}b.$$

We first show $h$ is continuously differentiable on $\mathbb{R}$. For $x \in (\text{supp}(p_{\mathcal{B}}))^{\text{o}}$, we have that

$$h(x) = \frac{1}{2}\int_m^x \alpha(1, b)p_{\mathcal{B}}(b)(x - b)\mathrm{d}b + \frac{1}{2}\int_x^M \alpha(-1, -b)p_{\mathcal{B}}(b)(b - x)\mathrm{d}b.$$

By Leibniz integral rule, differentiating $h(x)$ with respect to $x$ gives

$$h'(x) = \frac{1}{2}\int_m^x \alpha(1, b)p_{\mathcal{B}}(b)\mathrm{d}b - \frac{1}{2}\int_x^M \alpha(-1, -b)p_{\mathcal{B}}(b)\mathrm{d}b. \tag{54}$$

Note (54) implies that $h'$ is continuous on $(\text{supp}(p_{\mathcal{B}}))^{\text{o}}$. If $M$ is a finite number, we examine the continuity of $h'$ at $x \geq M$. For $x > M$, we have

$$h(x) = \frac{1}{2}\int_m^M \alpha(1, b)p_{\mathcal{B}}(b)(x - b)\mathrm{d}b.$$

Again by Leibniz integral rule we have

$$h'(x) = \frac{1}{2}\int_m^M \alpha(1, b)p_{\mathcal{B}}(b)\mathrm{d}b. \tag{55}$$

Clearly, $h'(x)$ is continuous when $x > M$. According to (54) and (55), we have $\lim_{x \to M^+} h'(x) = \lim_{x \to M^-} h'(x)$. By Lemma (19), $h$ is differentiable at $x = M$ and $h'$ is continuous at $x = M$. Therefore, $h'(x)$ is continuous when $x \geq M$. Noticing a similar discussion can be applied to $m$, we conclude that $h'$ is continuous on $\mathbb{R}$.

Next, we show $h'(\infty)$ and $h'(-\infty)$ are well-defined. Note by (54) and (55) we have

$$h'(+\infty) = \frac{1}{2} \int_m^M \alpha(1,b)p_{\mathcal{B}}(b)\mathrm{d}b. \tag{56}$$

This integral is always well-defined since $\alpha$ is uniformly continuous and $p_{\mathcal{B}}(b)$ has heavier tail than $e^{-b^2}$. Similarly, we have

$$h'(-\infty) = -\frac{1}{2} \int_m^M \alpha(-1,-b)p_{\mathcal{B}}(b)\mathrm{d}b, \tag{57}$$

which implies $h'(-\infty)$ is also well-defined.

We then examine the second-order differentiability of $h$. For $x \in (\mathrm{supp}(p_{\mathcal{B}}))^\circ$, differentiating (54) gives:

$$\begin{aligned} h''(x) &= \frac{1}{2}\alpha(1,x)p_{\mathcal{B}}(b) + \frac{1}{2}\alpha(-1,-x)p_{\mathcal{B}}(b) \\ &= p_{\mathcal{B}}(b)\alpha^+(1,x), \end{aligned} \tag{58}$$

which implies $h$ is twice continuously differentiable on $(\mathrm{supp}(p_{\mathcal{B}}))^\circ$. For $x \in R \setminus (\mathrm{supp}(p_{\mathcal{B}}))^\circ$, notice such $x$ exists only if $m$ or $M$ is a finite number. For the case of $M < \infty$, by (55), we have $h''(x) = 0$ when $x > M$. Similarly, if $m > -\infty$, we have $h''(x) = 0$ when $x < m$.

Finally, if $m = -\infty$ or $M = \infty$, by (58) and by noticing that $\alpha$ is uniformly continuous and that $\mathcal{B}$ is sub-Gaussian, we have $\int h''(b)|b|\mathrm{d}b$ is well-defined. If both $m$ and $M$ are finite numbers, then we've just shown $h''$ is compactly supported. Therefore, $\int h''(b)|b|\mathrm{d}b$ is also well-defined. By this, we conclude the proof. □

Recall that, for any $\alpha \in \Theta_{\mathrm{ReLU}}$, $\alpha$ can be uniquely decomposed into the sum of an even function and an odd function[6]: $\alpha(w,b) = \alpha^+(w,b) + \alpha^-(w,b)$, where $\alpha^+(w,b) = (\alpha(w,b) + \alpha(-w,-b))/2$ and $\alpha^-(w,b) = (\alpha(w,b) - \alpha(-w,-b))/2$. In the following result, we present the closed-form solution for the minimal representation cost problem (17) with the cost functional described in problem (42).

**Proposition 21.** *Given $h \in \mathcal{F}_{\mathrm{ReLU}}$, consider the minimal representation cost problem*

$$\min_{\alpha \in \Theta_{\mathrm{ReLU}}} \int (\alpha(w,b))^2 \mathrm{d}\mu \quad \text{s.t. } h(x) = g(x,\alpha), \forall x \in \mathbb{R}, \tag{59}$$

*where $g(\cdot,\alpha)$ is defined in (41). The solution to problem (59), denoted by $\mathcal{R}(h)$, is given by*

$$\begin{cases} (\mathcal{R}(h))^+(1,b) &= \dfrac{h''(b)}{p_{\mathcal{B}}(b)} \\ (\mathcal{R}(h))^-(1,b) &= \dfrac{\int_{\mathbb{R}} h''(b)|b|\mathrm{d}b - 2h(0)}{E[\mathcal{B}^2]}b + h'(+\infty) + h'(-\infty)s \end{cases} \tag{60}$$

*where $h'(+\infty) = \lim_{x \to +\infty} h'(x)$ and $h'(-\infty) = \lim_{x \to -\infty} h'(x)$.*

**Proof.** [Proof of Proposition 21] Fix an $h \in \mathcal{F}_{\mathrm{ReLU}}$. We now show that the constraints $g(\cdot,\alpha) = h(\cdot)$ in problem (59) is equivalent to

$$\begin{cases} \alpha^+(1,b) = \dfrac{h''(b)}{p_{\mathcal{B}}(b)}, \\ \int \alpha^-(1,b)\mathrm{d}\mu_{\mathcal{B}} = h'(+\infty) + h'(-\infty), \\ \int \alpha^-(1,b)b\,\mathrm{d}\mu_{\mathcal{B}} = \int_{\mathbb{R}} h''(b)|b|\mathrm{d}b - 2h(0). \end{cases} \tag{61}$$

We first show (61) is sufficient for the equality $g(\cdot,\alpha) = h(\cdot)$. For $\alpha \in \Theta_{\mathrm{ReLU}}$ satisfying (61), consider the function

$$s(x) = h(x) - g(x,\alpha).$$

---

[6] Recall a function $\alpha$ on $\{-1,1\} \times [-B_b, B_b]$ is called an even function if $\alpha(w,b) = \alpha(-w,-b)$ for all $(w,b)$ in its domain; and an odd function if $\alpha(w,b) = -\alpha(-w,-b)$ for all $(w,b)$ in its domain.

Clearly, $s(x) \in \mathcal{F}_{\text{ReLU}}$. By Proposition 20, for $x \notin \text{supp}(p_{\mathcal{B}})$, we have $s''(x) = 0 - 0 = 0$. For $x \in \text{supp}(p_{\mathcal{B}})$, applying the computation given in (58), we have

$$s''(x) = h''(x) - p_{\mathcal{B}}(b)\alpha^+(1, b) = 0.$$

This implies $s'(x)$ is a piece-wise constant function on $\mathbb{R}$. According to Proposition 20, $s'(x)$ is continuous. Hence, $s'(x)$ is constant on $\mathbb{R}$. Using similar computations as presented in (56) and (57), we have that $\lim_{x \to \infty} \partial_x g(x, \alpha) + \lim_{x \to -\infty} \partial_x g(x, \alpha) = \int \alpha^-(1, b)\mathrm{d}\mu_{\mathcal{B}}$. Therefore,

$$s'(+\infty) + s'(-\infty) = h'(+\infty) + h'(-\infty) - \int \alpha^-(1, b)\mathrm{d}\mu_{\mathcal{B}} = 0.$$

Therefore, $s'(x) = 0$ for all $x \in \mathbb{R}$ and $s(x)$ is a constant function. By (61), we have

$$
\begin{aligned}
g(0, \alpha) &= \int \alpha(w, b)[-b]_+ \mathrm{d}\mu \\
&= \int \alpha^+(w, b)\frac{|-b|}{2} + \alpha^-(w, b)\frac{-b}{2}\mathrm{d}\mu \\
&= \int \alpha^+(1, b)\frac{|-b|}{2}\mathrm{d}\mu_{\mathcal{B}} + \int \alpha^-(1, b)\frac{-b}{2}\mathrm{d}\mu_{\mathcal{B}} \\
&= \int_{\mathbb{R}} h''(b)\frac{|b|}{2}\mathrm{d}b - \int \alpha^-(1, b)\frac{b}{2}\mathrm{d}\mu_{\mathcal{B}} \\
&= h(0),
\end{aligned}
\tag{62}
$$

which implies $s = 0$ and hence $h(x) = g(x, \alpha)$ for $x \in \mathbb{R}$.

We now show (61) is also necessary for the equality $g(\cdot, \alpha) = h(\cdot)$. Assume $g(\cdot, \alpha) = h(\cdot)$. It's clear from (56), (57), and (58) that $\alpha$ must satisfy the first and second equations in (61). Noting that $g(0, \alpha) = h(0)$ and according to (62), we have $\alpha$ must satisfy the third equation in (61) as well. Therefore, $g(\cdot, \alpha) = h(\cdot)$ is equivalent to (61) and the minimal representational cost problem (59) is equivalent to the following problem:

$$\min_{\alpha \in \Theta_{\text{ReLU}}} \int \alpha^2(w, b)\mathrm{d}\mu \quad \text{s.t. } \alpha \text{ satisfies (61)}. \tag{63}$$

We now explicitly solve problem (63). To simplify the notation, we let

$$
\begin{cases}
S_h = h'(+\infty) + h'(-\infty), \\
C_h = \int_{\mathbb{R}} h''(b)|b|\mathrm{d}b - 2h(0).
\end{cases}
\tag{64}
$$

Since the measure $\mu$ is symmetric, the following Pythagorean-type equality holds for any $\alpha$ in $\Theta_{\text{ReLU}}$:

$$
\begin{aligned}
\int (\alpha(w, b))^2 \mathrm{d}\mu &= \int (\alpha^+(w, b))^2 \mathrm{d}\mu + \int (\alpha^-(w, b))^2 \mathrm{d}\mu \\
&= \int (\alpha^+(1, b))^2 \mathrm{d}\mu_{\mathcal{B}} + \int (\alpha^-(1, b))^2 \mathrm{d}\mu_{\mathcal{B}}.
\end{aligned}
$$

Note in (61), $\alpha^+(1, b)$ is completely determined. Hence, to solve (63) it suffices to solve

$$\min_{\substack{\alpha^- \in \Theta_{\text{ReLU}} \\ \alpha^-: \text{ odd}}} \int (\alpha^-(w, b))^2 \mathrm{d}\mu \quad \text{s.t. }
\begin{cases}
\int \alpha^-(1, b)\mathrm{d}\mu_{\mathcal{B}} = S_h, \\
\int \alpha^-(1, b)b\mathrm{d}\mu_{\mathcal{B}} = C_h.
\end{cases}
\tag{65}$$

The first order optimality condition for the above question can be written as

$$2\alpha^-(1, b) - \lambda_1 - \lambda_2 b = 0, \tag{66}$$

where $\lambda_1, \lambda_2 \in \mathbb{R}$ are Lagrangian multipliers. Plugging in (66) to the constraints in (65) gives

$$
\begin{cases}
\frac{1}{2}\lambda_1 + \frac{\mathbb{E}[\mathcal{B}]}{2}\lambda_2 = S_h \\
\frac{\mathbb{E}[\mathcal{B}]}{2}\lambda_1 + \frac{\mathbb{E}[\mathcal{B}^2]}{2}\lambda_2 = C_h.
\end{cases}
$$

Since the measure $\mu$ is symmetric, the density function $p_\mathcal{B}$ must be an even function on $\mathbb{R}$. This implies $\mathbb{E}[\mathcal{B}] = 0$. Therefore, $\lambda_1 = 2S_h$ and $\lambda_2 = 2C_h/\mathbb{E}[\mathcal{B}^2]$. Finally, $\mathcal{R}(h)$, which is the solution to (63), is determined by

$$\begin{cases} (\mathcal{R}(h))^+(1,b) = \dfrac{h''(b)}{p_\mathcal{B}(b)} \\[2mm] (\mathcal{R}(h))^-(1,b) = \dfrac{C_h}{\mathbb{E}[\mathcal{B}^2]}b + S_h, \end{cases}$$

which concludes the proof. $\qquad\square$

Finally, with Proposition 12 and Proposition 21, we can write the implicit bias in the function space.

**Proposition 22.** *Assume $\bar\alpha$ solves (42), then $\bar{h}(x) = \int \bar\alpha(w,b)[wx-b]_+ \mathrm{d}\mu$ solves the following variational problem*

$$\min_{h \in \mathcal{F}_{\mathrm{ReLU}}} \mathcal{G}_1(h) + \mathcal{G}_2(h) + \mathcal{G}_3(h) \quad \text{s.t. } h(x_i) = y_i - f(x_i, \hat\theta), \; \forall i \in [m],$$

$$\text{where } \begin{cases} \mathcal{G}_1(h) = \displaystyle\int_{-B_b}^{B_b} \dfrac{(h''(x))^2}{p_\mathcal{B}(x)} \mathrm{d}x \\[3mm] \mathcal{G}_2(h) = (h'(+\infty) + h'(-\infty))^2 \\[2mm] \mathcal{G}_3(h) = \dfrac{1}{\mathbb{E}[\mathcal{B}^2]} \Big( \displaystyle\int_\mathbb{R} h''(b)|b|\mathrm{d}b - 2h(0) \Big)^2. \end{cases}$$

**Proof.** [Proof of Proposition 22] By Proposition 12, $\bar{h}(\cdot)$ solves the following problem

$$\min_{h \in \mathcal{F}_{\mathrm{ReLU}}} \int \big(\mathcal{R}(h)(w,b)\big)^2 \mathrm{d}\mu \quad \text{s.t. } h(x_i) = y_i - f(x_i, \hat\theta) \; \forall i \in [m].$$

Notice that

$$\int \big(\mathcal{R}(h)(w,b)\big)^2 \mathrm{d}\mu = \int ((\mathcal{R}(h))^+(w,b))^2 + ((\mathcal{R}(h))^-(w,b))^2 \mathrm{d}\mu$$
$$= \int ((\mathcal{R}(h))^+(1,b))^2 + ((\mathcal{R}(h))^-(1,b))^2 \mathrm{d}\mu_\mathcal{B}. \tag{67}$$

Recall $S_h$ and $C_h$ as defined in (64). Then plugging in (60) to (67) gives:

$$\int \big(\mathcal{R}(h)(w,b)\big)^2 \mathrm{d}\mu = \int \Big(\dfrac{h''(x)}{p_\mathcal{B}(x)}\Big)^2 \mathrm{d}\mu_\mathcal{B} + \int \Big(\dfrac{C_h}{\mathbb{E}[\mathcal{B}^2]}x + S_h\Big)^2 \mathrm{d}\mu_\mathcal{B}$$
$$= \int_{-B_b}^{B_b} \dfrac{(h''(x))^2}{p_\mathcal{B}(x)} \mathrm{d}x + \Big(\dfrac{C_h}{\mathbb{E}[\mathcal{B}^2]}\Big)^2 \int_{-B_b}^{B_b} x^2 \mathrm{d}\mu_\mathcal{B} + S_h^2 \int_{-B_b}^{B_b} \mathrm{d}\mu_\mathcal{B}$$
$$= \int_{-B_b}^{B_b} \dfrac{(h''(x))^2}{p_\mathcal{B}(x)} \mathrm{d}x + \dfrac{C_h^2}{\mathbb{E}[\mathcal{B}^2]} + S_h^2,$$

which concludes the proof. $\qquad\square$

The above results can be summarized by the following commutative diagram:

$$\begin{array}{ccccc} \Theta_{\mathrm{ReLU}} & = & \Theta_{\mathrm{Even}} & \oplus & \Theta_{\mathrm{Odd}} \\ \uparrow {\scriptstyle \mathcal{R}_\Phi} & & {\scriptstyle \mathcal{R}_\Phi}\uparrow & & {\scriptstyle \mathcal{R}_\Phi}\uparrow \\ \mathcal{C}_\Phi \Big( \mathcal{F}_{\mathrm{ReLU}} & = & \mathcal{F}_{\mathrm{Abs}} \Big)\mathcal{C}_\Phi \oplus & & \Big(\mathcal{F}_{\mathrm{Affine}}\Big)\mathcal{C}_\Phi \\ \downarrow {\scriptstyle \mathcal{G}_1+\mathcal{G}_2+\mathcal{G}_3} & & {\scriptstyle \mathcal{G}_1}\downarrow & & {\scriptstyle \mathcal{G}_2+\mathcal{G}_3}\downarrow \\ \mathbb{R} & & \mathbb{R} & & \mathbb{R}. \end{array} \tag{68}$$

We state Proposition 21 for networks with absolute value activations below. The proof can be viewed as a special case of the proof of Proposition 21 and thus is omitted.

**Proposition 23 (Proposition 21 for networks with absolute value activations).** *Let $\mathcal{F}_{\mathrm{Abs}} = \{h(x) = \int \alpha(w,b)|wx - b|\mathrm{d}\mu\colon \alpha \in C(\mathrm{supp}(\mu)), \alpha$ is even and uniformly continuous$\}$. Given $h \in \mathcal{F}_{\mathrm{Abs}}$, consider the minimal representation cost problem*

$$\min_{\alpha \in \Theta_{\mathrm{Abs}}} \int (\alpha(w,b))^2 \mathrm{d}\mu \quad \text{s.t. } h(x) = g(x,\alpha), \forall x \in \mathbb{R}. \tag{69}$$

*The solution to problem (69), denoted by $\mathcal{R}(h)$, is given by*

$$\mathcal{R}(h)(1,b) = \frac{h''(b)}{2p_{\mathcal{B}}(b)}.$$

With Proposition 14, Proposition 16, Proposition 18, and Proposition 23, we obtain Theorem 2 for networks with absolute value functions, which is stated below.

**Theorem 24 (Implicit bias of mirror flow for wide univariate absolute value network).** *Consider a two layer absolute value network (1) with $d$ input units and $n$ hidden units, where we assume $n$ is sufficiently large. Consider parameter initialization (2) and, specifically, let $p_{\mathcal{B}}(b)$ denote the density function for random variable $\mathcal{B}$. Consider any finite training data set $\{(x_i, y_i)\}_{i=1}^m$ that satisfies $x_i \neq x_j$ when $i \neq j$ and $\{\|x_i\|_2\}_{i=1}^m \subset \mathrm{supp}(p_{\mathcal{B}})$, and consider mirror flow (4) with a potential function $\Phi$ satisfying Assumption 1 and learning rate $\eta = \Theta(1/n)$. Then, there exist constants $C_1, C_2 > 0$ such that with high probability over the random parameter initialization, for any $t \geq 0$,*

$$\|\theta(t) - \hat{\theta}\|_\infty \leq C_1 n^{-1}, \quad \lim_{n \to \infty} \|H(t) - H(0)\|_2 = 0, \quad \|\boldsymbol{f}(t) - \boldsymbol{y}\|_2 \leq e^{-\eta_0 C_2 t}\|\boldsymbol{f}(0) - \boldsymbol{y}\|_2.$$

*Moreover, letting $\theta(\infty) = \lim_{t \to \infty} \theta(t)$, we have for any given $x \in \mathbb{R}^d$, that $\lim_{n \to \infty} |f(x, \theta(\infty)) - f(x, \theta_{\mathrm{GF}}(\infty))| = 0$, where $\theta_{\mathrm{GF}}(\infty)$ denotes the limiting point of gradient flow, i.e., mirror flow (4) with $\Phi(\theta) = \|\theta\|_2^2$, on the same training data and initial parameter.*

*Assuming univariate input data, i.e., $d = 1$, we have for any given $x \in \mathbb{R}$, that $\lim_{n \to \infty} |f(x, \theta(\infty)) - f(x, \hat{\theta}) - \bar{h}(x)| = 0$, where $\bar{h}(\cdot)$ is the solution to the following variational problem:*

$$\min_{h \in \mathcal{F}_{\mathrm{Abs}}} \int_{\mathrm{supp}(p_{\mathcal{B}})} \frac{(h''(x))^2}{p_{\mathcal{B}}(x)} \mathrm{d}x \quad \text{s.t. } h(x_i) = y_i - f(x_i, \hat{\theta}), \ \forall i \in [m].$$

*Here $\mathcal{F}_{\mathrm{Abs}} = \{h(x) = \int \alpha(w,b)|wx - b|\mathrm{d}\mu\colon \alpha \in C(\mathrm{supp}(\mu)), \alpha$ is even and uniformly continuous$\}$ is the space of functions that can be represented by an infinitely wide absolute value network with an even and uniformly continuous output weight function $\alpha$.*

Before ending this section, we present the proof of Proposition 12.

**Proof.** [Proof of Proposition 12] Note $\mathcal{R}_\Phi$ is a well-defined map since $\mathcal{C}_\Phi$ is strictly convex on $\Theta$. By assumption, problem (14) has a unique solution. Notice $\mathcal{R}_\Phi(g(\cdot, \bar{\alpha}))$ is feasible for (14). Hence, by the optimality of $\bar{\alpha}$, we have $\mathcal{R}_\Phi(g(\cdot, \bar{\alpha})) = \bar{\alpha}$, i.e. $\bar{\alpha}$ is the parameter with the minimal cost for $g(\cdot, \bar{\alpha})$. Now assume $\hat{h}$ solves (17). By noticing $g(\cdot, \bar{\alpha})$ is feasible for (18), we have $\mathcal{C}_\Phi \circ \mathcal{R}_\Phi(\hat{h}) \leq \mathcal{C}_\Phi \circ \mathcal{R}_\Phi(g(\cdot, \bar{\alpha})) = \mathcal{C}_\Phi(\bar{\alpha})$. Since $\mathcal{R}_\Phi(\hat{h})$ is also feasible for problem (14), $\mathcal{R}_\Phi(\hat{h}) = \bar{\alpha}$ and therefore $\hat{h}(\cdot) = g(\cdot, \bar{\alpha})$. $\qquad\square$

## A.4 Theorem 2 for uncentered potential

In this subsection, we characterize the implicit bias of mirror flow with "uncentered" and unscaled potential functions that satisfying the following assumptions.

**Assumption 25.** *The potential function $\Phi\colon \mathbb{R}^{3n+1} \to \mathbb{R}$ satisfies: (i) $\Phi$ can be written in the form $\Phi(\theta) = \sum_{k=1}^{3n+1} \phi(\theta_k)$, where $\phi$ is a real-valued function on $\mathbb{R}$; (ii) $\Phi$ is twice continuously differentiable; (iii) $\nabla^2 \Phi(\theta)$ is positive definite for all $\theta \in \mathbb{R}^{3n+1}$.*

**Theorem 26 (Theorem 2 for uncentered potentials).** *Consider a two layer ReLU network (1) with $d$ input units and $n$ hidden units, where we assume $n$ is sufficiently large. Consider parameter initialization (2) and assume $\mathcal{B}$ is compactly supported on $[-B, B]$ for some $B > 0$. Let $p_{\mathcal{B}}(b)$ denote the density function for random variable $\mathcal{B}$. Consider any finite training data set $\{(x_i, y_i)\}_{i=1}^m$*

*that satisfies $x_i \neq x_j$ when $i \neq j$ and $\{\|x_i\|_2\}_{i=1}^m \subset \mathrm{supp}(p_\mathcal{B})$, and consider mirror flow (4) with a potential function $\Phi$ satisfying Assumption 25 and learning rate $\eta = \Theta(1/n)$. Then, there exist constants $C_1, C_2 > 0$ such that with high probability over the random parameter initialization, for any $t \geq 0$,*

$$\|\theta(t) - \hat{\theta}\|_\infty \leq C_1 n^{-1/2}, \quad \lim_{n \to \infty} \|H(t) - H(0)\|_2 = 0, \quad \|\boldsymbol{f}(t) - \boldsymbol{y}\|_2 \leq e^{-\eta_0 C_2 t} \|\boldsymbol{f}(0) - \boldsymbol{y}\|_2. \quad (70)$$

*Moreover, letting $\theta(\infty) = \lim_{t \to \infty} \theta(t)$, we have for any given $x \in \mathbb{R}^d$, that $\lim_{n \to \infty} |f(x, \theta(\infty)) - f(x, \theta_{\mathrm{GF}}(\infty))| = 0$, where $\theta_{\mathrm{GF}}(\infty)$ denotes the limiting point of gradient flow, i.e., mirror flow (4) with $\Phi(\theta) = \|\theta\|_2^2$, on the same training data and initial parameter.*

*Assuming univariate input data, i.e., $d = 1$, we have for any given $x \in \mathbb{R}$, that $\lim_{n \to \infty} |f(x, \theta(\infty)) - f(x, \hat{\theta}) - \bar{h}(x)| = 0$, where $\bar{h}(\cdot)$ is the solution to the following variational problem:*

$$\min_{h \in \mathcal{F}_{\mathrm{ReLU}}} \mathcal{G}_1(h) + \mathcal{G}_2(h) + \mathcal{G}_3(h) \quad \text{s.t. } h(x_i) = y_i - f(x_i, \hat{\theta}), \ \forall i \in [m],$$

$$where \quad \begin{cases} \mathcal{G}_1(h) = \displaystyle\int_{\mathrm{supp}(p_\mathcal{B})} \frac{(h''(x))^2}{p_\mathcal{B}(x)} \mathrm{d}x \\[2mm] \mathcal{G}_2(h) = (h'(+\infty) + h'(-\infty))^2 \\[2mm] \mathcal{G}_3(h) = \dfrac{1}{\mathbb{E}[\mathcal{B}^2]}(B(h'(+\infty) - h'(-\infty)) - (h(B) + h(-B)))^2. \end{cases} \quad (71)$$

*Here $\mathcal{F}_{\mathrm{ReLU}} = \{h(x) = \int \alpha(w, b)[wx - b]_+ \mathrm{d}\mu \colon \alpha \in C(\mathrm{supp}(\mu)), \alpha \text{ is uniformly continuous}\}$, $h'(+\infty) = \lim_{x \to +\infty} h'(x)$, and $h'(-\infty) = \lim_{x \to -\infty} h'(x)$.*

The proof of Theorem 26 mirrors the proof of Theorem 2, which we discussed in previous subsections. For brevity, below we only provide the proof sketch and highlight the differences brought by the change in assumptions.

First, we show Lemma 13 holds for uncentered potentials. Notice that

$$\begin{aligned} H_{ij}(0) &= \frac{1}{n} \sum_{k=1}^{3n+1} \nabla_{\theta_k} f(x_i; \hat{\theta}) \frac{1}{\phi''(\hat{\theta}_k)} \nabla_{\theta_k} f(x_j; \hat{\theta}) \\ &= \frac{1}{n} \sum_{k=1}^{n} \frac{1}{\phi''(\hat{a}_k)} \sigma(\hat{w}_k^T x_i - \hat{b}_k) \sigma(\hat{w}_k^T x_j - \hat{b}_k) + \frac{1}{n \phi''(\hat{d})} \\ &\quad + \frac{1}{n} \sum_{k=1}^{n} \hat{a}_k^2 \sigma'(\hat{w}_k^T x_i - \hat{b}_k) \sigma'(\hat{w}_k^T x_j - \hat{b}_k) \Big( \frac{1}{\phi''(\hat{b}_k)} + \sum_{r=1}^{d} \frac{x_{i,r} x_{j,r}}{\phi''(\hat{w}_{k,r})} \Big). \end{aligned}$$

By sub-Gaussian concentration inequality, we have $\max_{k \in [n]} |\hat{a}_k| = O_p(\sqrt{(\log n)/n})$. By noticing that $\max_{k \in [n], i \in [m]} \sigma(\hat{w}_k^T x_i - \hat{b}_k) = O(1)$ since $\mathcal{W}$ and $\mathcal{B}$ are bounded, we have

$$\frac{1}{n} \sum_{k=1}^{n} \Big( \frac{1}{\phi''(\hat{a}_k)} - \frac{1}{\phi''(0)} \Big) \sigma(\hat{w}_k^T x_i - \hat{b}_k) \sigma(\hat{w}_k^T x_j - \hat{b}_k) \xrightarrow{P} 0,$$

which implies $H_{ij}(0) \xrightarrow{P} H^\infty$. The remainder analysis proceeds as in the proof of Lemma 13.

Next, we show Proposition 14 and Proposition 16 hold. Note the key ingredients which are needed in proving these two results and which relate to the potentials are: (i) there exists a constant $\beta > 0$ such that with high probability, $\|Q(0)\|_2 \leq \beta$ and $\|Q(t)\|_2 \leq \beta$ if $\|\theta(t) - \theta(0)\|_\infty = o_p(1)$; and (ii) $\|Q(t) - Q(0)\|_2 = o_p(1)$ if $\|\theta(t) - \theta(0)\|_\infty = o_p(1)$, where we use the same notation as in Proposition 14. We verify these two properties for uncentered potentials. Since $\mathcal{B}$ is compactly supported, we have with high probability, $\max_{k \in [3n+1]} |\theta_k(0)| < C$ for some constant $C$ independent of $n$. Since $1/\phi''(\cdot)$ is uniformly continuous on $[-2C, 2C]$, we have

$$\|Q(t) - Q(0)\|_2 = \max_{k \in [3n+1]} \Big| \frac{1}{\phi''(\theta_k(t))} - \frac{1}{\phi''(\theta_k(0))} \Big| = o_p(1).$$

given $\|\theta(t) - \theta(0)\|_\infty = o_p(1)$. Meanwhile, notice that

$$\|Q(t)\|_2 = \max_{k \in [3n+1]} \frac{1}{\phi''(\theta_k(t))} \leq \max_{k \in [3n+1]} \frac{1}{\phi''(\theta_k(0))} + \left| \frac{1}{\phi''(\theta_k(t))} - \frac{1}{\phi''(\theta_k(0))} \right|.$$

By the continuity of $\phi''$ we have $\|Q(t)\|_2$ can be bounded by some constant $\beta > 0$ with high probability, given $\|\theta(t) - \theta(0)\|_\infty = o_p(1)$.

Then, we show Proposition 17 holds. Notice that for uncentered potentials,

$$D_\Phi(\boldsymbol{a}, \boldsymbol{a}(0)) = \sum_{k \in [n]} \Big( \phi(a_k) - \phi(a_k(0)) - \phi'(a_k(0))(a_k - a_k(0)) \Big)$$

$$= \sum_{k \in [n]} \Big( \frac{\phi''(a_k(0))}{2} + R(a_k - a_k(0)) \Big)(a_k - a_k(0))^2,$$

where $R$ captures the error produced by Taylor approximation and satisfies $\lim_{z \to 0} R(z) = 0$. Consider $L_n(\alpha) = \int v(w, b)\big(\alpha(w, b)\big)^2 \mathrm{d}\mu_n$ where $v(w_k(0), b_k(0)) = \frac{\phi''(a_k(0))}{2} + R(a_k - a_k(0))$ for $k \in [n]$. Hence, if $\bar{\boldsymbol{a}}$ solves (40), $\bar{\alpha}_n$, which satisfies $\bar{\alpha}_n(w_k(0), b_k(0)) = \bar{a}_k - a_k(0)$ for $k \in [n]$, solves the following problem:

$$\min_{\alpha \in \Theta_{\mathrm{ReLU}}} L_n(\alpha) \quad \text{s.t. } \langle \alpha, \sigma \rangle_n = y_i - f(x_i, \hat{\theta}).$$

Let $\bar{\alpha}'_n$ be the solution to the following problem:

$$\min_{\alpha \in \Theta_{\mathrm{ReLU}}} L'_n(\alpha) \quad \text{s.t. } \langle \alpha, \sigma_i \rangle_n = y_i - f(x_i, \hat{\theta}), \ i \in [m],$$

$$\text{where } L'_n(\alpha) \triangleq \int \frac{\phi''(0)}{2}\big(\alpha(w, b)\big)^2 \mathrm{d}\mu_n.$$

Notice that

$$|L_n(\alpha) - L'_n(\alpha)| = \left| \int \Big( v(w, b) - \frac{\phi''(0)}{2} \Big) \alpha^2(w, b) \mathrm{d}\mu_n \right|$$

$$\leq \max_{k \in [n]} \left| \frac{\phi''(a_k(0))}{2} - \frac{\phi''(0)}{2} + R(a_k - a_k(0)) \right| \cdot \|\alpha\|_n^2.$$

Following the same analysis as in the proof of Proposition 17, we have with high probability $\max_{k \in [n]} \bar{a}_k - a_k(0) = o_p(1)$, $\max_{k \in [n]} \bar{a}'_k - a_k(0) = o_p(1)$, $\|\bar{\alpha}_n\|_n^2 = O_p(1)$, and $\|\bar{\alpha}'_n\|_n^2 = O_p(1)$. Meanwhile, noticing that $\max_{k \in [n]} |a_k(0)| = O_p(\sqrt{(\log n)/n})$ holds with high probability and by the continuity of $\phi''$, we have $\max_{k \in [n]} |\phi''(a_k(0)) - \phi''(0)| = o_p(1)$. Therefore, we have

$$|L_n(\bar{\alpha}_n) - L'_n(\bar{\alpha}_n)|, \ |L_n(\bar{\alpha}'_n) - L'_n(\bar{\alpha}'_n)| = o_p(1).$$

The remainder analysis proceeds as in the proof of Proposition 17.

So far, we have shown that the training converges to zero training error, the lazy training occurs, and the implicit bias in the infinite width limit can be exactly described as in Proposition 17. Then Proposition 22 directly applies and gives the claimed results.

## B  PROOF OF THEOREM 9

In this section, we present the proof of Theorem 9. We follow the strategy that we introduced in Section 3.3 and break down the main Theorem into several intermediate results. Specifically, in Proposition 27 we show the mirror flow training converges to zero training error and is in the lazy regime. In Proposition 32, we characterize the implicit bias of mirror flow with scaled potentials in the parameter space. In Proposition 37, we obtain the function space description of the implicit bias.

We use $\mathrm{plim}_{n \to \infty} X_n = X$ or $X_n \xrightarrow{P} X$ to denote that random variables $X_n$ converge to $X$ in probability. We use the notations $O_p$ and $o_p$ to denote the standard mathematical orders in probability (see Appendix A for a detailed description).

### B.1 Linearization

We show that for mirror flow with scaled potentials, if the Hessian of the potential function is flat enough, the mirror flow converges to zero training error and is in the lazy regime (Proposition 27), and the network obtained by training all parameters can be well-approximated by that obtained by training only output weights (Proposition 29). Further, we show that if the potential is induced a univariate function of the form $\phi(z) = \frac{1}{\omega+1}(\psi(z) + \omega z^2)$ with convex function $\psi$ and positive constant $\omega$ as in Assumption 8, one can always choose sufficiently large $\omega$ to obtain the sufficient "flatness" (Proposition 30).

Note that under mirror flow with scaled potential, the evolution of the network output is governed by the following matrix:

$$H(t) = \frac{1}{n} J_\theta \boldsymbol{f}(t)(\nabla^2 \Phi(\theta))^{-1} J_\theta \boldsymbol{f}(t)^T,$$

where, by Assumption 8, the inverse Hessian matrix takes the form of

$$(\nabla^2 \Phi(\theta(t)))^{-1} = \mathrm{Diag}\left(\frac{1}{\phi''(n(\theta_k - \hat{\theta}))}\right).$$

It's clear that the behavior of the Hessian map is determined by the behavior of $\phi$. In the following analysis, we consider the following functions to measure the flatness of the Hessian map: for $C \geq 0$,

$$\kappa_1(C) = \inf_{z \in [-C,C]} (\phi''(z))^{-1}, \ \kappa_2(C) = \sup_{z \in [-C,C]} (\phi''(z))^{-1}, \ \kappa_3(C) = \sup_{z \in [-C,C]} |\phi'''(z)|.$$

Note function $\kappa_1(C)$ characterizes the smallest possible eigenvalue of $(\nabla^2 \Phi(\theta))^{-1}$ when $\theta$ satisfies $\|\theta - \hat{\theta}\|_\infty \leq Cn^{-1}$. Similarly, $\kappa_2$ characterizes the largest possible eigenvalue. Hence, $\kappa_1/\kappa_2$ captures the largest possible condition number of the inverse Hessian matrix. By definition, $\kappa_1/\kappa_2$ is monotonically decreasing. In the following Proposition, we show that if $\kappa_1/\kappa_2$ decays with a moderate rate such that it takes a larger value than $L/C$ for a particular constant $L$ determined by the training data and the network architecture at some point $C$, the mirror flow converges to zero training error and it is in the lazy training regime.

**Proposition 27.** *Consider a single-hidden-layer univariate network* (1) *with ReLU or absolute value activation function, $d$ input units and $n$ hidden units. Assume $n$ is sufficiently large. Consider parameter initialization* (2). *Assume $\mathcal{B}$ has a compact support. Given any finite training data set $\{(x_i, y_i)\}_{i=1}^m$ that satisfies $x_i \neq x_j$ when $i \neq j$ and $\{\|x_i\|_2\}_{i=1}^m \subset \mathrm{supp}(p_\mathcal{B})$, consider mirror flow* (4) *with learning rate $\eta = \eta_0/n$, $\eta_0 > 0$ and potential function $\Phi$. Assume $\Phi$ can be written in the form of $\Phi(\theta) = \frac{1}{n^2} \sum_{k=1}^{3n+1} \phi(n(\theta_k - \hat{\theta}_k))$, where $\phi$ is a real-valued $C^3$-smooth function on $\mathbb{R}$ and has strictly positive second derivatives everywhere. Assume there exists constant $C_1 > 0$ satisfying*

$$\frac{\kappa_1(C_1)}{\kappa_2(C_1)} \geq \frac{8\sqrt{m}(B_x + 1)K\|\boldsymbol{f}(0) - \boldsymbol{y}\|_2}{\lambda_0} \cdot \frac{1}{C_1}, \tag{72}$$

*where $\lambda_0 = \mathrm{plim}_{n \to \infty} \lambda_{\min}(\frac{1}{n} J_\theta \boldsymbol{f}(0) J_\theta \boldsymbol{f}(0)^T)$, $m$ is the size of the training data set, $B_x = \max_{i \in [m]} \|x_i\|_2$, and $K$ is a constant determined by $\mathcal{B}$ and $\mathcal{A}$. Then with high probability over random initialization,*

$$\sup_{t \geq 0} \|\theta(t) - \hat{\theta}\|_\infty \leq C_1 n^{-1}, \ \inf_{t \geq 0} \lambda_{\min}(H(t)) \geq \frac{\kappa_1(C_1)\lambda_0}{4}.$$

*Moreover,*

$$\|\boldsymbol{f}(t) - \boldsymbol{y}\|_2 \leq \exp\left(-\frac{\kappa_1(C_1)\lambda_0}{4}\eta_0 t\right)\|\boldsymbol{f}(0) - \boldsymbol{y}\|_2, \ \forall t \geq 0.$$

**Proof.** [Proof of Proposition 27] To ease the notation, let $J(t) = J_\theta \boldsymbol{f}(t)$ and $Q(t) = (\nabla^2 \Phi(\theta(t)))^{-1}$. For $i \in [m]$ and $k \in [n]$, let $\sigma_{ik}(t) = \sigma(w_k(t)^T x_i - b_k(t))$ and $\sigma'_{ik}(t) = \sigma'(w_k(t)^T x_i - b_k(t))$.

Assume $C_1$ satisfies (72). Consider the map $\tau \colon [0, \infty) \to \mathbb{R}, \tau(t) = n\|\theta(t) - \hat{\theta}\|_\infty$. Consider $t_1 = \inf\{t \geq 0 \colon \tau(t) > C_1\}$. Clearly, $t_1 > 0$. Now we aim to show $t_1 = +\infty$. Assume $t_1 < \infty$ for the sake of contradiction. Since $\tau(\cdot)$ is continuous, $\tau(t_1) = C_1$ and $\tau(t) \leq C_1$ for $t \in [0, t_1]$.

In the following, let $t$ be any point in $[0, t_1]$. Since $n\|\theta(t) - \hat{\theta}\|_\infty \leq C_1$ we have

$$\lambda_{\min}(Q(t)) = \min_k \left\{ \frac{1}{\phi''(n(\theta(t) - \hat{\theta}))} \right\} \geq \kappa_1(C_1) > 0.$$

By Lemma 13, with sufficiently large $n$ and with high probability, $\lambda_{\min}(n^{-1}J(0)J(0)^T) > \frac{\lambda_0}{2}$. For $i, j \in [m]$ we have

$$|(n^{-1}J(t)J(t)^T)_{ij} - (n^{-1}J(0)J(0)^T)_{ij}| \leq I_1 + I_2 + I_3, \tag{73}$$

where

$$\begin{cases} I_1 = \dfrac{1}{n} \sum_{k=1}^{n} |\sigma_{ik}(t)\sigma_{jk}(t) - \sigma_{ik}(0)\sigma_{jk}(0)|; \\[3mm] I_2 = |x_i^T x_j| \dfrac{1}{n} \sum_{k=1}^{n} |a_k(t)^2 \sigma'_{ik}(t)\sigma'_{jk}(t) - a_k(0)^2 \sigma'_{ik}(0)\sigma'_{jk}(0)|; \\[3mm] I_3 = \dfrac{1}{n} \sum_{k=1}^{n} \left| a_k(t)^2 \sigma'_{ik}(t)\sigma'_{jk}(t) - a_k(0)^2 \sigma'_{ik}(0)\sigma'_{jk}(0) \right|. \end{cases} \tag{74}$$

Since $\mathcal{A}$ is sub-Gaussian random variable, and $\mathcal{B}$ are bounded random variable, there exists $K > 1$ such that with high probability,

$$\|\boldsymbol{b}(0)\|_\infty \leq K, \quad \|\boldsymbol{a}(0)\|_\infty \leq \|\boldsymbol{a}(0)\|_2 \leq K. \tag{75}$$

Then we have for any $i, j \in [m], k \in [n]$,

$$|\sigma_{ik}(t)| \leq |w_k(t)^T x_i| + |b_k(t)| \leq (B_x + 1)2K,$$

and

$$|\sigma_{ik}(t) - \sigma_{jk}(t)| \leq \|w_k(t) - w_k(0)\|_2 \|x_i\|_2 + |b_k(t) - b_k(0)| \leq \sqrt{d}(B_x + 1)C_1 n^{-1}.$$

Therefore, we have

$$\begin{aligned} I_1 &\leq \frac{1}{n} \sum_{k=1}^{n} \left( |\sigma_{ik}(t) - \sigma_{ik}(0)||\sigma_{jk}(t)| + |\sigma_{ik}(0)||\sigma_{jk}(t) - \sigma_{jk}(0)| \right) \\ &\leq \frac{1}{n} \sum_{k \in [n]} O_p(1) o_p(1) + O_p(1) o_p(1) \\ &= o_p(1). \end{aligned}$$

Noticing $\|\boldsymbol{a}(0)\|_2 = O_p(1)$, for $I_2$ we have

$$\begin{aligned} I_2 &\leq \frac{B_x^2}{n} \sum_{k \in [n]} \left( a_k(0)^2 |\sigma'_{ik}(t)\sigma'_{jk}(t) - \sigma'_{ik}(0)\sigma'_{ik}(0)| + |a_k(t)^2 - a_k(0)^2| \right) \\ &= \frac{B_x^2}{n} \left( 2\|\boldsymbol{a}(0)\|_2^2 + \sum_{k \in [n]} o_p(1) \right) \\ &= o_p(1). \end{aligned}$$

Using a technique similar to that applied for $I_2$ above, we have $I_3 = o_p(1)$. By applying a union bound we have $\|n^{-1}J(t)J(t)^T - n^{-1}J(0)J(0)^T\|_2 = o_p(1)$. Then for large enough $n$, we have $\lambda_{\min}(n^{-1}J(t)J(t)^T) \geq \lambda_0/4$. Therefore, we have

$$\lambda_{\min}(H(t)) \geq \lambda_{\min}(n^{-1}J(t)J(t)^T)\lambda_{\min}(Q(t)) > \frac{\kappa_1(C_1)\lambda_0}{4}. \tag{76}$$

Then we have

$$\frac{\mathrm{d}}{\mathrm{d}t} \|\boldsymbol{f}(t) - \boldsymbol{y}\|_2^2 \leq -2\eta_0 \frac{\kappa_1(C_1)\lambda_0}{4} \|\boldsymbol{f}(t) - \boldsymbol{y}\|_2^2$$

and

$$\|\boldsymbol{f}(t) - \boldsymbol{y}\|_2 \leq \exp(-\frac{\kappa_1(C_1)\lambda_0}{4}\eta_0 t)\|\boldsymbol{f}(0) - \boldsymbol{y}\|_2. \tag{77}$$

Notice that,

$$\|J_{a_k}\boldsymbol{f}(t)\|_2 \leq \sqrt{m}\|J_{a_k}\boldsymbol{f}\|_\infty \leq \sqrt{m}(B_x\|w_k(t)\|_2 + |b_k(t)|) \leq 2\sqrt{m}K(B_x+1). \quad (78)$$

Then with large enough $n$ and for all $k \in [n]$, we have

$$\begin{aligned}
|\frac{\mathrm{d}}{\mathrm{dt}}a_k(t)| &\leq \frac{\eta_0}{n}\|Q(t)\|_2\|J_{a_k}\boldsymbol{f}(t)\|_2\|\boldsymbol{f}(t) - \boldsymbol{y}\|_2 \\
&\leq \frac{\eta_0}{n}\cdot\kappa_2(C_1)\cdot 2\sqrt{m}K(B_x+1)\cdot\exp(-\frac{\kappa_1(C_1)\lambda_0}{4}\eta_0 t)\|\boldsymbol{f}(0) - \boldsymbol{y}\|_2.
\end{aligned} \quad (79)$$

and hence

$$\begin{aligned}
|a_k(t) - a_k(0)| &\leq \int_0^t |\frac{\mathrm{d}}{\mathrm{dt}}a_k(s)|\mathrm{d}s \\
&< \frac{8\kappa_2(C_1)\sqrt{m}K(B_x+1)\|\boldsymbol{f}(0) - \boldsymbol{y}\|_2}{\kappa_1(C_1)\lambda_0}n^{-1} \\
&\leq C_1 n^{-1}.
\end{aligned} \quad (80)$$

Similarly, by noticing that with large enough $n$,

$$\begin{cases}
\|J_{w_{k,r}}\boldsymbol{f}(t)\|_2 \leq \sqrt{m}B_x|a_k(t)| \leq 2\sqrt{m}B_x K, \quad \forall k \in [n], r \in [d]; \\
\|J_{b_k}\boldsymbol{f}(t)\|_2 \leq \sqrt{m}|a_k(t)| \leq 2\sqrt{m}K, \quad \forall k \in [n]; \\
\|J_d\boldsymbol{f}(t)\|_2 = \sqrt{m}.
\end{cases} \quad (81)$$

Comparing (81) to (78) and using a similar computations as in (79) and (80), we conclude that $\tau(t_1) < C_1$, which yields a contradiction. It then follows that (76) and (77) hold for $t \geq 0$. This concludes the proof. $\qquad\square$

We point out that Proposition 27 can be extended to the case where only output weights are trained, without requiring any additional efforts. We present the results in the following proposition.

**Proposition 28.** *Assume the same assumptions in Proposition 27. Assume constant $C_1 > 0$ satisfies (72). Then with high probability over random initialization, for any $t \in [0, \infty)$,*

$$\|\tilde{\theta}(t) - \hat{\theta}\|_\infty \leq C_1 n^{-1}, \quad \|\tilde{\boldsymbol{f}}(t) - \boldsymbol{y}\|_2 \leq \exp(-\frac{\kappa_1(C_1)\lambda_0}{4}\eta_0 t)\|\boldsymbol{f}(0) - \boldsymbol{y}\|_2.$$

In the following result we show that training only the output weights well approximates training all parameters.

**Proposition 29.** *Assume the same assumptions as in Proposition 27. Assume there exists a constant $C_1$ which satisfies (72) and*

$$\kappa_2(C_1)\kappa_3(C_1)\Big(\lambda_0\frac{\kappa_2(C_1)}{\kappa_1(C_1)} + 10m(B_x+1)^2K^2\Big(\frac{\kappa_2(C_1)}{\kappa_1(C_1)}\Big)^2\Big) \leq \frac{(\lambda_0)^2}{16(B_x+1)K\sqrt{m}\|\boldsymbol{f}(0) - \boldsymbol{y}\|_2}. \quad (82)$$

*Then for any $x \in \mathbb{R}^d$ and with high probability over initialization,*

$$\lim_{n\to\infty}\sup_{t\geq 0}\|f(x, \theta(t)) - f(x, \tilde{\theta}(t))\|_2 = 0.$$

**Proof.** [Proof of Proposition 29] We define the following function defined for $t \geq 0$:

$$\iota(t) = n^{3/2}\|\boldsymbol{a}(t) - \tilde{\boldsymbol{a}}(t)\|_\infty.$$

For any fixed $C > 0$, consider $t_1 = \inf\{t \geq 0: \iota(t) > C\}$. Now we aim to prove $t_1 = +\infty$. For the sake of contradiction, assume $t_1 < +\infty$. Then we have for $t \in [0, t_1]$, $\iota(t) \leq C$ and in particular, $\iota(t_1) = C$.

Assume that $C_1 > 0$ satisfies (72) and (82). According to Proposition 27 and Proposition 28, we have $\sup_{t\geq 0}\|n(\theta(t) - \hat{\theta})\|_\infty \leq C_1$ and $\sup_{t\geq 0}\|n(\tilde{\theta}(t) - \hat{\theta})\|_\infty \leq C_1$. By assumption, $(\phi''(x))^{-1}$ is Lipschitz continuous on $[-C_1, C_1]$. Let $L$ denote the corresponding Lipschitz constant. Note that

$$L \leq \sup_{x\in[C_1,C_1]}|(\frac{1}{\phi''(x)})'| = \sup_{x\in[C_1,C_1]}\frac{|\phi'''(x)|}{(\phi''(x))^2} \leq \kappa_2(C_1)^2\kappa_3(C_1).$$

Next we bound the difference between $H(t)$ and $\tilde{H}(t)$. To simplify the notation, we let $\sigma_{ik}(t) = \sigma(w_k(t)^T x_i - b_k(t))$ for $i \in [m]$ and $k \in [n]$. For $i, j \in [m]$ we have the following decomposition:

$$|H_{ij}(t) - \tilde{H}_{ij}(t)| \le I_1 + I_2 + I_3, \tag{83}$$

where

$$
\begin{cases}
I_1 = \dfrac{1}{n} \displaystyle\sum_{k=1}^n \left| \dfrac{\sigma_{ik}(t)\sigma_{jk}(t)}{\phi''(n(a_k(t) - a_k(0)))} - \dfrac{\sigma_{ik}(0)\sigma_{jk}(0)}{\phi''(n(\tilde{a}_k(t) - a_k(0)))} \right|; \\[3mm]
I_2 = \dfrac{1}{n} \displaystyle\sum_{k=1}^n \sum_{r=1}^d \left( |x_{i,r} x_{j,r}| \left| \dfrac{a_k(t)^2 \sigma'_{ik}(t) \sigma'_{jk}(t)}{\phi''(n(w_{k,r}(t) - w_{k,r}(0)))} \right| \right) + \left| \dfrac{a_k(t)^2 \sigma'_{ik}(t) \sigma'_{jk}(t)}{\phi''(n(b_k(t) - b_k(0)))} \right| \\[3mm]
I_3 = \dfrac{1}{n} \left| \dfrac{1}{\phi''(n(d(t) - d(0)))} \right|.
\end{cases}
\tag{84}
$$

Since $\mathcal{A}$ is sub-Gaussian and $\mathcal{B}$ is bounded, there exists $K > 1$ such that with high probability,

$$\|\boldsymbol{b}(0)\|_\infty \le K, \quad \|\boldsymbol{a}(0)\|_\infty \le \|\boldsymbol{a}(0)\|_2 \le K. \tag{85}$$

Noticing $\|\theta(t) - \hat{\theta}\|_\infty < C_1 n^{-1}$, we have for large enough $n$ and for any $i, j \in [m], k \in [n], t \ge 0$,

$$|\sigma_{ik}(t)| \le |w_k(t)^T x_i| + |b_k(t)| \le (B_x + 1)2K$$

and

$$|\sigma_{ik}(t) - \sigma_{jk}(t)| \le \|w_k(t) - w_k(0)\|_2 \|x_i\|_2 + |b_k(t) - b_k(0)| \le \sqrt{d}(B_x + 1)C_1 n^{-1}.$$

Recall $\kappa_2(C_1) = \sup_{z \in [-C_1, C_1]} (\phi''(z))^{-1}$. By noticing $\|\boldsymbol{a}(t) - \tilde{\boldsymbol{a}}(t)\|_\infty < Cn^{-3/2}$, we have for $t \in [0, t_1]$,

$$
\begin{aligned}
I_1 &\le \frac{1}{n} \sum_{k=1}^n \Big( |\sigma_{ik}(t) - \sigma_{ik}(0)||\sigma_{jk}(t)|\kappa_2(C_1) + |\sigma_{ik}(0)||\sigma_{jk}(t) - \sigma_{jk}(0)|\kappa_2(C_1) + \\
&\qquad |\sigma_{ik}(0)||\sigma_{jk}(0)| \big( \frac{1}{\phi''(na_k(t))} - \frac{1}{\phi''(n\tilde{a}_k(t))} \big) \Big) \\
&\le \frac{1}{n} \sum_{k=1}^n \Big( O_p(n^{-1})\kappa_2(C_1) + O_p(n^{-1})\kappa_2(C_1) + 4K^2(B_x + 1)^2 LCn^{-1/2} \Big) \\
&= 4K^2(B_x + 1)^2 LCn^{-1/2} + O_p(n^{-1})\kappa_2(C_1).
\end{aligned}
$$

In the meantime, note that $\|\boldsymbol{a}(t) - \boldsymbol{a}(0)\|_2 \le \sqrt{n}\|\boldsymbol{a}(t) - \boldsymbol{a}(0)\|_\infty = O(n^{-1/2})$. Therefore, $\|\boldsymbol{a}(t)\|_2 \le \|\boldsymbol{a}(0)\|_2 + \|\boldsymbol{a}(t) - \boldsymbol{a}(0)\|_2 = O_p(1)$. Then we have that for $t \in [0, t_1]$,

$$I_2 \le \frac{d}{n}(B_x^2 + 1)\|\boldsymbol{a}(t)\|_2^2 \kappa_2(C_1) = O_p(n^{-1})\kappa_2(C_1), \quad I_3 \le O_p(n^{-1})\kappa_2(C_1).$$

Then with large enough $n$ we have for all $i, j \in [m]$ that

$$\sup_{t \in [0, t_1]} |H_{ij}(t) - \tilde{H}_{ij}(t)| \le 5K^2(B_x + 1)^2 LCn^{-1/2}$$

and hence

$$\sup_{t \in [0, t_1]} \|H(t) - \tilde{H}(t)\|_2 \le 5mK^2(B_x + 1)^2 LCn^{-1/2}. \tag{86}$$

Recall $\kappa_1(C_1) = \inf_{z \in [-C_1, C_1]} (\phi''(z))^{-1}$. Let $\boldsymbol{r}(t) = \boldsymbol{f}(t) - \tilde{\boldsymbol{f}}(t)$ and $C_2 = (\kappa_1(C_1)\lambda_0)/4$. Consider the function $u(t) = \|\boldsymbol{r}(t)\|_2$ defined on $t \in [0, t_1]$. Clearly, $u(t)$ is continuous on $[0, t_1]$. Since $u(t) = 0$ if and only if $\boldsymbol{r}(t) = 0$, we have that $u(t)$ is differentiable whenever $u(t) \ne 0$. In

particular, when $u(t) \neq 0$, notice that

$$
\begin{aligned}
\frac{\mathrm{d}}{\mathrm{dt}}(e^{C_2\eta_0 t}u(t))^2 &= \frac{\mathrm{d}}{\mathrm{dt}}\|e^{C_2\eta_0 t}\boldsymbol{r}(t)\|_2^2 \\
&= 2e^{C_2\eta_0 t}\boldsymbol{r}(t)^T \frac{\mathrm{d}}{\mathrm{dt}}(e^{C_2\eta_0 t}\boldsymbol{r}(t)) \\
&= 2e^{2C_2\eta_0 t}\boldsymbol{r}(t)^T\Big(C_2\eta_0\boldsymbol{r}(t) - \eta_0 H(t)(\boldsymbol{f}(t) - \boldsymbol{y}) + \eta_0\tilde{H}(t)(\tilde{\boldsymbol{f}}(t) - \boldsymbol{y})\Big) \\
&= 2\eta_0 e^{2C_2\eta_0 t}\Big(\boldsymbol{r}(t)^T(\tilde{H}(t) - H(t))(\tilde{\boldsymbol{f}}(t) - \boldsymbol{y}) - \boldsymbol{r}(t)^T(H(t) - C_2 I_n)\boldsymbol{r}(t)\Big) \\
&\leq 2\eta_0 e^{2C_2\eta_0 t}\boldsymbol{r}(t)^T(\tilde{H}(t) - H(t))(\tilde{\boldsymbol{f}}(t) - \boldsymbol{y}) \\
&\leq 2\eta_0 e^{2C_2\eta_0 t}u(t)\|\tilde{H}(t) - H(t)\|_2\|\tilde{\boldsymbol{f}}(t) - \boldsymbol{y}\|_2,
\end{aligned}
\tag{87}
$$

where in the first inequality we use the property $\lambda_{\min}(H(t)) > C_2$ for any $t \geq 0$ from Proposition 27. Meanwhile, we have

$$
\frac{\mathrm{d}}{\mathrm{dt}}(e^{C_2\eta_0 t}u(t))^2 = 2e^{C_2\eta_0 t}u(t)\frac{\mathrm{d}}{\mathrm{dt}}(e^{C_2\eta_0 t}u(t))
$$

and hence by (87),

$$
\frac{\mathrm{d}}{\mathrm{dt}}(e^{C_2\eta_0 t}u(t)) \leq \eta_0 e^{C_2\eta_0 t}\|\tilde{H}(t) - H(t)\|_2\|\tilde{\boldsymbol{f}}(t) - \boldsymbol{y}\|_2.
$$

Now consider $t \in [0, t_1]$. Let $t' = \inf\{s \in [0, t]: \boldsymbol{r}(s) \neq 0\}$. Then we have

$$
e^{C_2\eta_0 t}u(t) \leq \int_{t'}^t \eta_0 e^{C_2\eta_0 s}\|\tilde{H}(s) - H(s)\|_2\|\tilde{\boldsymbol{f}}(s) - \boldsymbol{y}\|_2 \mathrm{d}s.
$$

Hence for all $t \in [0, t_1]$, we have

$$
\begin{aligned}
\|\boldsymbol{r}(t)\|_2 &\leq e^{-C_2\eta_0 t}\int_{t'}^t \eta_0 e^{C_2\eta_0 s}\|\tilde{H}(s) - H(s)\|_2\|\tilde{\boldsymbol{f}}(s) - \boldsymbol{y}\|_2 \mathrm{d}s \\
&\leq \eta_0 e^{-C_2\eta_0 t}(5mK^2(B_x + 1)^2 LCn^{-1/2})\int_{t'}^t \|\boldsymbol{f}(0) - \boldsymbol{y}\|_2 \mathrm{d}s \\
&\leq \eta_0 e^{-C_2\eta_0 t}5mK^2(B_x + 1)^2 LC\|\boldsymbol{f}(0) - \boldsymbol{y}\|_2 tn^{-1/2}.
\end{aligned}
\tag{88}
$$

We have that for all $k \in [n]$ and with sufficiently large $n$,

$$
\begin{aligned}
&|\frac{\mathrm{d}}{\mathrm{dt}}(a_k(t) - \tilde{a}_k(t))| \\
&= \frac{\eta_0}{n}\Big|\frac{1}{\phi''(na_k(t))}J_{a_k}\boldsymbol{f}(t)^T(\boldsymbol{f}(t) - \boldsymbol{y}) - \frac{1}{\phi''(n\tilde{a}_k(t))}J_{a_k}\boldsymbol{f}(0)^T(\tilde{\boldsymbol{f}}(t) - \boldsymbol{y})\Big| \\
&\leq \frac{\eta_0}{n}\Big(\Big|\frac{1}{\phi''(na_k(t))} - \frac{1}{\phi''(n\tilde{a}_k(t))}\Big|\|J_{a_k}\boldsymbol{f}(t)\|_2\|\boldsymbol{f}(t) - \boldsymbol{y}\|_2 \\
&\quad + \Big|\frac{1}{\phi''(n\tilde{a}_k(t))}\Big|\|J_{a_k}\boldsymbol{f}(t) - J_{a_k}\boldsymbol{f}(0)\|_2\|\boldsymbol{f}(t) - \boldsymbol{y}\|_2 \\
&\quad + \Big|\frac{1}{\phi''(n\tilde{a}_k(t))}\Big|\|J_{a_k}\boldsymbol{f}(0)\|_F\|\boldsymbol{r}(t)\|_2\Big)
\end{aligned}
$$

Noticing $\|\theta(t) - \hat{\theta}\|_\infty \le C_1 n^{-1}$ and using (88), we have that

$$
\begin{aligned}
&|\frac{\mathrm{d}}{\mathrm{d}t}(a_k(t) - \tilde{a}_k(t))| \\
\le& \frac{\eta_0}{n}\Big( LCn^{-1/2} \cdot \sqrt{m}(B_x\|w_k(t)\|_2 + |b_k(t)|) \cdot \|\boldsymbol{f}(0) - \boldsymbol{y}\|_2 e^{-C_2\eta_0 t} \\
&+ \kappa_2(C_1) \cdot \sqrt{m}(B_x\|w_k(t) - w_k(0)\|_2 + |b_k(t) - b_k(0)|) \cdot \|\boldsymbol{f}(0) - \boldsymbol{y}\|_2 e^{-C_2\eta_0 t}) \\
&+ \kappa_2(C_1) \cdot \sqrt{m}(B_x\|w_k(0)\|_2 + |b_k(0)|) \cdot \eta_0 e^{-C_2\eta_0 t} 5mK^2(B_x + 1)^2 LC\|\boldsymbol{f}(0) - \boldsymbol{y}\|_2 tn^{-1/2}\Big) \\
\le& \frac{2\eta_0\sqrt{m}\|\boldsymbol{f}(0) - \boldsymbol{y}\|_2}{\exp(C_2\eta_0 t)}\Big( LC \cdot K(B_x + 1)n^{-3/2} + O_p(n^{-2})\kappa_2(C_1)C_1 \\
&+ 5\kappa_2(C_1)(B_x + 1)^3 K^3\eta_0 mLCtn^{-3/2}\Big) \\
\le& \frac{2LC(B_x + 1)K\eta_0\sqrt{m}\|\boldsymbol{f}(0) - \boldsymbol{y}\|_2}{\exp(C_2\eta_0 t)}\big(2 + 5\kappa_2(C_1)(B_x + 1)^2 K^2\eta_0 mt\big)n^{-3/2}
\end{aligned}
$$

Recall $L \le \kappa_2(C_1)^2\kappa_3(C_1)$ and $C_2 = (\kappa_1(C_1)\lambda_0)/4$. Then $t \in [0, t_1]$ and $k \in [n]$ we have

$$
\begin{aligned}
&|a_k(t) - \tilde{a}_k(t)| \\
<& 2LC(B_x + 1)K\eta_0\sqrt{m}\|\boldsymbol{f}(0) - \boldsymbol{y}\|_2\Big(\int_0^\infty \frac{2 + 5\kappa_2(C_1)(B_x + 1)^2 K^2\eta_0 ms}{\exp(C_2\eta_0 s)}\mathrm{d}s\Big)n^{-3/2} \\
=& 2LC(B_x + 1)K\sqrt{m}\|\boldsymbol{f}(0) - \boldsymbol{y}\|_2\frac{8\kappa_1(C_1)\lambda_0 + 80\kappa_2(C_1)m(B_x + 1)^2 K^2}{(\kappa_1(C_1)\lambda_0)^2}n^{-3/2} \\
\le& \kappa_2(C_1)^2\kappa_3(C_1)\frac{\kappa_1(C_1)\lambda_0 + 10m(B_x + 1)^2 K^2\kappa_2(C_1)}{\kappa_1(C_1)^2} \cdot \frac{16(B_x + 1)K\sqrt{m}\|\boldsymbol{f}(0) - \boldsymbol{y}\|_2}{(\lambda_0)^2}Cn^{-3/2} \\
\le& Cn^{-3/2},
\end{aligned}
$$

where the last inequality comes from the fact that $C_1$ satisfies (82). Hence, we have $\iota(t) < C$ for $t \in [0, t_1]$. Note this holds at $t = t_1$, which yields a contradiction. Hence, $t_1 = +\infty$. By (88) we have that all $t \ge 0$,

$$
\|\boldsymbol{f}(t) - \tilde{\boldsymbol{f}}(t)\|_2 = o_p(1)e^{-C_2\eta_0 t}t. \tag{89}
$$

Next we fix the constants $C_1$ and $C$ and aim to show $f(\cdot, \theta(t))$ agrees with $f(\cdot, \tilde{\theta}(t))$ on any given test data $x \in \mathbb{R}^d$. For a fix $x \in \mathbb{R}^d$, consider

$$
\begin{aligned}
\frac{\mathrm{d}}{\mathrm{d}t}f(x, \theta(t)) &= -\eta_0 H(t, x)(\boldsymbol{f}(t) - \boldsymbol{y}) \\
H(t, x) &\triangleq \nabla_\theta f(x, \theta(t))(\nabla^2\Phi(\theta(t)))^{-1}J_\theta\boldsymbol{f}(t)^T
\end{aligned}
$$

and

$$
\begin{aligned}
\frac{\mathrm{d}}{\mathrm{d}t}f(x, \tilde{\theta}(t)) &= -\eta_0\tilde{H}(t, x)(\tilde{\boldsymbol{f}}(t) - \boldsymbol{y}) \\
\tilde{H}(t, x) &= \nabla_{\boldsymbol{a}}f(x, \hat{\theta})(\nabla^2\Phi(\tilde{\boldsymbol{a}}(t)))^{-1}J_{\boldsymbol{a}}\boldsymbol{f}(0)^T
\end{aligned}
$$

Using the fact that $\|\boldsymbol{a}(t) - \tilde{\boldsymbol{a}}(t)\| \le Cn^{-3/2}$ and applying a decomposition similar to (83) and (84), we have

$$
\sup_{t \ge 0}\|H(t, x) - \tilde{H}(t, x)\|_2 = O_p(n^{-1/2}). \tag{90}
$$

For large enough $n$, by (85) and $\|\theta(t) - \hat{\theta}\|_\infty \le C_1 n^{-1}$, we have

$$
\begin{aligned}
\|J_\theta\boldsymbol{f}(t)\|_F^2 &= \sum_{i=1}^m\Big(1 + \sum_{k=1}^n|\sigma_{ik}(t)|^2 + \sum_{k=1}^n|a_k(t)|^2|\sigma'_{ik}(t)|^2(\|x_i\|_2^2 + 1)\Big) \\
&\le m + \sum_{i=1}^m\sum_{k=1}^n\Big(2\|w_k(t)\|_2^2\|x_i\|^2 + 2|b_k(t)|^2 + |a_k(t)|^2(\|x_i\|^2 + 1)\Big) \\
&= O_p(n),
\end{aligned} \tag{91}
$$

and hence $\|J_\theta \boldsymbol{f}(t)\|_2 = O_p(\sqrt{n})$. Note a similar bound applies to $\|J_\theta f(x, \theta(t))\|_2$. Then with large enough $n$ we have for $t \geq 0$,

$$\|H(t, x)\|_2 \leq \frac{1}{n}\|J_\theta f(x, \theta(t))\|_2\|Q(t)\|_2\|J_\theta \boldsymbol{f}(t)\|_2 = O_p(1).$$

Finally, using (89) and (90), we have for $t \geq 0$,

$$
\begin{aligned}
&|f(x, \theta(t)) - f(x, \tilde{\theta}(t))| \\
&\leq \int_0^t \left|\frac{\mathrm{d}}{\mathrm{d}s}(f(x, \theta(s)) - f(x, \theta(s)))\right|\mathrm{d}s \\
&\leq \int_0^t \left|\eta_0 H(s, x)(\boldsymbol{f}(s) - \boldsymbol{y}) - \eta_0 \tilde{H}(s, x)(\tilde{\boldsymbol{f}}(s) - \boldsymbol{y})\right|\mathrm{d}s \\
&\leq \int_0^t \eta_0\|H(s, x)\|_2\|\boldsymbol{f}(s) - \tilde{\boldsymbol{f}}(s)\|_2 \\
&\quad + \eta_0\|H(s, x) - \tilde{H}(s, x)\|_2\|\tilde{\boldsymbol{f}}(s) - \boldsymbol{y}\|_2\mathrm{d}s \\
&\leq \int_0^\infty O_p(1)o_p(1)e^{-C_2\eta_0 s}s\mathrm{d}s + O_p(n^{-1/2})\int_0^\infty \|\boldsymbol{f}(0) - \boldsymbol{y}\|_2^2 e^{-C_2\eta_0 s}\mathrm{d}s \\
&= o_p(1),
\end{aligned}
$$

which gives the claimed results. $\qquad\square$

Next we show that, if the potential $\Phi$ satisfies Assumption 8, one can select a sufficiently large $\omega$ to ensure that the conditions in Proposition 27 and 29 are satisfied.

**Proposition 30.** *Consider a potential function that satisfies the assumptions in Proposition 27. Further, assume that $\phi(x) = \frac{1}{\omega+1}(\psi(x) + \omega x^2)$ with convex function $\psi \in C^3(\mathbb{R})$ and positive constant $\omega > 0$. Let $D_1, D_2$ be constants defined as follows:*

$$D_1 = \frac{8\sqrt{m}(B_x + 1)K\|\boldsymbol{f}(0) - \boldsymbol{y}\|_2}{\lambda_0}, \quad D_2 = 10m(B_x + 1)^2 K^2.$$

*Consider functions $\zeta_1$ and $\zeta_2$ defined as follows:*

$$
\begin{cases}
\zeta_1(C) = \dfrac{\sup_{z\in[-C,C]}\psi''(z) \cdot D_1}{2(C - D_1)}, & \forall C > D_1; \\
\zeta_2(C) = \dfrac{\sup_{|z|<C}|\psi'''(z)| \cdot D_1}{\lambda_0}\left(\lambda_0\left(1 + \sup_{|z|<C}\psi''(z)\right) + D_2\left(1 + \sup_{|z|<C}\psi''(z)\right)^2\right), & \forall C > 0.
\end{cases}
$$
(92)

*Then there exists $C_1 > 0$ that satisfies (72) and (82) whenever $\omega > \omega_0$ with $\omega_0$ defined as follows:*

$$\omega_0 = \max\{\frac{1}{2}, 1 + \inf_{C'>D_1}\zeta_1(C'), \zeta_2(C^*)\}, \quad C^* = \sup\{C > D_1 : \zeta_1(C) < 1 + \inf_{C'>D_1}\zeta_1(C')\}.$$

**Proof.** [Proof of Proposition 30] Since $\psi'' \geq 0$, we have that for $z \in [-C, C]$,

$$\frac{\omega+1}{\sup_{z\in[-C,C]}\psi''(z) + 2\omega} \leq \frac{1}{\phi''(z)} = \frac{\omega+1}{\psi''(z) + 2\omega} \leq \frac{\omega+1}{2\omega}.$$

Hence we can take

$$\kappa_1(C) = \frac{\omega+1}{\sup_{z\in[-C,C]}\psi''(z) + 2\omega}, \quad \kappa_2(C) \equiv \frac{\omega+1}{2\omega}.$$

Then we have

$$\frac{\kappa_1(C)}{\kappa_2(C)} = \frac{\omega+1}{\sup_{z\in[-C,C]}\psi''(z) + 2\omega}\bigg/\frac{\omega+1}{2\omega} = \frac{2\omega}{\sup_{z\in[-C,C]}\psi''(z) + 2\omega}.$$

Then (72) can be reformulated as

$$\frac{2\omega}{\sup_{z\in[-C,C]} \psi''(z) + 2\omega} \cdot C \geq D_1. \tag{93}$$

When $C > D_1$, we can rearrange (93) as follows

$$\omega \geq \frac{\sup_{z\in[-C,C]} \psi''(z) \cdot D_1}{2(C - D_1)} \triangleq \zeta_1(C) \tag{94}$$

Note $\zeta_1(C)$ is a continuous function on $C > D_1$. This implies whenever $\omega$ satisfies

$$\omega > \inf_{C'>D_1} \zeta(C') \tag{95}$$

there exists $C > D_1$ such that (94) holds and thus (72) holds.

On the other side, notice that

$$\kappa_3(C) = \sup_{z\in[-C,C]} |\phi'''(z)| = \frac{\sup_{z\in[-C,C]} |\psi'''(z)|}{\omega + 1}.$$

Then (82) can be reformulated as

$$\frac{\sup_{z\in[-C,C]} |\psi'''(z)|}{2\omega} \left(\lambda_0 \left(1 + \frac{\sup_{z\in[-C,C]} \psi''(z)}{2\omega}\right) + D_2 \left(1 + \frac{\sup_{z\in[-C,C]} \psi''(z)}{2\omega}\right)^2\right) \leq \frac{\lambda_0}{2D_1}. \tag{96}$$

If $\omega > 1/2$, we have

$$1 + \frac{\sup_{z\in[-C,C]} \psi''(z)}{2\omega} < 1 + \sup_{z\in[-C,C]} \psi''(z).$$

Therefore, (96) holds if the following holds

$$\omega \geq \frac{\sup_{|z|<C} |\psi'''(z)| \cdot D_1}{\lambda_0} \left(\lambda_0 \left(1 + \sup_{|z|<C} \psi''(z)\right) + D_2 \left(1 + \sup_{|z|<C} \psi''(z)\right)^2\right) \triangleq \zeta_2(C). \tag{97}$$

Now, assume $\omega$ satisfies

$$\omega > \max\{1 + \inf_{C'>D_1} \zeta_1(C'), \zeta_2(C^*)\}, \quad C^* = \sup\{C > D_1 : \zeta_1(C) < 1 + \inf_{C'>D_1} \zeta_1(C')\}.$$

Since $\omega > 1 + \inf_{C'>D_1} \zeta_1(C')$, then for any $C$ that satisfies

$$C \in \{C > D_1 : \zeta_1(C) < 1 + \inf_{C'>D_1} \zeta_1(C')\}, \tag{98}$$

we have (94) holds and thus (72) holds. At the same time, since $\zeta_2$ is non-decreasing, we have for any $C$ that satisfies (98),

$$\omega > \zeta_2(C^*) \geq \zeta_2(C),$$

which implies (82) also holds. This concludes the proof. $\square$

In the following result, we show that when $\phi$ is the hypentropy potential $\rho_\beta$, the conditions in Proposition 27 and 29 are satisfied with sufficiently large $\beta$.

**Proposition 31.** *Consider a potential function that satisfies the assumptions in Proposition 27. Further, assume that $\phi(x) = \rho_\beta(x)$ is the hypentropy function with sufficiently large $\beta$. Then there exists $C_1 > 0$ that satisfies (72) and (82).*

**Proof.** Notice that

$$\rho_\beta''(x) = \frac{1}{\sqrt{x^2 + \beta^2}}, \quad \rho_\beta'''(x) = -\frac{x}{(x^2 + \beta^2)^{3/2}}.$$

Therefore, we can take $\kappa_1$, $\kappa_2$ and $\kappa_3$ as follows:

$$\kappa_1(C) = \beta, \ \kappa_2(C) = \sqrt{C^2 + \beta^2}, \ \kappa_3(C) = \left|\frac{C}{(C^2 + \beta^2)^{3/2}}\right|.$$

Therefore, given any fixed $C > 0$, we have

$$\lim_{\beta \to \infty} \frac{C\kappa_1(C)}{\kappa_2(C)} = \lim_{\beta \to \infty} \frac{C\beta}{\sqrt{C^2 + \beta^2}} = C.$$

Therefore, by selecting $C_1 = 2\frac{8\sqrt{m}(B_x+1)K\|\boldsymbol{f}(0)-\boldsymbol{y}\|_2}{\lambda_0}$, we have that for sufficiently large $\beta$,

$$\frac{C_1\kappa_1(C_1)}{\kappa_2(C_1)} \geq \frac{C_1}{2} = \frac{8\sqrt{m}(B_x+1)K\|\boldsymbol{f}(0) - \boldsymbol{y}\|_2}{\lambda_0}.$$

This implies (72) holds. Now fix $C_1$. Since $\kappa_2(C_1)\kappa_3(C_1) = |\frac{C_1}{\sqrt{C_1^2+\beta^2}}|$, we have $\kappa_2(C_1)\kappa_3(C_1) \to$

$0$ as $\beta \to \infty$. Meanwhile, notice that $\frac{\kappa_2(C_1)}{\kappa_1(C_1)} \to 1$ as $\beta \to \infty$. It then follows that

$$\lim_{\beta \to \infty} \kappa_2(C_1)\kappa_3(C_1)\Big(\lambda_0 \frac{\kappa_2(C_1)}{\kappa_1(C_1)} + 10m(B_x+1)^2 K^2 \Big(\frac{\kappa_2(C_1)}{\kappa_1(C_1)}\Big)^2\Big) = 0.$$

Therefore, with large enough $\beta$, we have (82) also holds for $C_1$. This concludes the proof. $\qquad\square$

## B.2 Implicit Bias in Parameter Space

By Theorem 11, the final parameter of the network with absolute value activation function whose only output weights are trained is the solution to following problem:

$$\min_{\tilde{\boldsymbol{a}} \in \mathbb{R}^n} D_\Phi(\tilde{\boldsymbol{a}}, \boldsymbol{a}(0)) \quad \text{s.t.} \sum_{k \in [n]} \tilde{a}_k |w_k(0)^T x_i - b_k(0)| = y_i, \ \forall i \in [m]. \tag{99}$$

Similar to the case of ReLU networks, we can write infinitely wide absolute-value networks as follows:

$$g(x, \alpha) = \int \alpha(w, b)|w^T x - b| \, \mathrm{d}\mu.$$

Note the above integral always annihilates the odd component of $\alpha$ since $|w^T x - b|$ is an even function of $(w, b)$ and $\mu$ is symmetric. Therefore, for networks with absolute value activation functions, we lose no generality to restrict the parameter space to the space of even and continuous functions:

$$\Theta_{\mathrm{Abs}} = \{\alpha \in \Theta_{\mathrm{ReLU}} : \alpha \text{ is even and uniformly continuous}\}. \tag{100}$$

In the following result, we characterize the limit of the solution to (99) when $n$ tends to infinity.

**Proposition 32.** *Let $g_n$ and $g$ be defined as follows:*

$$\begin{cases} g_n(x, \alpha) = \int \alpha(w, b)|w^T x - b| \, \mathrm{d}\mu_n, \\ g(x, \alpha) = \int \alpha(w, b)|w^T x - b| \, \mathrm{d}\mu. \end{cases} \tag{101}$$

*Assume $\bar{\boldsymbol{a}}$ is the solution to (99). Assume the potential function $\Phi$ can be written in the form of $\Phi(\theta) = \frac{1}{n^2}\sum_{k=1}^{3n+1} \phi\big(n(\theta_k - \hat{\theta}_k)\big)$, where $\phi$ is a real-valued $C^3$-smooth function on $\mathbb{R}$ and has strictly positive second derivatives everywhere. Let $\Theta_{\mathrm{Abs}}$ be defined as in (100) and $\bar{\alpha}_n \in \Theta_{\mathrm{Abs}}$ be any function that satisfies $\bar{\alpha}_n(w_k(0), b_k(0)) = n(\bar{a}_k - a_k(0))$. Assume $\bar{\alpha}$ is the solution of the following problem:*

$$\min_{\alpha \in \Theta_{\mathrm{Abs}}} \int D_\phi\big(\alpha(w, b), 0\big) \mathrm{d}\mu, \quad \text{s.t. } g(x_i, \alpha) = y_i - f(x_i, \hat{\theta}), \ \forall i \in [m]. \tag{102}$$

*Then given any $x \in \mathbb{R}^d$, with high probability over initialization, $\lim_{n \to \infty} |g_n(x, \bar{\alpha}_n) - g(x, \bar{\alpha})| = 0$.*

We will need the following two lemmas for the proof. We omit the proof of the Lemmas to maintain clarity.

**Lemma 33** (Jacod & Sørensen, 2018, Theorem 2.5). *For $n \in \mathbb{N}$, let $G_n(\cdot) = G_n(\cdot, \boldsymbol{z})$ be a random $\mathbb{R}^m$-valued function defined on $\mathbb{R}^m$, where $\boldsymbol{z} = (z_1, \cdots, z_n)$ consists of $n$ independent samples from a pre-specified random variable $z_k \sim \mathcal{Z}$. Let $G(\cdot)$ be a deterministic $\mathbb{R}^m$-valued function defined on $\mathbb{R}^m$. Assume there exist $\bar{\lambda} \in \mathbb{R}^m$ and a neighborhood $U$ of $\bar{\lambda}$ such that the following hold:*

i) $G_n(\bar{\lambda}) \xrightarrow{P} 0$ *as* $n \to \infty$, *and* $G(\bar{\lambda}) = 0$.

ii) *For any* $z$, $G_n(\cdot)$ *and* $G(\cdot)$ *are continuously differentiable on* $\mathbb{R}^m$, *and as* $n \to \infty$

$$\sup_{\lambda \in U} \|J_\lambda G_n(\lambda) - J_\lambda G(\lambda)\|_F \xrightarrow{P} 0.$$

iii) *The matrix* $J_\lambda G(\bar{\lambda})$ *is non-singular.*

*Then there exists a sequence* $\{\bar{\lambda}^{(n)}\}_{n \in \mathbb{N}}$ *such that* $G_n(\lambda^{(n)}) = 0$ *and*

$$\|\bar{\lambda}^{(n)} - \bar{\lambda}\|_\infty \xrightarrow{P} 0.$$

**Lemma 34** (Newey & McFadden, 1994, Lemma 2.4). *Let* $a(\cdot, z)$ *be a random real-valued function on a compact set* $U \subset \mathbb{R}^m$. *Let* $z_1, \ldots, z_n$ *be independent samples from a pre-specified random variable* $\mathcal{Z}$. *Let* $\mu_n$ *denote the empirical measure associated with samples* $\{z_k\}_{k=1}^n$ *and* $\mu$ *denote the probability measure of* $\mathcal{Z}$. *Assume the following hold:*

i) *For any* $z$, $a(\cdot, z)$ *is continuous.*

ii) *There exists* $d(z)$ *such that* $|a(\lambda, z)| < d(z)$ *holds for all* $\lambda \in U$ *and* $\mathbb{E}_{\mathcal{Z}}[d(z)] < \infty$.

*Then*

$$\sup_{\lambda \in U} \left| \int a(\lambda, z) \mathrm{d}\mu_n - \int a(\lambda, z) \mathrm{d}\mu \right| \xrightarrow{P} 0.$$

We now present the proof of Proposition 32.

**Proof.** [Proof of Proposition 32] To ease the notation, for $k \in [n]$, let $z_k = (w_k(0), b_k(0)) \in \mathbb{R}^{d+1}$ denote the input weight and bias associated with the $k$-th neuron at initialization. For $j \in [m]$, define $\sigma_j(z) = \sigma_j(w, b) = |w^T x_j - b|$. Let $\mathcal{Z}$ denote the random vector $(\mathcal{W}, \mathcal{B})$, and $\tilde{y}_i = y_i - f(x_i, \hat{\theta})$.

Notice that for any $\boldsymbol{a} \in \mathbb{R}^n$ and $\alpha$ that satisfies $\alpha(w_k(0), b_k(0)) = n(a_k - a_k(0))$ for $k \in [n]$, we have

$$D_\Phi(\boldsymbol{a}, \boldsymbol{a}(0)) = \frac{1}{n^2} \sum_{k \in [n]} \Big( \phi(n(a_k - a_k(0))) - \phi(0) - n\phi'(0)(a_k - a_k(0)) \Big)$$

$$= \frac{1}{n} \int \phi(\alpha(w, b)) - \phi(0) - \phi'(0)\alpha(w, b) \mathrm{d}\mu_n$$

$$= \frac{1}{n} \int D_\phi(\alpha(w, b), 0) \mathrm{d}\mu_n.$$

Therefore, since $\bar{\boldsymbol{a}}$ is the solution to (99), we have that $\bar{\alpha}_n$ solves:

$$\min_{\alpha_n \in \Theta_{\mathrm{Abs}}} \int D_\phi(\alpha_n(w, b), 0) \mathrm{d}\mu_n \quad \text{s.t.} \quad g_n(x_i, \alpha_n) = \tilde{y}_i, \quad i \in [m]. \tag{103}$$

The Lagrangian of problem 103 can be written as

$$L(\alpha_n, \boldsymbol{\lambda}) = \int D_\phi(\alpha_n(w, b), 0) \mathrm{d}\mu_n - \sum_{j=1}^m \lambda_j \Big( \int \alpha_n(z) \sigma_j(z) \mathrm{d}\mu_n - \tilde{y}_i \Big).$$

Here $\lambda = (\lambda_1, \ldots, \lambda_m)^T \in \mathbb{R}^m$ is the vector Lagrangian multipliers. The optimality condition $\nabla_{\alpha_n} L = 0$ implies that

$$0 = \phi'(\alpha_n(z_k)) - \phi'(0) - \sum_{j=1}^m \lambda_j \sigma_j(z_k), \quad \forall k \in [n].$$

By Assumption, $\phi''(x) > 0$ everywhere and $\phi \in C^3(\mathbb{R})$. Therefore, $\phi'$ is invertible and $(\phi')^{-1}$ is continuously differentiable. Therefore, $\bar{\alpha}_n$ must satisfy:

$$\bar{\alpha}_n(z_k) = (\phi')^{-1} \Big( \phi'(0) + \sum_{j=1}^m \bar{\lambda}_j^{(n)} \sigma_j(z_k) \Big), \quad \forall k \in [n],$$

where $\bar{\lambda}^{(n)}$ is determined by the following equations

$$\int (\phi')^{-1}\Big(\phi'(0) + \sum_{j=1}^{m} \bar{\lambda}_j^{(n)} \sigma_j(z)\Big) \cdot \sigma_i(z) \, \mathrm{d}\mu_n = \tilde{y}_i, \quad \forall i \in [m], \tag{104}$$

Problem (103) is strictly convex in the quotient space of $\Theta_{\mathrm{Abs}}$ by the subspace of functions with zero $\ell_2(\mu_n)$ semi-norm. Hence, $\bar{\alpha}_n$ is unique in the $\ell_2(\mu_n)$ sense. Note that $(\phi')^{-1}$ is injective, and that by Lemma 13, for large enough $n$ and with high probability, the matrix $J_{\boldsymbol{a}} \boldsymbol{f}(0) = (\sigma_j(z_k))_{j,k}$ is full row-rank. Therefore, the Lagrangian multipliers $\bar{\lambda}^{(n)}$ is uniquely determined by (104).

According to (104), we can select $\bar{\alpha}_n$ as

$$\bar{\alpha}_n(z) = (\phi')^{-1}\Big(\phi'(0) + \sum_{j=1}^{m} \bar{\lambda}_j^{(n)} \sigma_j(z)\Big), \quad \forall z \in \mathrm{supp}(\mu). \tag{105}$$

Note $\mathrm{supp}(\mu)$ is a compact set and $(\phi')^{-1}$ is continuous, we have $\bar{\alpha}_n$ defined above is even and uniformly continuous and therefore lies in $\Theta_{\mathrm{Abs}}$.

Let $\boldsymbol{z} = (z_1, \ldots, z_n)^T$. We define a random $\mathbb{R}^m$-valued function as follows:

$$\big(G_n(\lambda, \boldsymbol{z})\big)_i \triangleq \int (\phi')^{-1}\Big(\phi'(0) + \sum_{j=1}^{m} \lambda_j \sigma_j(z)\Big) \cdot \sigma_i(z) \, \mathrm{d}\mu_n - \tilde{y}_i, \quad i = 1, \ldots, m.$$

Then by (104), $\bar{\lambda}^{(n)}$ is a root of $G_n$, and, as we have shown, it is the unique root.

Similarly, according to the optimality condition, the solution $\bar{\alpha}$ to problem (102) can be given by

$$\bar{\alpha}(z) = (\phi')^{-1}\Big(\phi'(0) + \sum_{j=1}^{m} \bar{\lambda}_j \sigma_j(z)\Big), \quad \forall z \in \mathrm{supp}(\mu), \tag{106}$$

where $\bar{\lambda}$ is a root of the following deterministic $\mathbb{R}^m$-valued function:

$$\big(G(\lambda)\big)_i \triangleq \int (\phi')^{-1}\Big(\phi'(0) + \sum_{j=1}^{m} \lambda_j \sigma_j(z)\Big) \cdot \sigma_i(z) \, \mathrm{d}\mu - \tilde{y}_i, \quad i = 1, \ldots, m.$$

Since problem (102) is strictly convex in $\Theta_{\mathrm{Abs}}$, $\bar{\alpha}$ is unique. At the same time, note that $(\phi)^{-1}$ is injective, and that $\{\sigma_j\}_{j=1}^{m}$ are linearly independent in $L^2(\mu)$, as we have shown in the proof of Proposition 13. Hence, $\bar{\lambda}$ is the unique root of $G$.

Now we aim to prove that $\bar{\lambda}^{(n)}$ converges to $\bar{\lambda}$ in probability. Notice that, by law of large number, $G_n(\lambda)$ converges to $G(\lambda)$ in probability for any fixed $\lambda$. This implies that

$$G_n(\bar{\lambda}) \xrightarrow{P} G(\bar{\lambda}) = 0.$$

Since $\phi \in C^3(\mathbb{R})$, we know for any fixed $\boldsymbol{z}$, $G_n(\lambda, \boldsymbol{z})$ and $G(\lambda)$ are continuously differentiable in $\lambda$. To ease the notation, let $\xi(x) = (\phi')^{-1}(x)$. Then we have that

$$\begin{cases} \big(J_\lambda G_n(\lambda)\big)_{ij} &= \int \xi'\Big(\phi'(0) + \sum_{l=1}^{m} \lambda_l \sigma_l(z)\Big) \sigma_i(z) \sigma_j(z) \, \mathrm{d}\mu_n, \\ \big(J_\lambda G(\lambda)\big)_{ij} &= \int \xi'\Big(\phi'(0) + \sum_{l=1}^{m} \lambda_l \sigma_l(z)\Big) \sigma_i(z) \sigma_j(z) \, \mathrm{d}\mu. \end{cases}$$

Let $U$ be a compact neighborhood of $\bar{\lambda}$. Notice that $\mu$ is compactly supported and $\xi'$ is continuous. Then there exists a constant $K$ such that

$$\xi'\big(\phi'(0) + \sum_{l=1}^{m} \lambda_l \sigma_l(z)\big) \le K, \quad \forall \lambda \in U, z \in \mathrm{supp}(\mu).$$

Define $d_{ij}(z) = K\sigma_i(z)\sigma_j(z)$. Then we have that

$$\left|\xi'\Big(\phi'(0) + \sum_{l=1}^{m}\lambda_l\sigma_l(z)\Big)\sigma_i(z)\sigma_j(z)\right| \leq K\sigma_i(z)\sigma_j(z) = d_{ij}(z), \quad \forall\lambda \in U.$$

Meanwhile, it's clear that for any $i, j \in [m]$, there exists a polynomial $P_{ij}(z)$ of $z$, whose degree is independent of $n$, such that $|d_{ij}(z)| \leq |P_{ij}(z)|$ holds for any $z \in \text{supp}(\mu)$. Recall $\mathcal{Z}$ is a bounded random variable. Therefore, $\mathbb{E}_{\mathcal{Z}}(d_{ij}(z)) < \infty$. Then, by Lemma 34, we have

$$\sup_{\lambda \in U}\left|\Big(J_\lambda G_n(\lambda)\Big)_{ij} - \Big(J_\lambda G(\lambda)\Big)_{ij}\right| \xrightarrow{P} 0.$$

Notice the size of the matrix $J_\lambda G_n$ and $J_\lambda G$ is independent of $n$. With a union bound we then have

$$\sup_{\lambda \in U}\|J_\lambda G_n(\lambda) - J_\lambda G(\lambda)\|_F \xrightarrow{P} 0.$$

Next we show $J_\lambda G(\bar{\lambda})$, given below, is non-singular,

$$\Big(J_\lambda G(\bar{\lambda})\Big)_{ij} = \int \xi'\Big(\phi'(0) + \sum_{l=1}^{m}\bar{\lambda}_l\sigma_l(z)\Big)\sigma_i(z)\sigma_j(z)\,\mathrm{d}\mu.$$

To this end, we consider the map $\langle\cdot,\cdot\rangle_* \colon L_2(\mu) \times L_2(\mu) \to \mathbb{R}$ defined by

$$\langle f, g\rangle_* \triangleq \int \xi'\Big(\phi'(0) + \sum_{l=1}^{m}\bar{\lambda}_l\sigma_l(z)\Big)f(z)g(z)\,\mathrm{d}\mu, \quad \forall f, g \in L_2(\mu).$$

Clearly, $\langle\cdot,\cdot\rangle_*$ is symmetric and linear. For any $f \in L_2(\mu)$ with $f \neq 0$, i.e. $\mu(\{z\colon f(z) \neq 0\}) > 0$. Then there exists a compact set $A \subset \text{supp}(\mu)$ such that $\mu(A \cap \{f(z) \neq 0\}) > 0$. Since $\{\sigma_l\}_{l\in[m]}$ are continuous and $\{\bar{\lambda}_l\}_{l\in[m]}$ are fixed, the set

$$A' \triangleq \{\sum_{l=1}^{m}\bar{\lambda}_l\sigma_l(z)\colon z \in A\} \subset \mathbb{R}$$

is non empty and bounded. Then by noticing $\xi'$ is continuous and $\xi'(x) > 0, \forall x \in \mathbb{R}$, we have

$$\langle f, f\rangle_* = \int \xi'\Big(\phi'(0) + \sum_{l=1}^{m}\bar{\lambda}_l\sigma_l(z)\Big)f^2(z)\,\mathrm{d}\mu$$

$$\geq \inf_{x\in A'}\xi'(x)\cdot\int_A (f(z))^2\,\mathrm{d}\mu$$

$$> 0.$$

Therefore, $\langle\cdot,\cdot\rangle_*$ is a well-defined inner product on $L^2(\mu)$. Noting that $J_\lambda G(\bar{\lambda})$ can be the viewed as the Gram matrix of $\{\sigma_l\}_{l=1}^{m}$ with respect to $\langle\cdot,\cdot\rangle_*$, and that $\{\sigma_l\}_{l=1}^{m}$ are linearly independent in $L^2(\mu)$, we have $J_\lambda G(\bar{\lambda})$ is non-singular. Since $\bar{\lambda}^{(n)}$ and $\bar{\lambda}$ is the unique root of $G_n(\cdot)$ and $G(\cdot)$, respectively, by Lemma 33 we have

$$\|\bar{\lambda}^{(n)} - \bar{\lambda}\|_\infty = o_p(1). \tag{107}$$

Now consider a fixed $x \in R$. We aim to show that $|g_n(x,\bar{\alpha}) - g(x,\bar{\alpha})| = o_p(1)$. Recall that

$$\begin{cases} g_n(x,\bar{\alpha}) = \int \bar{\alpha}(w,b)|w^Tx - b|\,\mathrm{d}\mu_n, \\ g(x,\bar{\alpha}) = \int \bar{\alpha}(w,b)|w^Tx - b|\,\mathrm{d}\mu. \end{cases}$$

Since $\bar{\alpha}$ and the absolute value functions are continuous and $\mu$ is compactly supported, by law of large number, we have

$$|g_n(x,\bar{\alpha}) - g(x,\bar{\alpha})| = o_p(1). \tag{108}$$

Hence, it remains to show $|g_n(x, \bar{\alpha}_n) - g_n(x, \bar{\alpha})| = o_p(1)$. Notice that $\bar{\lambda}$ is a fixed point in $\mathbb{R}^m$ and $\bar{\lambda}^{(n)}$ converges to $\bar{\lambda}$ in probability, and that $\mu$ is compact support. Then there exists a constant $K' > 0$ independent of $n$ such that with high probability,

$$\left| \phi'(0) + \sum_{j=1}^{m} \bar{\lambda}_j^{(n)} \sigma_j(w, b) \right|, \ \left| \phi'(0) + \sum_{j=1}^{m} \bar{\lambda}_j \sigma_j(w, b) \right| \le K', \quad \forall (w, b) \in \operatorname{supp}(\mu).$$

Notice $(\phi')^{-1}$ is continuously differentiable on $\mathbb{R}$ and hence is Lipschitz continuous on $[-K', K']$. Let $L$ denote the corresponding Lipschitz coefficient. Then we have that

$$|g_n(x, \bar{\alpha}_n) - g_n(x, \bar{\alpha})|$$

$$\le \int \left| \bar{\alpha}_n(w, b) - \bar{\alpha}(w, b) \right| \sigma(w^T x - b) \, \mathrm{d}\mu_n$$

$$\le \int \left| (\phi')^{-1} \left( \phi'(0) + \sum_{j=1}^{m} \bar{\lambda}_j^{(n)} \sigma_j(w, b) \right) - (\phi')^{-1} \left( \phi'(0) + \sum_{j=1}^{m} \bar{\lambda}_j \sigma_j(w, b) \right) \right| \sigma(w^T x - b) \, \mathrm{d}\mu_n$$

$$\le \int L \left| \sum_{j=1}^{m} (\bar{\lambda}_j^{(n)} - \bar{\lambda}_j) \sigma_j(w, b) \right| \sigma(w^T x - b) \, \mathrm{d}\mu_n$$

$$\le \int L \|\bar{\lambda}^{(n)} - \bar{\lambda}\|_\infty \sum_{j=1}^{m} \left( \sigma(w^T x_j - b) \right) \cdot \sigma(w^T x - b) \, \mathrm{d}\mu_n$$

$$\le \left( L \int \sigma(w^T x - b) \sum_{j=1}^{m} \sigma(w^T x_j - b) \, \mathrm{d}\mu_n \right) \|\bar{\lambda}^{(n)} - \bar{\lambda}\|_\infty.$$

By the law of large numbers,

$$\int \sigma(w^T x - b) \sum_{j=1}^{m} \sigma(w^T x_j - b) \, \mathrm{d}\mu_n \xrightarrow{P} \int \sigma(w^T x - b) \sum_{j=1}^{m} \sigma(w^T x_j - b) \, \mathrm{d}\mu.$$

Hence $\int \sigma(w^T x - b) \sum_{j=1}^{m} \sigma(w^T x_j - b) \, \mathrm{d}\mu_n$ can be bounded by a finite number independent of $n$. Then by (107), we have

$$|g_n(x, \bar{\alpha}_n) - g_n(x, \bar{\alpha})| = o_p(1). \tag{109}$$

Finally, combine (108) and (109), we have

$$|g_n(x, \bar{\alpha}_n) - g(x, \bar{\alpha})| < |g_n(x, \bar{\alpha}_n) - g_n(x, \bar{\alpha})| + |g_n(x, \bar{\alpha}) - g(x, \bar{\alpha})| = o_p(1),$$

which concludes the proof. $\qquad\square$

### B.3 FUNCTION DESCRIPTION OF THE IMPLICIT BIAS

Assume the input dimension is one, i.e., $d = 1$. Now we study the solution to problem (102) in the function space

$$\mathcal{F}_{\text{Abs}} = \left\{ g(x, \alpha) = \int \alpha(w, b) |wx - b| \mathrm{d}\mu \colon \alpha \in \Theta_{\text{Abs}} \right\}.$$

In the following result, we show that $\mathcal{F}_{\text{Abs}}$ is a subspace of $\mathcal{F}_{\text{ReLU}}$ which can be obtained by eliminating all non-zero affine functions in $\mathcal{F}_{\text{ReLU}}$.

**Proposition 35.** *Let $\mathcal{F}_{\text{Affine}}$ denote the space of affine functions on $\mathbb{R}$. Then $\mathcal{F}_{\text{ReLU}} = \mathcal{F}_{\text{Abs}} \oplus \mathcal{F}_{\text{Affine}}$. Moreover, if $h \in \mathcal{F}_{\text{Abs}}$, then $\mathcal{G}_2(h) = 0$ and $\mathcal{G}_3(h) = 0$, where $\mathcal{G}_2$ and $\mathcal{G}_3$ are defined in (7).*

**Proof.** [Proof of Proposition 35] We first show both $\mathcal{F}_{\text{Abs}}$ and $\mathcal{F}_{\text{Affine}}$ are subspaces of $\mathcal{F}_{\text{ReLU}}$. Assume $h(x) = \int \alpha(w, b) |wx - b| \mathrm{d}\mu \in \mathcal{F}_{\text{Abs}}$. Notice the ReLU activation function can be decomposed as $[z]_+ = z/2 + |z|/2$. Since $\alpha$ is an even function, we have for all $x \in \mathbb{R}$,

$$h(x) = \int \alpha(w, b) |wx - b| \mathrm{d}\mu = \int 2\alpha(w, b) [wx - b]_+ \mathrm{d}\mu.$$

Since $2\alpha \in \Theta_{\text{ReLU}}$, $\mathcal{F}_{\text{Abs}} \subset \mathcal{F}_{\text{ReLU}}$. Now assume $h(x) = zx + w$ with $z, w \in \mathbb{R}$. Consider $\alpha_h$ defined by

$$\begin{cases} \alpha_h(1, b) = -\dfrac{2w}{\mathbb{E}[\mathcal{B}^2]}b + 2z; \\ \alpha_h(-1, b) = -\alpha_h(1, -b). \end{cases}$$

Note $\alpha_h$ is an odd function. Then we have for any $x \in \mathbb{R}$,

$$\int \alpha_h(w, b)[wx - b]_+ \mathrm{d}\mu = \int \alpha_h(1, b)\frac{x - b}{2}\mathrm{d}\mu_{\mathcal{B}}$$

$$= \left(\int_{\text{supp}(p_{\mathcal{B}})} \frac{\alpha_h(1, b)}{2}p_{\mathcal{B}}(b)\mathrm{d}b\right)x - \left(\int_{\text{supp}(p_{\mathcal{B}})} \frac{\alpha_h(1, b)}{2}p_{\mathcal{B}}(b)b\mathrm{d}b\right)$$

$$= \left(\int_{\text{supp}(p_{\mathcal{B}})} zp_{\mathcal{B}}(b)\mathrm{d}b\right)x - \left(\int_{\text{supp}(p_{\mathcal{B}})} -\frac{w}{\mathbb{E}[\mathcal{B}^2]}p_{\mathcal{B}}(b)b^2\mathrm{d}b\right)$$

$$= zx + w.$$

Hence, $\mathcal{F}_{\text{Affine}} \subset \mathcal{F}_{\text{ReLU}}$.

Next we show $\mathcal{F}_{\text{Abs}} \cap \mathcal{F}_{\text{Affine}} = \{0\}$, i.e. the only affine function in $\mathcal{F}_{\text{Abs}}$ is the zero function. Assume $h(x) = \int \alpha(w, b)|wx - b|\mathrm{d}\mu \in \mathcal{F}_{\text{Abs}}$ is an affine function. For $x \in (\text{supp}(p_{\mathcal{B}}))^{\circ}$, by Leibniz integral rule we have

$$h''(x) = 2\alpha(1, x)p_{\mathcal{B}}(x). \tag{110}$$

Since $h$ is an affine function, we have $2\alpha(1, x)p_{\mathcal{B}}(x) = 0$. Now since $p_{\mathcal{B}}(x) > 0$, $\alpha(1, x) = \alpha(-1, -x) = 0$, which implies $h(x) = 0$ for $x \in \mathbb{R}$.

Next we show that for any $h \in \mathcal{F}_{\text{ReLU}}$, there exists $h_1 \in \mathcal{F}_{\text{Abs}}$ and $h_2 \in \mathcal{F}_{\text{Affine}}$ such that $h = h_1 + h_2$. This can be directly verified by the following computation: for any $\alpha \in \Theta_{\text{ReLU}}$,

$$\int \alpha(w, b)[wx - b]_+ \mathrm{d}\mu = \int \alpha^+(w, b)\frac{|wx - b|}{2}\mathrm{d}\mu + \int \alpha^-(w, b)\frac{wx - b}{2}\mathrm{d}\mu$$

$$= \int \alpha^+(w, b)\frac{|wx - b|}{2}\mathrm{d}\mu + \int \frac{\alpha^-(1, b)}{2}\mathrm{d}\mu_{\mathcal{B}} \cdot x - \int \frac{\alpha^-(1, b)b}{2}\mathrm{d}\mu_{\mathcal{B}}.$$

Therefore, we have $\mathcal{F}_{\text{ReLU}} = \mathcal{F}_{\text{Abs}} \oplus \mathcal{F}_{\text{Affine}}$.

We now show $\mathcal{G}_2$ and $\mathcal{G}_3$ vanish on $\mathcal{F}_{\text{Abs}}$. Without loss of generality, we assume $\text{supp}(p_{\mathcal{B}}) = [m, M]$ where $m, M$ are finite numbers. Assume $h(x) = \int \alpha(w, b)|wx - b|\mathrm{d}\mu \in \mathcal{F}_{\text{Abs}}$ Then for $x > M$, we have

$$h(x) = \int_m^M \alpha(1, b)(x - b)p_{\mathcal{B}}(b)\mathrm{d}b$$

and

$$h'(x) = \int_m^M \alpha(1, b)p_{\mathcal{B}}(b)\mathrm{d}b.$$

Similarly, for $x < m$ we have

$$h(x) = \int_m^M \alpha(1, b)(-x + b)p_{\mathcal{B}}(b)\mathrm{d}b$$

and

$$h'(x) = -\int_m^M \alpha(1, b)p_{\mathcal{B}}(b)\mathrm{d}b.$$

Therefore, $h'(+\infty) + h'(-\infty) = 0$ and $\mathcal{G}_2(h) = 0$.

To show $\mathcal{G}_3(h) = 0$, we consider the following function

$$s(x) = h(x) - \int \alpha_h(w, b)|wx - b|\mathrm{d}\mu,$$

where $\alpha_h \in \Theta_{\text{Abs}}$ is defined by

$$\alpha_h(1, b) = \frac{h''(b)}{2p_{\mathcal{B}}(b)}.$$

It is clear that $s(x) \in \mathcal{F}_{\mathrm{Abs}} \subset \mathcal{F}_{\mathrm{Relu}}$. By Proposition 20, $s''$ vanishes when $x \neq \mathrm{supp}(p_{\mathcal{B}})$. For $x \in (\mathrm{supp}(p_{\mathcal{B}}))^{\circ}$, similar to computation in (110) we also have $s''(x) = 0$. Therefore, $s'$ is a piece-wise affine function. According to Proposition 20, $s'$ is continuous. Therefore, $s'$ is a constant function. By what we just shown above, the only constant function in $\mathcal{F}_{\mathrm{Abs}}$ is the zero function. Therefore, $h(x) = g(x, \alpha_h)$ for all $x \in \mathbb{R}$. In particular, we have that

$$h(0) = g(0, \alpha_h) = \int \alpha_h(w, b)|-b|\mathrm{d}\mu = \int_{\mathrm{supp}(p_{\mathcal{B}})} h''(b) \frac{|b|}{2} \mathrm{d}b.$$

Therefore, $\int_{\mathrm{supp}(p_{\mathcal{B}})} h''(b)|b|\mathrm{d}b - 2h(0) = 0$ and $\mathcal{G}_3(h) = 0$. With this, we conclude the proof. $\quad\square$

In the following result, we solve the minimal representation cost problem (17) with the cost functional described in problem (102), i.e. $\mathcal{C}_{\Phi}(\alpha) = \int D_{\phi}(\alpha(w, b), 0) \, \mathrm{d}\mu$.

**Proposition 36.** *Let $g$ be as defined in (101). Given $h \in \mathcal{F}_{\mathrm{Abs}}$, consider the following minimal representation cost problem*

$$\min_{\alpha \in \Theta_{\mathrm{Abs}}} \int D_{\phi}(\alpha(w, b), 0)\mathrm{d}\mu \quad \text{s.t. } g(x, \alpha) = h(x), \ \forall x \in \mathbb{R}. \tag{111}$$

*Then the solution to problem (111), denoted by, $\mathcal{R}_{\Phi}(h)$, is given by*

$$\mathcal{R}_{\Phi}(h)(1, b) = \frac{h''(b)}{2p_{\mathcal{B}}(b)}. \tag{112}$$

**Proof.** [Proof of Proposition 36] Note it suffices to show that the constraint $g(\cdot, \alpha) = h(\cdot)$ in problem (111) is equivalent to

$$\alpha(1, b) = \frac{h''(b)}{2p_{\mathcal{B}}(b)}. \tag{113}$$

If $h(\cdot) = g(\cdot, \alpha)$, then by calculation as in (110) we have $\alpha$ must satisfy (113). This implies (113) is necessary for the equality $g(\cdot, \alpha) = h(\cdot)$. Assume $\alpha$ satisfies (113). Consider the following function

$$s(x) = h(x) - \int \alpha(w, b)|wx - b|\mathrm{d}\mu.$$

As we shown in the proof for Proposition 35, $s(x) = 0$ on $\mathbb{R}$. Therefore, the solution to (111) is given by (112) and we conclude the proof. $\quad\square$

Now we can write down the implicit bias problem in the function space. The following Proposition is a direct result of Proposition 12 and Proposition 36.

**Proposition 37.** *Assume $\bar{\alpha}$ solves problem (102). Then $\bar{h}(x) = g(x, \bar{\alpha})$, as defined in (101), solves the following variational problem*

$$\min_{h \in \mathcal{F}_{\mathrm{Abs}}} \int_{\mathrm{supp}(p_{\mathcal{B}})} D_{\phi}\Big(\frac{h''(x)}{2p_{\mathcal{B}}(x)}, 0\Big)p_{\mathcal{B}}(x)\mathrm{d}x \quad \text{s.t. } h(x_i) = y_i - f(x_i, \hat{\theta}), \ \forall i \in [m]. \tag{114}$$

In the following result, we show that Theorem 9 also holds for hypentropy potentials $\rho_{\beta}(x) = x \, \mathrm{arcsinh}(x/\beta) + \sqrt{x^2 - \beta^2}$ when $\beta$ is sufficiently large.

**Theorem 38** (**Implicit bias of mirror flow with scaled hypentropy**). *Assume the same assumptions as in Theorem 9, except that the potential is given by: $\Phi(\theta) = n^{-2} \sum_{k \in [3n+1]} \rho_{\beta}(n(\theta_k - \hat{\theta}_k))$, where $\rho_{\beta}$ is the hypentropy potential. Assume $\beta$ is sufficiently large. Then with high probability over the random initialization, there exist constant $C_1, C_2 > 0$ such that for any $t \geq 0$,*

$$\|\theta(t) - \hat{\theta}\|_{\infty} \leq C_1 n^{-1}, \ \|\boldsymbol{f}(t) - \boldsymbol{y}\|_2 \leq e^{-\eta_0 C_2 t}\|\boldsymbol{f}(0) - \boldsymbol{y}\|_2$$

*Moreover, assuming univariate input data, i.e., $d = 1$, and letting $\theta(\infty) = \lim_{t \to \infty} \theta(t)$, we have for any given $x \in \mathbb{R}$ that $\lim_{n \to \infty} |f(x, \theta(\infty)) - f(x, \hat{\theta}) - \bar{h}(x)| = 0$, where $\bar{h}(\cdot)$ is the solution to the following variational problem:*

$$\min_{h \in \mathcal{F}_{\mathrm{Abs}}} \int_{\mathrm{supp}(p_{\mathcal{B}})} D_{\rho_{\beta}}\Big(\frac{h''(x)}{2p_{\mathcal{B}}(x)}, 0\Big)p_{\mathcal{B}}(x)\mathrm{d}x \quad \text{s.t. } h(x_i) = y_i - f(x_i, \hat{\theta}), \ \forall i \in [m].$$

*Here $D_{\rho_{\beta}}$ denotes the Bregman divergence on $\mathbb{R}$ induced by the hypentropy potential and is given by $D_{\rho_{\beta}}(x, y) = x \, (\mathrm{arcsinh}(x/\beta) - \mathrm{arcsinh}(y/\beta)) - \sqrt{x^2 + \beta^2} + \sqrt{y^2 + \beta^2}$.*

**Proof.** [Proof of Theorem 38] By Proposition 31, with sufficiently large $\beta$, Proposition 27 and 29 holds for $\phi = \rho_\beta$. Note in Proposition 32 we only require function $\phi$ to be $C^3$-smooth and to have positive second derivatives everywhere, which is satisfied by the hypentropy function. Also, notice Proposition 36 does not use information on the potential function. Therefore, Theorem 9 holds in this case and we conclude the proof. □

## C    DISCUSSION ON PARAMETRIZATION AND INITIALIZATION

In this section, we comment on the parametrization (1) and initialization scheme (2) that we considered in this work, and compare these to NTK parametrization and Anti-Symmetrical Initialization.

**Unit-norm initialized input weights**    In initialization (2), we require the input weights to be sampled from the unit sphere $\mathbb{S}^{d-1}$. This choice aims to make the minimal representation cost problem (17) more tractable. Specifically, by this, for univariate input data we have $\mathrm{supp}(\mu) = \{-1, 1\} \times \mathbb{R}$ and the parameter space for infinitely wide ReLU networks, i.e. the space of uniformly continuous functions on $\mathrm{supp}(\mu)$, becomes much smaller than the case when input weights are sampled from a general sub-Gaussian distribution and $\mathrm{supp}(\mu) = \mathbb{R} \times \mathbb{R}$. We note this is a common setting in literature on representation cost for overparameterized networks (Savarese et al., 2019; Ongie et al., 2020). In Appendix E.2, we further discuss the challenge of solving the minimal representation cost problem for input weights sampled from general distributions.

**Dependence of lazy training on parametrization and initialization**    Lazy training is known to be dependent on the way the model is parameterized and how the parameters are initialized (Chizat et al., 2019). In this work, we consider the parametrization (1) and initialization (2) for shallow ReLU networks. However, there are other settings that also lead to lazy training in gradient descent. Generally, consider a two-layer ReLU network of the following form:

$$f(x) = \frac{1}{s_n} \sum_{i=1}^{n} a_k [w_k^T x - b]_+ + d,$$

where $s_n$ is a scaling factor dependent on the number of hidden units $n$. According to Williams et al. (2019), for gradient descent training, there are two ways to enter the lazy regime: (1) initializing the input weights and biases significantly larger than output weights; and (2) setting $s_n = o(n)$ and considering $n \to \infty$. Infinitely wide networks with standard parametrization, which we consider in this work, aligns with the second approach, as $s_n$ is set to 1.

**Standard vs. NTK parametrization**    When $s_n$ is set to $\sqrt{n}$, we obtain the NTK parametrization (Jacot et al., 2018), in which the networks are parametrized as follows:

$$f(x) = \frac{1}{\sqrt{n}} \sum_{i=1}^{n} a_k \sigma(w_k^T x - b_k) + d.$$

Under NTK parametrization, a typical initialization scheme is to independently sample all parameters from a standard Gaussian distribution or, more generally, from a sub-Gaussian distribution. Note that, compared to standard parametrization in (1) and (2), the factor $1/\sqrt{n}$ appears before the sum, rather than as a scaling factor in the initialization distribution of the output weights. As a result, not only the output weights are normalized at initialization, their gradients are also normalized throughout training. It is known that, although gradient descent under both settings is in the kernel regime (lazy regime), but they correspond to training different kernel models and yield different implicit biases (Williams et al., 2019; Ghorbani et al., 2021). Specifically, noted by Jin & Montúfar (2023), under NTK parametrization one can not approximate the training dynamics of the full model by that of the model whose only output weights are trained. This implies the approach in this work, in particular the formulation of problem 14, can not be directly applied to networks with NTK parametrization.

**Anti-Symmetrical Initialization**    Anti-Symmetrical Initialization (ASI) (Zhang et al., 2020) is a common technique in gradient descent analysis to ensure zero function output at initialization. In the study of implicit bias of gradient descent, Jin & Montúfar (2023) utilize ASI as a base case and then extend to other initializations. However, we observe that for mirror flow, ASI could in contrast greatly complicate the implicit bias problem. In the following we briefly discuss why this happens.

To achieve zero function output at initialization, ASI creates duplicate hidden units for the network with flipped output weight signs. Specifically, a network with ASI is described as follows:

$$f(x, \theta) = \frac{1}{\sqrt{2}} \sum_{k=1}^{n} a_k [w_k^T x - b_k]_+ - \frac{1}{\sqrt{2}} \sum_{k=1}^{n} a_k' [(w_k')^T x - b_k']_+. \tag{115}$$

Here $\theta = \text{vec}(\boldsymbol{W}, \boldsymbol{b}, \boldsymbol{a}, \boldsymbol{W}', \boldsymbol{b}', \boldsymbol{a}')$ and the initial parameter $\theta(0)$ is set as

$$\boldsymbol{W}(0) = \boldsymbol{W}'(0) = \hat{\boldsymbol{W}}, \quad \boldsymbol{b}(0) = \boldsymbol{b}'(0) = \hat{\boldsymbol{b}}, \quad \boldsymbol{a}(0) = \boldsymbol{a}'(0) = \hat{\boldsymbol{a}},$$

where $\hat{\boldsymbol{W}}, \hat{\boldsymbol{b}}$ and $\hat{\boldsymbol{a}}$ are as described in (2). Assume one can simplify the implicit bias of training model (115) to that of training the following linear model as we did for networks without ASI:

$$f(x, (\boldsymbol{a}, \boldsymbol{a}')) = \frac{1}{\sqrt{2}} \sum_{k=1}^{n} (a_k - a_k') \sigma(\hat{w}_k^T x - \hat{b}_k). \tag{116}$$

The corresponding mirror flow is given by

$$\frac{\mathrm{d}}{\mathrm{dt}} (\boldsymbol{a}(t), \boldsymbol{a}'(t)) = -\eta (\nabla^2 \Phi((\boldsymbol{a}(t), \boldsymbol{a}'(t))))^{-1} \cdot J_{(\boldsymbol{a}, \boldsymbol{a}')} \bar{\boldsymbol{f}}(t) \cdot (\bar{\boldsymbol{f}}(t) - \boldsymbol{y}), \tag{117}$$

where $\bar{\boldsymbol{f}}(t) = (f(x_1, (\boldsymbol{a}(t), \boldsymbol{a}'(t))), \dots, f(x_m, (\boldsymbol{a}(t), \boldsymbol{a}'(t))))^T \in \mathbb{R}^m$. Assume (117) is able to attain global optimality. Then by Theorem 11, the implicit bias is described by the following problem:

$$\min_{(\boldsymbol{a}, \boldsymbol{a}') \in \mathbb{R}^{2n}} D_\Phi((\boldsymbol{a}, \boldsymbol{a}'), (\hat{\boldsymbol{a}}, \hat{\boldsymbol{a}}')) \quad \text{s.t.} \ f(x_i, (\boldsymbol{a}, \boldsymbol{a}')) = y_i, \ \forall i \in [m]. \tag{118}$$

However, observe that model (116) is "essentially" parameterized by $\boldsymbol{a} - \boldsymbol{a}'$ rather than by $(\boldsymbol{a}, \boldsymbol{a}')$, meaning that the function properties of $f$ are tied to $\boldsymbol{a} - \boldsymbol{a}'$ instead of $(\boldsymbol{a}, \boldsymbol{a}')$. A gap thus appears: the implicit bias in parameter space describes the behavior of $(\boldsymbol{a}, \boldsymbol{a}')$, whereas the function properties are related to $\boldsymbol{a} - \boldsymbol{a}'$. To fill this gap, one can either solve the minimal representation cost problem with respect to a cost function defined on $(\boldsymbol{a}, \boldsymbol{a}')$ as given in (118), which can be quite challenging, or to characterize $\boldsymbol{a}(\infty) - \boldsymbol{a}'(\infty)$ returned by (117). The latter is also challenging, as $\boldsymbol{a}(t) - \boldsymbol{a}'(t)$ evolves according to the following differential equation:

$$\frac{\mathrm{d}}{\mathrm{dt}} (\boldsymbol{a}(t) - \boldsymbol{a}'(t)) = -\eta \Big( (\nabla^2 \Phi(\boldsymbol{a}(t)))^{-1} + (\nabla^2 \Phi(\boldsymbol{a}'(t)))^{-1} \Big) J_{\boldsymbol{a}} \bar{\boldsymbol{f}}(0) (\bar{\boldsymbol{f}}(t) - \boldsymbol{y}). \tag{119}$$

Noted by Azulay et al. (2021), a possible way to characterize $\boldsymbol{a}(\infty) - \boldsymbol{a}'(\infty)$ is to solve the following equation for $\Psi$:

$$\Big( (\nabla^2 \Phi(\boldsymbol{a}(t)))^{-1} + (\nabla^2 \Phi(\boldsymbol{a}'(t)))^{-1} \Big)^{-1} = \nabla^2 \Psi \Big( \boldsymbol{a}(t) - \boldsymbol{a}'(t) \Big). \tag{120}$$

If such a $\Psi$ exists, we have $\frac{\mathrm{d}}{\mathrm{dt}} (\nabla \Psi(\boldsymbol{a}(t) - \boldsymbol{a}'(t))) = -\eta J_{\boldsymbol{a}} \bar{\boldsymbol{f}}(0) (\bar{\boldsymbol{f}}(t) - \boldsymbol{y})$. It follows that $\boldsymbol{a}(\infty) - \boldsymbol{a}'(\infty)$ is the solution to the following problem:

$$\min_{\boldsymbol{a} - \boldsymbol{a}'} \Psi(\boldsymbol{a} - \boldsymbol{a}') \quad \text{s.t.} \ f(x_i, (\boldsymbol{a}, \boldsymbol{a}')) = y_i, \forall i \in [m].$$

In the special case of gradient flow, i.e., $\Phi(\boldsymbol{a}) = \frac{1}{2} \|\boldsymbol{a}\|_2^2$, we have $\nabla^2 \Phi(\boldsymbol{a}) \equiv I$ for any $\boldsymbol{a} \in \mathbb{R}^n$. Therefore, (120) has a simple solution. However, it is known that (120) has a solution only if $((\nabla^2 \Phi(\boldsymbol{a}(t)))^{-1} + (\nabla^2 \Phi(\boldsymbol{a}'(t)))^{-1})^{-1}$ satisfies the Hessian-map condition (see, e.g., Azulay et al., 2021). This is a very special property and typically does not hold for most potentials.

## D  GRADIENT FLOW WITH REPARAMETRIZATION

In this section, we apply our results to study the implicit bias of gradient flow with a particular class of reparametrization. Recall in this work, we focus on the following mirror flow with respect to potential $\Phi$ for minimizing loss function $L$:

$$\frac{\mathrm{d}}{\mathrm{dt}} \theta(t) = -\eta (\nabla^2 \Phi(\theta(t)))^{-1} \nabla L(\theta(t)). \tag{121}$$

Consider a reparametrization map $\theta = G(\nu)$ and the gradient flow in $\nu$-space for minimizing $L \circ G$:

$$\frac{\mathrm{d}}{\mathrm{dt}} \nu(t) = -\eta \nabla (L \circ G)(\nu(t)). \tag{122}$$

By the chain rule, the dynamics of $\theta(t) = G(\nu(t))$ in the $\theta$-space induced by (122) is given by

$$\frac{\mathrm{d}}{\mathrm{d}t}\theta(t) = -\eta JG(\nu(t))(JG(\nu(t)))^T \nabla L(\theta(t)), \tag{123}$$

where $JG(\nu(t))$ denotes the Jacobian matrix of map $G$ at $\nu(t)$. Notably, the two dynamics, (121) and (123), are identical if the following equality holds for any $\theta = G(\nu)$:

$$\left(\nabla^2 \Phi(\theta)\right)^{-1} = JG(\nu)(JG(\nu))^T. \tag{124}$$

Based on this, we can directly apply our characterization of the implicit bias to gradient flow with a reparametrization that permits a potential function satisfying (124).

**Corollary 39 (Implicit bias of gradient flow with reparametrization).** *Consider a two-layer ReLU network* (1) *with one input unit and $n$ hidden units, where we assume $n$ is sufficiently large. Consider parameter initialization* (2). *Consider any finite training data set $\{(x_i, y_i)\}_{i=1}^m$ that satisfies $x_i \neq x_j$ when $i \neq j$ and $\{x_i\}_{i=1}^M \subset \mathrm{supp}(p_{\mathcal{B}})$. Consider a bijective reparametrization map $G \colon \mathbb{R}^{3n+1} \to \mathbb{R}^{3n+1}, G(\nu) = \theta$ and the gradient flow in $\nu$-space (122) with $\eta = \Theta(1/n)$. Assume there exists a function $\Phi \colon \mathbb{R}^{3n+1} \to \mathbb{R}$ such that $\Phi$ satisfies Assumption 1 and the equality (124) holds for any $\theta = G(\nu)$. Then letting $\nu(\infty) = \lim_{t\to\infty} \nu(t)$, we have for any given $x \in \mathbb{R}$, that $\lim_{n\to\infty} |f(x, G(\nu(\infty))) - f(x, \hat{\theta}) - \bar{h}(x)| = 0$, where $\bar{h}(\cdot)$ is the solution to variational problem* (7).

A similar result holds for scaled potentials and absolute value networks by Theorem 9, which we omit for brevity. We now give concrete examples of reparametrizations that satisfy (124) in Corollary 39. Consider the map $G$ determined by its inverse map $G^{-1}(\theta) = \nu$ as

$$(G^{-1}(\theta))_k = \int_0^{\theta_k} \sqrt{\phi''(x)}\, \mathrm{d}x, \quad k = 1, \ldots, 3n+1,$$

where $\phi \in C^2(\mathbb{R})$ satisfies $\phi''(x) > 0$ for all $x \in \mathbb{R}$. Notice that $JG^{-1}(\theta) = \mathrm{Diag}(\sqrt{\phi''(\theta_k)})$ and by the inverse function theorem, $JG(\nu) = (JG^{-1}(\theta))^{-1} = \mathrm{Diag}((\phi''(\theta_k))^{-1/2})$. It then it follows that (124) holds for $\Phi(\theta) = \sum_{k \in [3n+1]} \phi(\theta_k)$. In the special case when $\phi(x) = \frac{1}{12}x^4 + \frac{1}{2}x^2$, we have

$$(G^{-1}(\theta))_k = \int_0^{\theta_k} \sqrt{x^2 + 1}\, \mathrm{d}x = \frac{\theta_k}{2}\sqrt{\theta_k^2 + 1} + \frac{1}{2}\mathrm{arcsinh}\,\theta_k, \quad k = 1, \ldots, 3n+1.$$

We note that Li et al. (2022) established the equivalence between mirror flow and a broader class of reparametrizations, which the authors refer to as commutative reparametrizations. We did not include discussion on these reparametrizations and leave it as a future research direction.

## E    MINIMAL REPRESENTATION COST PROBLEM IN GENERAL SETTINGS

In this section, we discuss solving the minimal representation cost problems for ReLU networks with respect to a general cost function and for multivariate networks with general parameter initialization.

### E.1    REPRESENTATION COST FOR RELU NETWORK AND GENERAL COST FUNCTION

Consider the following the minimal representation cost problem for a given $h \in \mathcal{F}_{\mathrm{ReLU}}$:

$$\arg\min_{\alpha \in \Theta_{\mathrm{ReLU}}} \int D_\phi(\alpha(w, b), 0)\mathrm{d}\mu \quad \text{s.t. } h(x) = \int \alpha(w, b)[wx - b]_+ \mathrm{d}\mu, \ \forall x \in \mathbb{R}. \tag{125}$$

As we show in Appendix A.3, the constraints in problem (125) is equivalent to

$$\begin{cases} \alpha^+(1, b) = \dfrac{h''(b)}{p_{\mathcal{B}}(b)}, \\[2mm] \displaystyle\int \alpha^-(1, b)\mathrm{d}\mu_{\mathcal{B}} = h'(+\infty) + h'(-\infty), \\[2mm] \displaystyle\int \alpha^-(1, b)b\,\mathrm{d}\mu_{\mathcal{B}} = \int_{\mathbb{R}} h''(b)|b|\mathrm{d}b - 2h(0). \end{cases} \tag{126}$$

For simplicity, assume the function $\phi$ satisfies $\phi(0) = 0$ and $\phi'(0) = 0$. Then the objective function in (125) can be rewritten as follows:

$$\int D_\phi(\alpha(w,b),0)\mathrm{d}\mu = \frac{1}{2}\int \phi(\alpha(1,b)) + \phi(\alpha(-1,b))\mathrm{d}\mu_\mathcal{B}$$
$$= \frac{1}{2}\int \phi(\alpha^+(1,b) + \alpha^-(1,b)) + \phi(\alpha^+(1,-b) - \alpha^-(1,-b))\mathrm{d}\mu_\mathcal{B}.$$

Then by (126) solving problem (125) is equivalent to solving the following problem

$$\arg\min_{\substack{\alpha^-\in\Theta_{\mathrm{ReLU}} \\ \alpha^-:\ \mathrm{odd}}} \int \phi\Big(\frac{h''(b)}{p_\mathcal{B}(b)} + \alpha^-(1,b)\Big) + \phi\Big(\frac{h''(-b)}{p_\mathcal{B}(-b)} - \alpha^-(1,-b)\Big)\mathrm{d}\mu_\mathcal{B}$$

$$\text{s.t.} \begin{cases} \int \alpha^-(1,b)\mathrm{d}\mu_\mathcal{B} = h'(+\infty) + h'(-\infty), \\ \int \alpha^-(1,b)b\mathrm{d}\mu_\mathcal{B} = \int_\mathbb{R} h''(b)|b|\mathrm{d}b - 2h(0). \end{cases} \tag{127}$$

In the special case of $\phi(x) = x^2$, where the subspace of even functions is orthogonal to the subspace of odd functions, the objective functional in problem (127) simplifies to $\int(\alpha^-(1,b))^2\mathrm{d}\mu_\mathcal{B}$. This allows one to explicitly solve the problem. For a general $\phi$, however, obtaining a closed-form solution to problem (127) becomes much more challenging. We emphasize that, for any scaled potential that admits a closed-form solution to problem (127), one can obtain the function description of the corresponding implicit bias with the strategy we introduced in Section 3.3.

### E.2 REPRESENTATION COST FOR MULTIVARIATE NETWORKS AND GENERAL INITIALIZATION

Let $d \geq 2$ denote the input dimension for the network. Assume $(\mathcal{W}, \mathcal{B})$ is a random vector on $\mathbb{R}^d \times \mathbb{R}$, with distribution $\mu$ and density function $p_{\mathcal{W},\mathcal{B}}$. Consider the space of infinitely wide ReLU networks with $d$-dimensional inputs

$$\mathcal{F}_{\mathrm{ReLU}}^{(d)} = \Big\{g(x,\alpha) = \int \alpha(w,b)[w^T x - b]_+ \mathrm{d}\mu\colon \alpha \in \Theta_{\mathrm{ReLU}}^{(d)}\Big\}$$

where $\Theta_{\mathrm{ReLU}}^{(d)} = \{\alpha \in C(\mathrm{supp}(\mu))\colon \alpha \text{ is uniformly continuous}\}$. For a given $h \in \mathcal{F}_{\mathrm{ReLU}}^{(d)}$, consider the minimal representation cost problem with the squared $L^2$ norm cost:

$$\arg\min_{\alpha\in\Theta_{\mathrm{ReLU}}^{(d)}} \int (\alpha(w,b))^2\mathrm{d}\mu \quad \text{s.t. } h(x) = g(x,\alpha), \forall x \in \mathbb{R}^d. \tag{128}$$

For univariate networks, we solve this problem by translating the constraints, $h(x) = g(x,\alpha)$, to more manageable equations as presented in (126). However, this strategy could be more challenging for multivariate networks with general input weights initialization. We discuss this difficulty for multivariate networks with unit-norm initialized input weights and for univariate network with general input weights initialization separately.

For multivariate networks with unit-norm initialized input weights, i.e., $\mathrm{supp}(\mu) \subset \mathbb{S}^{d-1} \times \mathbb{R}$, let $\alpha$ be a candidate for Problem (128), i.e., $h(x) = g(x,\alpha), \forall x \in \mathbb{R}^d$. Determining the even component of $\alpha$ becomes quite challenging. We now present a potential approach with the Radon transform $\mathcal{R}$ and the dual Radon transform $\mathcal{R}^*$ (for formal definition see, e.g., Helgason, 1999). Note that

$$\Delta h(x) = \int \alpha(w,b)\delta(w^T x - b)\mathrm{d}\mu = \mathcal{R}^*\{\alpha p_{\mathcal{W},\mathcal{B}}\}(x). \tag{129}$$

Assuming that $\alpha p_{\mathcal{W},\mathcal{B}}$ is in the Schwartz space $\mathcal{S}(\mathbb{S}^{d-1} \times \mathbb{R})$,[7] then by the inversion formula for the dual Radon transform (Solmon, 1987, Theorem 8.1) one has:

$$\alpha^+(w,b) = \frac{-\mathcal{R}\{(-\Delta)^{\frac{d+1}{2}}h\}(w,b)}{2(2\pi)^{d-1}p_{\mathcal{W},\mathcal{B}}(w,b)}.$$

---

[7]$\alpha \in \mathcal{S}(\mathbb{S}^{d-1} \times \mathbb{R})$ if $\alpha(w,b)$ is $C^\infty$-smooth and all its partial derivatives decreases faster than $(1+|b|^2)^{-k}$ as $b \to \infty$ for any $k > 0$.

However, the assumption on $\alpha p_{\mathcal{W},\mathcal{B}}$ does not generally hold. In particular, consider $\bar{\alpha}$ as the solution to following problem:

$$\underset{\alpha \in \Theta_{\text{ReLU}}^{(d)}}{\arg \min} \int (\alpha(w,b))^2 \mathrm{d}\mu \quad \text{s.t. } g(x_i, \alpha) = y_i, \forall i \in [m]. \tag{130}$$

By analogy with Proposition 17, one can expect $\bar{\alpha}$ to characterize the parameter obtained by mirror flow training for infinitely wide ReLU networks in the multivariate setting. By the first-order optimality condition for problem (130), $\bar{\alpha}$ takes the following form:

$$\bar{\alpha}(w,b) = \sum_{i=1}^{m} \lambda_i [w^T x_i - b]_+,$$

where $\{\lambda_i\}_{i \in [m]}$ are Lagrangian multipliers to be determined. It is clear that either $\bar{\alpha} = 0$ or $\bar{\alpha}$ does not have continuous partial derivatives, implying that $\bar{\alpha} p_{\mathcal{W},\mathcal{B}}$ does not lie in $\mathcal{S}(\mathbb{S}^{d-1} \times \mathbb{R})$ even if $p_{\mathcal{W},\mathcal{B}}$ does. A similar observation has been made by Ongie et al. (2020), who note the difficulty in explicitly characterizing the representation cost with the $L^1$ norm in the multivariate setting. Some studies have relaxed the Schwartz space condition for the inversion formula of the *Radon transform* (Helgason, 1999, Theorem 6.2, Ch. I; Jensen, 2004, Theorem 5.1), which recovers a function $f$ on $\mathbb{R}^d$ using $\mathcal{R}(f)$. However, to the best of our knowledge, no corresponding relaxation has been established for the inversion formula of the *dual Radon transform*, which recovers a function $\alpha$ on $\mathbb{S}^{d-1} \times \mathbb{R}$ using $\mathcal{R}^*(\alpha)$. The latter is necessary for characterizing the solution to problem (128) with the relation given in (129).

For univariate ReLU networks with general input weight initializations, assume $(\mathcal{W}, \mathcal{B})$ is a random vector on $\mathbb{R} \times \mathbb{R}$. Let $h(x) = g(x, \alpha), \forall x \in \mathbb{R}$. When determining the even component of $\alpha$, one can differentiate $h(x)$ twice and obtain the following equality

$$\int_{\mathbb{R}} \alpha^+(w, wx) w^2 p_{\mathcal{W},\mathcal{B}}(w, wx) \mathrm{d}w = h''(x). \tag{131}$$

Then the minimal representation cost problem on the subspace of even functions can be written as

$$\underset{\substack{\alpha^+ \in \Theta_{\text{ReLU}} \\ \alpha^+: \text{ even}}}{\arg \min} \int (\alpha^+(w,b))^2 \mathrm{d}\mu \quad \text{s.t. } \alpha^+ \text{satisfies (131)}. \tag{132}$$

For unit-norm initialized input weights, i.e., $\mathcal{W} = \text{Unif}(\{-1, 1\})$, the constraint (131) becomes a discrete sum and it in fact completely determines $\alpha^+$. Under a general distribution for input weights, solving (132) can be challenging. We note that the unit-norm input weights constraint also appears in other works on representation cost for overparameterized networks (Savarese et al., 2019; Ongie et al., 2020). Jin & Montúfar (2023) considered general initialization but at the same time they required either intractable data adjustments or otherwise adding skip connections to the network.

## F  RIEMANNIAN GEOMETRY OF THE PARAMETER SPACE $(\Theta, \nabla^2 \Phi)$

In this section, we briefly discuss the Riemannian geometry induced by the metric tensor $\nabla^2 \Phi$ on the parameter space.

The Riemannian curvature tensor is a fundamental concept in differential geometry, describing how the space curves at different points and along various directions. For a detailed introduction and rigorous definition of the Riemannian curvature tensor, we refer the reader to Do Carmo & Flaherty Francis (1992). Next we compute the Riemannian curvature tensor of the manifold $(\mathbb{R}^{3n+1}, \nabla^2 \Phi)$, where $\Phi$ is a potential function satisfying Assumption 1.

With the trivial coordinate chart $\text{id}: \mathbb{R}^{3n+1} \to \mathbb{R}^{3n+1}$, the metric tensor $\nabla^2 \Phi$ takes the form of a diagonal matrix $\text{diag}(\phi''(\theta_k))$. The Riemannian curvature is determined by the Christoffel symbols for the Levi-Civita connection (see, e.g., Do Carmo & Flaherty Francis, 1992), which can be computed

as follows:

$$\Gamma_{ij}^k = \frac{1}{2} \sum_{k=1}^{p} g^{km} (\partial_i g_{jm} + \partial_j g_{mi} - \partial_m g_{ij})$$

$$= \frac{1}{2} g^{kk} (\partial_i g_{jk} + \partial_j g_{ki} - \partial_k g_{ij}) \tag{133}$$

$$= \begin{cases} (2\phi''(\theta_k))^{-1} \cdot \phi'''(\theta_k) & i = j = k; \\ 0 & \text{otherwise.} \end{cases}$$

Then the components of the Riemannian curvature tensor is given by:

$$R_{ijk}^l = \Gamma_{ik}^m \Gamma_{jm}^l - \Gamma_{jk}^m \Gamma_{im}^l + \partial_j \Gamma_{ik}^l - \partial_i \Gamma_{jk}^l = 0 \tag{134}$$

This implies the curvature tensor is everywhere zero and the Riemannian manifold $(\Theta, \nabla^2 \Phi)$ is flat everywhere. We remark that a manifold equipped with a metric tensor of the form $\nabla^2 \Phi$ for some general function $\Phi$ is commonly known as a Hessian manifold. We refer the reader to Shima & Yagi (1997) for a broader discussion on the geometry on Hessian manifolds.

## G    EXPERIMENT DETAILS

In this section, we provide details for experiments in Section 4.

**Training data**    For the experiments with unscaled potentials in Figure 1, we use the following training data set: $\{(-1, -0.15), (-0.2, -0.15), (0, 0.15), (0.2, -0.15), (1, -0.15)\}$. In the right panel of Figure 1, we apply the following shift values to the $y$-coordinates of all the data points: $2.5, 1, -0.5, -2, -3.5$. For the experiments with scaled potentials in Figure 2, we consider a different training data set in order to highlight the effect of different scaled potentials: $\{(-1, 0.15), (0.35, 0.15), (0.65, -0.15), (1, 0.15)\}$. Note the $x$-coordinates of all the training data points are contained in $[-1, 1]$.

**Mirror descent training**    We use full-batch mirror descent to train the network. For mirror descent with unscaled potentials, as the number of hidden units $n$ varies, we set the learning rate as $\eta = 1/n$ across all three potentials $\phi_1$, $\phi_2$ and $\phi_3$. For mirror descent with scaled potentials, the learning rate for $\phi_1$ and $\phi_2$ is set to $1/n$ and the learning rate for $\phi_3$ is set to $2/n$. We observe that for scaled potentials, a larger $p$ value in $\phi(x) = |x|^p + x^2$ leads to slower convergence and thus we use a larger learning rate for $\phi_3$. During each training iteration, we manually compute the inverse Hessian matrix, multiply it by the back-propagated gradients, and then update the parameters using the preconditioned gradients. The stopping criterion for the training is that the training loss is lower than the threshold value $10^{-7}$. In the middle panel of Figures 1 and 2, we repeat the training five times with different random initializations for the network. The training is implemented by PyTorch version 2.0.0.

**Numerical solution to the variational problem**    We solve the variational problem using discretization. We discretize the interval $[-1.5, 1.5]$ evenly with 501 points $-1.5 = t_0 < t_1 < \cdots < t_{500} = 1.5$ and then set the vector to be optimized as $u = [h_0, h_1, \ldots, h_{500}, h'_-, h'_+]$ where $h_i$ represents the function value $h(t_i)$ for $i = 0, \ldots, 500$, $h'_-$ represents the slope $h'(-\infty)$, and $h'_+$ represents $h'(+\infty)$. As we shown in Proposition 20, when $\mathcal{B}$ is compactly support on $[-1, 1]$, we have $h'(-\infty) = h'(x)$ for any $x < -1$ and $h'(+\infty) = h'(x)$ for any $x > 1$. Let $(x_k, y_k)$ for $k = 1, \ldots, m$ denote the training data points. s For $k = 1, \ldots, m$, let $i_k = \arg \min_{i \in \{0, \ldots, 500\}} |t_i - x_k|$ record the index for the nearest discretization point to $x_k$. To solve variational problem (7), we consider the following two sets of constraints applying to $u$: (1) $h_{i_k} = y_k$ for $k = 1, \ldots, m$; (2) $(h_k - h_{k-1})/(t_1 - t_0) = h'_-$ for $k \le i_1$ and $(h_k - h_{k-1})/(t_1 - t_0) = h'_+$ for $k \ge i_5$. The second set of constraints is used to ensure $h$ lies in $\mathcal{F}_{\text{ReLU}}$. We estimate the second derivative of $h$ by finite difference method, i.e. $h''_i = (h_{i+1} - 2h_i + h_{i-1})/(t_1 - t_0)^2$ for $i = 1, \ldots, 499$. Noting $\mathcal{G}_3$ has a closed-form formula under uniformly sampled input bias in Theorem 2, we estimate the objective functionals as follows:

$$\begin{cases} \hat{\mathcal{G}}_1(h) = \sum_{i=1}^{499} (h''_i)^2 (t_1 - t_0) \\ \hat{\mathcal{G}}_2(h) = (h'_+ + h'_-)^2 \\ \hat{\mathcal{G}}_3(h) = 3((h'_+ - h'_-) - (h_{i_1} + h_{i_m}))^2. \end{cases}$$

Then we optimize $u$ to minimize the above objective function. To solve variational problem (9), we consider two additional sets of constraints (1) $\hat{\mathcal{G}}_2(h) = 0$; and (2) $\hat{\mathcal{G}}_3(h) = 0$. This is to ensure $h \in \mathcal{F}_{\text{Abs}}$ according to Proposition 35. Then we optimize $u$ to minimize $\hat{\mathcal{G}}_1$. We implement the optimization by CVXPY (Diamond & Boyd, 2016) version 1.4.1. To estimate the $L^\infty$-error between the solution to the variational and the trained networks, we evaluate the trained network on $\{t_i\}_{i=0}^{500}$, denoted by $\{f_i\}_{i=0}^{500}$, and compute the $\ell_\infty$ norm of the difference vector $[f_0 - h_0, \ldots, f_{500} - h_{500}]$.

## H  ADDITIONAL NUMERICAL EXPERIMENTS

In this section, we provide additional numerical experiments on mirror descent training.

**Training trajectory in parameter space**   The left panel in Figure 3 illustrates that, training wide ReLU networks using mirror descent with different unscaled potentials not only have the same implicit bias, they also yield very close parameter trajectories. The right panel in Figure 3 illustrates that, in contrast, training wide networks using mirror descent with different scaled potentials converge to different solution points and have significantly different trajectories.

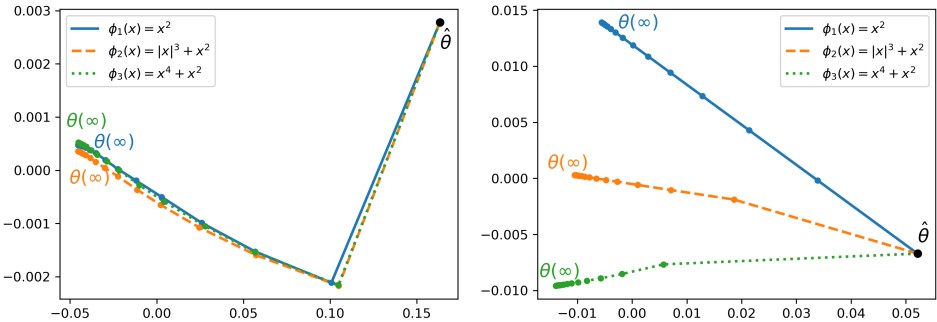

Figure 3: Left: 2D PCA representation of parameter trajectories under mirror descent with unscaled potentials $\phi_1 = x^2$, $\phi_2 = |x|^3 + x^2$, and $\phi_3 = x^4 + x^2$, for ReLU networks with 4860 hidden units. Right: 2D PCA representation of parameter trajectories under mirror descent with scaled potentials $\phi_1$, $\phi_2$, and $\phi_3$, for networks with absolute value activations and 4860 hidden units.

**Lazy regime and kernel regime**   In Figure 4, we examine whether mirror descent with different unscaled potentials: $\phi_1 = x^2$, $\phi_2 = |x|^3 + x^2$, and $\phi_3 = x^4 + x^2$, is in the lazy regime and whether it is in the kernel regime. According to the plot, as the network width $n$ increases, both the distance between the final parameter $\theta(\infty)$ and initial parameter $\hat{\theta}$, and the distance between the final kernel matrix $H(\infty)$ and initial kernel matrix $H(0)$ decrease to zero. This verifies that mirror descent with unscaled potentials is in both lazy regime and kernel regime, as shown in Theorem 2.

In Figure 5, we examine whether mirror descent with scaled potentials falls into lazy regime and whether it falls into kernel regime. Specifically, we observe that $\|\theta(\infty) - \hat{\theta}\|_\infty$ decreases to zero as $n$ increases, which verifies that the training is in lazy regime as we shown in Theorem 9. In the right panel of Figure 5, we see that for potentials having $\phi$ different from $x^2$, the kernel matrix moves significantly from its initial value during training, verifying our claim that networks trained by mirror descent with diverse scaled potentials do not fall in the kernel regime. Note $\phi_1$ is 2-homogeneous. Therefore it induces the same potential function $\Phi$ under Assumption 1 and Assumption 8: $\frac{1}{n^2}\sum_{k \in [n]} \phi_1(n(\theta_k - \hat{\theta}_k)) = \sum_{k \in [n]} \phi_1(\theta_k - \hat{\theta}_k)$. This explains why mirror descent with the scaled potential induced by $\phi_1$ is also in the kernel regime.

**Effect of $\ell_p + \ell_2$ potential**   We examine how scaled potentials of the form $\phi(x) = |x|^p + 0.1x^2$ affect the trained networks. In the left panel of Figure 6, we compare the solution to the variational problem (9) with a constant $p_{\mathcal{B}}$ and $\phi(x) = |x|^p + 0.1x^2$ for different $p$ values, versus B-spline interpolations of the data with varying degrees. Interestingly, as $p$ approaches 1, the variational solution behaves more like linear interpolation. This trend gets clearer in the right panel, which shows the 2D PCA representation of these interpolation functions. Here we directly consider the variational solution for two reasons: (1) these solutions well approximate absolute value networks trained by

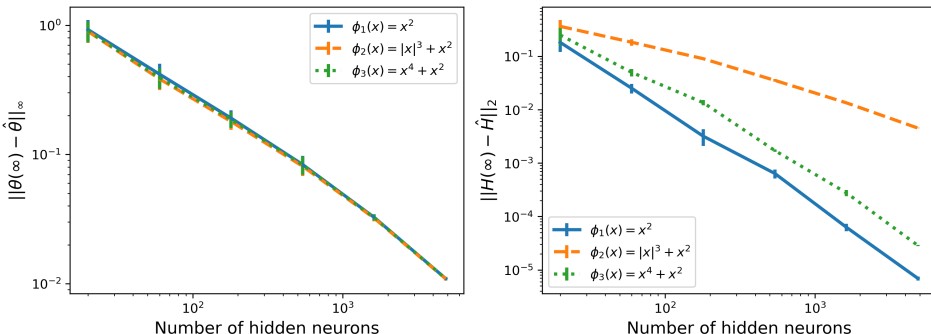

Figure 4: Left: $\ell_\infty$ norm of the difference between the final parameter and initial parameter of ReLU networks trained by mirror descent with different unscaled potentials: $\phi_1 = x^2$, $\phi_2 = |x|^3 + x^2$, and $\phi_3 = x^4 + x^2$, plotted against the number of hidden units. Right: spectral norm (2-norm) of the difference between the final kernel matrix and the initial kernel matrix from the same training instances as the left panel, plotted against the number of hidden units.

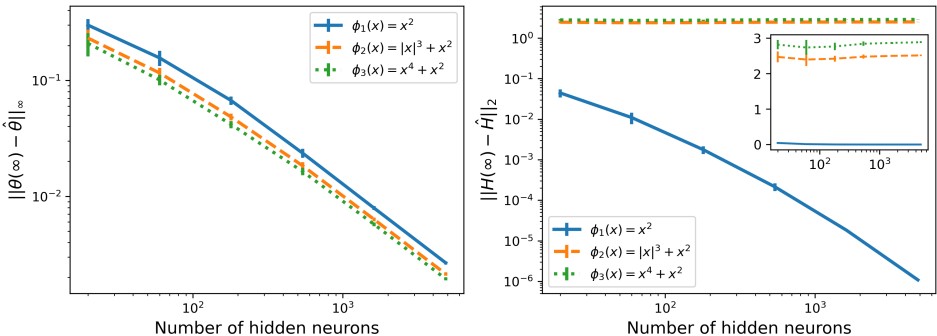

Figure 5: Left panel: $\ell_\infty$ norm of the difference between the final parameter and initial parameter of absolute value networks trained by mirror descent with different scaled potentials: $\phi_1 = x^2$, $\phi_2 = |x|^3 + x^2$, and $\phi_3 = x^4 + x^2$, plotted against the number of hidden units. Right panel: spectral norm (2-norm) of the difference between the final kernel matrix and initial kernel matrix from the same training instances as the left panel, plotted against the number of hidden units. The inset shows the same data with the y-axis in linear scale instead of log-scale.

mirror descent with scaled potentials, as shown in Theorem 9, and (2) we observe that when $p$ takes a large value, the mirror descent converges in a very slow rate and solving the variational problem is more efficient. Also note, potentials $\phi(x) = |x|^p + x^2$ with $p < 2$ are not covered by Assumption 8, as $\phi$ does not have a second derivative at $x = 0$ and hence the trajectory (4) can be ill-defined.

**Effect of hypentropy potentials**  We examine the effect of hypentropy potentials $\rho_\beta(x) = x \operatorname{arcsinh}(x/\beta) - \sqrt{x^2 + \beta^2}$ (Ghai et al., 2020) on the implicit bias of mirror descent training. The motivation behind these potentials is that, as $\beta \to 0^+$ their second derivatives, $\rho_\beta''(x) = 1/\sqrt{x^2 + \beta^2}$, converges to the second derivative of the negative entropy. Moreover, the Bregman divergence induced by the hypentropy function interpolates the squared Euclidean distance and the relative entropy or Kullback–Leibler (KL) divergence, as $\beta$ varies from infinity to zero (Ghai et al., 2020). In agreement with this property, in the left panel of Figure 7, we observe that when $\beta$ takes large value, the trained networks behaves like the variational solution with $\phi(x) = x^2$, which is closely related to training under Euclidean distance. Interestingly, as $\beta$ tends to zero, the trained networks behaves more like the variational solution with $\phi(x) = |x|^3$. This pattern is also shown in the right panel, where we plot the 2D PCA representations for these interpolation functions.

**Trained network vs. width**  Figure 8 compares ReLU networks of different width, trained by mirror descent with different unscaled potentials, against the solution to the variational problem (7).

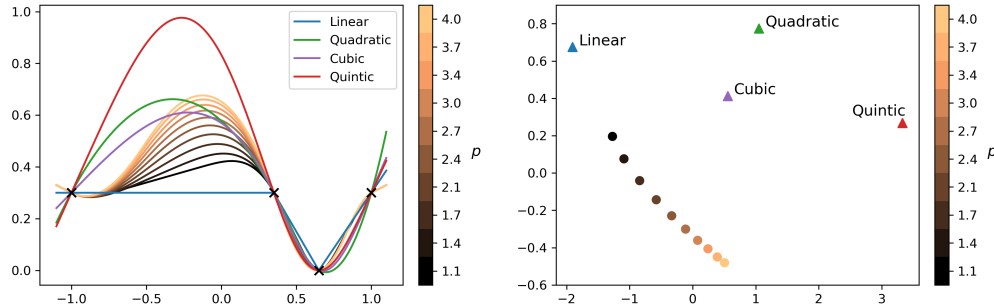

Figure 6: Left: comparison of multiple interpolating functions on a common data set (black crossing points): B-spline interpolations with degree $1$ (blue), $2$ (green), $3$ (purple), and $5$ (red); solutions to the variational problem with a constant $p_{\mathcal{B}}$ and $\phi(x) = |x|^p + 0.1x^2$, where $p$ ranges from $1.1$ to $4.0$ with the corresponding color indicated in the colorbar. Right: 2D PCA representation of the interpolation functions in the left panel.

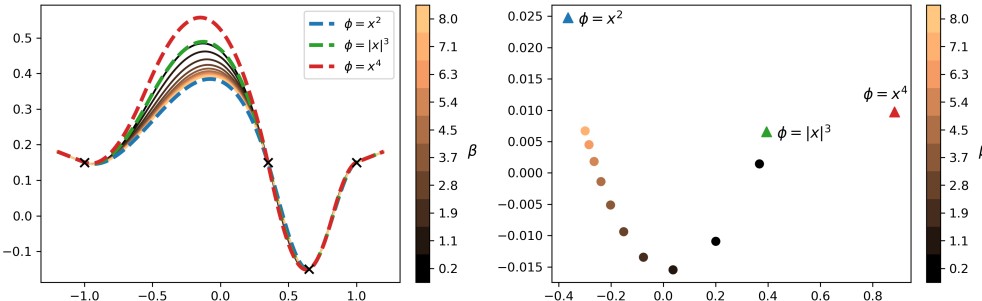

Figure 7: Left: comparison of multiple interpolating functions on a common data set (black crossing points): solutions to the variational problem with a constant $p_{\mathcal{B}}$ and $\phi(x) = |x|^p$ for $p = 2, 3, 4$, and wide absolute value networks with uniformly initialized input biases, unit-norm initialized input weights and zero-initialized output weights and biases, trained by mirror flow with scaled hypentropy potentials $\phi_\beta$ where $\beta$ ranges from $0.2$ to $8.0$ with the corresponding color indicated in the colorbar. Right: 2D PCA representation of the interpolation functions in the left panel.

In agreement with Theorem 2 and Figure 1 (middle panel), the network function converges to the solution to the variational problem as the width increases. A further experiment is conducted for absolute value networks and scaled mirror descent, with results shown in Figure 9. In agreement with Theorem 9, we observe that the network function trained with different scaled potentials converges to the solution to the corresponding variational problem as the width increases.

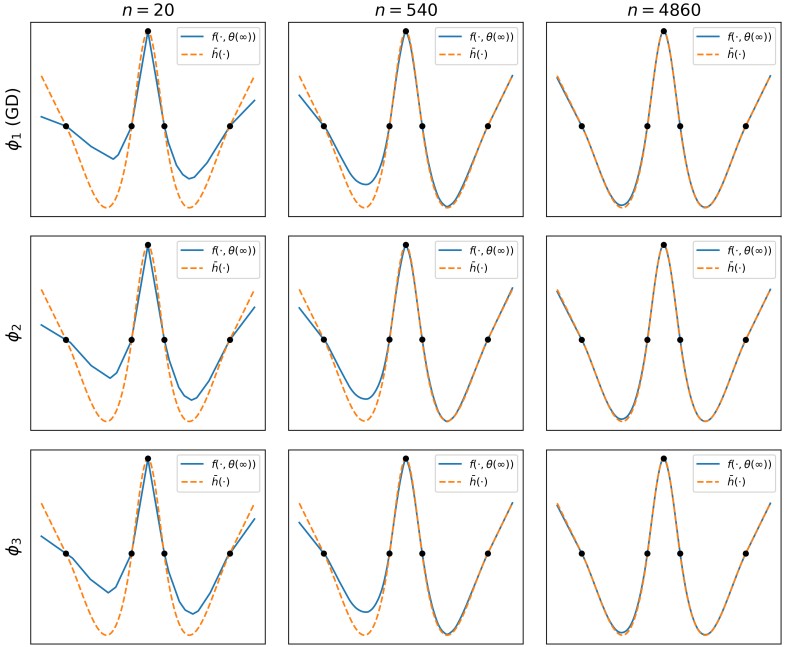

Figure 8: Comparison of ReLU networks of different width, trained by mirror descent with unscaled potentials (solid blue), against the solution to the variational problem (dashed orange). Networks of widths $n = 30, 540, 4860$ (columns) are trained using three different unscaled potentials: $\phi_1 = x^2$, $\phi_2 = |x|^3 + x^2$, and $\phi_3 = x^4 + x^2$ (rows). All networks have uniformly initialized input biases, unit-norm initialized input weights and zero-initialized output weights and biases.

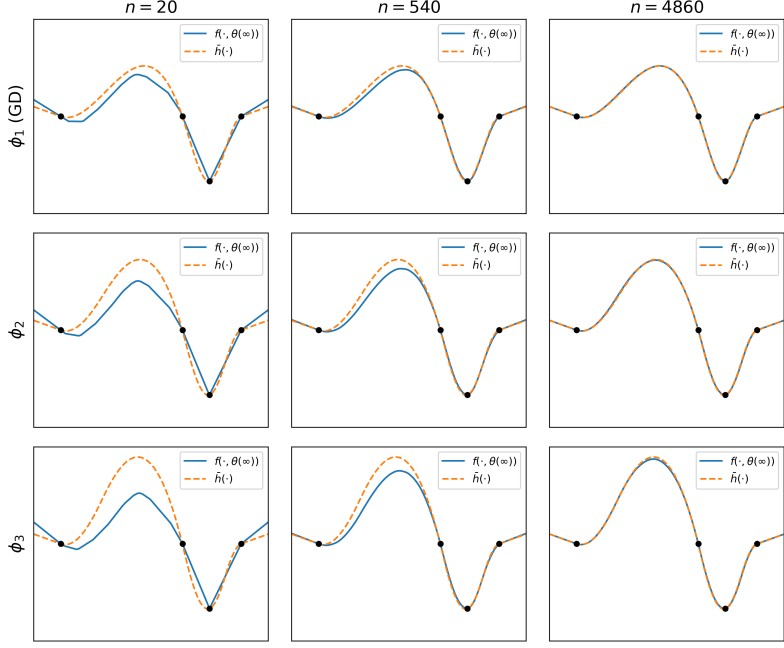

Figure 9: Comparison of absolute value networks of different width, trained by mirror descent with scaled potentials (solid blue), against solutions to the variational problem (dashed orange). Networks of widths $n = 30, 540, 4860$ (columns) are trained using three different unscaled potentials: $\phi_1 = x^2$, $\phi_2 = |x|^3 + x^2$, and $\phi_3 = x^4 + x^2$ (rows). All networks have uniformly initialized input biases, unit-norm initialized input weights and zero-initialized output weights and biases.

