# OpenReview forum: "Implicit Bias of Mirror Flow for Shallow Neural Networks in Univariate Regression"
_ICLR.cc/2025/Conference — ICLR 2025 Spotlight_

### Official Review · Reviewer_8kZ2 · 2024-10-22

**Soundness:** 4
**Presentation:** 4
**Contribution:** 3
**Rating:** 8
**Confidence:** 3

**Summary:**

The paper studies the implicit bias of mirror flow in univariate two-layer networks in the lazy regime. For a broad class of potential functions, they show that the implicit bias at the limit of infinite width is similar to the implicit bias of mirror flow. They also provide a characterization of this implicit bias in function space which generalized prior results. Moreover, they introduce scaled potential functions, where training is in the lazy regime but not in the kernel regime, and characterize its implicit bias.

**Strengths:**

The implicit bias of mirror descent/flow has been studied mostly for linear models. Here, the authors extend our understanding of mirror flow to the case of univariate two-layer networks in the lazy regime, and also give a more precise characterization of the implicit bias in function space for gradient flow in this setting. I think that the contribution is of interest to the implicit bias community, and continues two lines of work: the works on the implicit bias of mirror descent, and the works on the implicit bias of univariate shallow networks.

I believe that the contribution is significant and hence recommend acceptance.

**Weaknesses:**

It would be nice to extend Theorem 2 beyond Assumption 1, which restricts the class of potential functions covered here, and also to relax the many assumptions in Theorem 8. But I understand that the authors needed these restrictions for technical reasons.

**Questions:**

In both Theorem 2 and Theorem 8:
“with high probability over the random parameter initialization,
there exist constants C_1, C_2 such that …”. It sounds like the constants C_1,C_2 may depend on the initialization (which also depends on n). Did you mean to write “there exist constants C_1, C_2 such that w.h.p. over the initialization …”?

---

> ### Author Response · Authors · 2024-11-17
> **Initial response to Reviewer 8kZ2**
>
> Thank you for taking your time to review our manuscript and for your valuable feedback.
>
> > It would be nice to extend Theorem 2 beyond Assumption 1, which restricts the class of potential functions covered here, and also to relax the many assumptions in Theorem 8. But I understand that the authors needed these restrictions for technical reasons.
>
> Thanks for the positive feedback. In the updated manuscript, the theorem assumptions have been relaxed as detailed below:
> * In both Theorem 2 and Theorem 9 (previously Theorem 8), the lazy training and convergence results now hold under general input dimension $d\geq 1$. Specifically, for multivariate input data we have that (i) mirror flow with unscaled potentials induces the same implicit bias as gradient flow; and (ii) mirror flow with scaled potentials induces a Bregman-divergence-minimization bias in the parameter space.
>
> * In Assumption 1, we required the potential to be located at the initial parameter, i.e., $\Phi(\theta)=\sum_k\phi(\theta_k-\theta_k(0))$. Now we showed that, with bounded parameter initialization, Theorem 2 also holds with potentials of the form $\Phi(\theta)=\sum_k\phi(\theta_k)$ (Theorem 26 in Appendix A.4).
>
> &nbsp;
>
> > In both Theorem 2 and Theorem 8: "with high probability over the random parameter initialization, there exist constants $C_1$, $C_2$ such that …". It sounds like the constants $C_1,C_2$ may depend on the initialization (which also depends on n). Did you mean to write "there exist constants $C_1, C_2$ such that w.h.p. over the initialization …"?
>
> Thank you for pointing this out! We did mean that the constants $C_1$ and $C_2$ are independent of the sampling and the width. We have rephrased our results as suggested.

---

> > ### Comment · Reviewer_8kZ2 · 2024-11-22
> >
> > Thanks for the response.

---

### Official Review · Reviewer_QTJu · 2024-11-04

**Soundness:** 3
**Presentation:** 3
**Contribution:** 2
**Rating:** 8
**Confidence:** 4

**Summary:**

The paper establishes various implicit biases for wide shallow networks trained via mirror flow in the context of univariate regression which roughly fall into two categories: (i) parameters do not move far from initialization while asymptotically minimizing risk (ii) the asymptotic solution (asymptotic with respect to time) can be characterized via a variational problem.

**Strengths:**

Characterizing the asymptotic solution of mirror flow via a variational problem is interesting.

**Weaknesses:**

The work only considers univariate models.


minor comment:
Considering the work only deals with mirror flow, I believe the title should be remedied to "Implicit bias of Mirror Flow ...".

**Questions:**

1) All the convex potentials bake in the initial parameter $\hat \theta := \theta_0$. I wonder how much of the implicit bias results are affected by the choice of $\hat \theta$. If $\hat \theta$ was arbitrary (or any point other than $\theta_0$), how do the lazy training results change (i.e. is it still true that the parameters do not move far from initialization)? Of course when the coordinate function $\phi(x) = \frac 1 2 x^2$ (i.e. GF), the choice of $\hat \theta$ does not matter but for $\phi(x) = x^2 + x^4$, the mirror flow update explicitly penalizes movement away from $\hat \theta$ and hence it less clear whether lazy training should occur.

2) Why is the mirror flow update scaled by $\eta$? Do similar results hold if $\eta$ is dropped?

3) Is the restriction to the univariate case only needed for the variational characterization of the asymptotic solution?
(i.e. can the univariate requirement be dropped for establishing lazy training results and risk minimization?)

---

> ### Author Response · Authors · 2024-11-17
> **Initial response to Reviewer QTJu (part 1)**
>
> Thank you for taking your time to review our manuscript and for your valuable feedback. We address your comments and questions as detailed below.
>
> > Paper considers only univariate setting.
>
> > Is the restriction to the univariate case only needed for the variational characterization of the asymptotic solution? (i.e. can the univariate requirement be dropped for establishing lazy training results and risk minimization?)
>
> Thank you for pointing this out! Yes, the restriction to univariate input is only needed for the variational characterization. We have updated Theorem 2 and Theorem 9 (previously Theorem 8) and now the lazy training and risk minimization results are presented for general input dimension. **Notably, for multivariate input data we still have that (i) mirror flow with unscaled potentials induces the same implicit bias as gradient flow; and (ii) mirror flow with scaled potentials induces a Bregman-divergence-minimization bias in the parameter space.**
>
> &nbsp;
>
> > All the convex potentials bake in the initial parameter $\hat{\theta} = \theta(0)$. I wonder how much of the implicit bias results are affected by the choice of $\hat{\theta}$. If $\hat{\theta}$ was arbitrary (or any point other than $\theta(0)$), how do the lazy training results change (i.e. is it still true that the parameters do not move far from initialization)? Of course when the coordinate function $\phi(x)=\frac{1}{2}x^2$ (i.e. GF), the choice of $\hat{\theta}$ does not matter but for $\phi(x)= x^2 + x^4$, the mirror flow update explicitly penalizes movement away from $\hat{\theta}$ and hence it is less clear whether lazy training should occur.
>
> This is a great question! For unscaled potentials, whether the potential is centered at the initial parameter does not fundamentally affect the lazy training and the implicit bias results. In fact, **the entire Theorem 2 also holds for potentials of the form $\Phi(\theta)=\sum_k \phi(\theta_k)$,  with a restriction to bounded random parameter initialization**, instead of sub-Gaussian initialization. We formally presented the results in Theorem 26 in Appendix A.4, and added a remark following Theorem 2 (Remark 3).
> For "uncentered" and scaled potentials $\Phi(\theta)=\sum_k \phi(n\theta_k)$, the analysis becomes tricky,
> as $\Phi$ has an intractable behavior at initialization $\theta=\theta(0)$ as $n\rightarrow \infty$. Below, we provide more details on the unscaled case:
>
> * Lazy training and convergence: recall the training dynamics of the function output: $\frac{d}{dt}f(t)=-H(t)  (f(t)-y)$ where $H(t)=\eta J_\theta f(t)(\nabla^2\Phi(\theta(t)))^{-1} J_\theta f(t)^T$. The main ingredients for lazy training to occur are:
> $H(0)$ is well conditioned, $H(\cdot)$ has certain smoothness properties, and the parameter's updates are slow (good conditioning ensures the loss decreases rapidly at initialization and, due to the smoothness and the slow parameter update, the loss decreases to zero before the parameters move significantly). In our approach, "centering" the potential is only used to control the eigenvalues of the Hessian $\nabla^2 \Phi(\theta(0))$.Specifically, under sub-Gaussian initialization the input biases exhibit the behavior $\max_{k\in[n]}|b_k(0)| = O_p(\sqrt{\log n})$, making it hard to control $\phi^{\prime\prime}(b_k(0))$. This can be addressed by either considering bounded initialization where $\max_{k\in[n]}|b_k(0)| = O(1)$, or centering the potential at the initial parameter. You are right that certain potentials like $\phi(\theta_k)=(\theta_k-\hat{\theta}_k)^2+(\theta_k-\hat{\theta}_k)^4$ penalize movement away from $\hat{\theta}$ in terms of $1/\phi^{\prime\prime}=1/(2+12(\theta-\hat{\theta})^2)$. Though this penalization becomes effective only when $\theta_k-\hat{\theta}_k$ reaches a moderate scale.  In our case, the loss decreases to zero before such scale is reached, and hence such effect is not the driving force of lazy training.
>
> * Implicit bias: recall the implicit bias is described by minimizing the Bregman divergence on the output weights $D_\Phi(\boldsymbol{a}, \boldsymbol{a}(0))$, or its Taylor approximation $I(\boldsymbol{a})=\nabla^2\Phi(\boldsymbol{a}(0))(\boldsymbol{a}-\boldsymbol{a}(0), \boldsymbol{a}-\boldsymbol{a}(0))$. For "uncentered'' potentials $\Phi(\theta)=\sum_k \phi(\theta_k)$, $I(\boldsymbol{a})$ becomes $\sum_k \phi^{\prime\prime}(a_k(0))(a_k - a_k(0))^2$. Since $a_k(0)$'s are sampled from a sub-Gaussian distribution scaled by $1/\sqrt{n}$, we have $\max_k|a_k(0)|=O_p(\sqrt{(\log n)/n})=o_p(1)$. Consequently, $I(\boldsymbol{a})$ is approximately $\phi^{\prime\prime}(0)\sum_{k}(a_k-a_k(0))^2$. Therefore, the implicit bias is also independent of the center or location of the potential.

---

> ### Author Response · Authors · 2024-11-17
> **Initial response to Reviewer QTJu (part 2)**
>
> > Why is the mirror flow update scaled by $\eta$? Do similar results hold if $\eta$ is dropped?
>
> The scale $\eta$ ensures the training dynamics of the network's output, $\frac{d}{dt}f(t)=-H(t)  (f(t)-y)$, where $H(t)=\eta J_\theta f(t)(\nabla^2\Phi(\theta(t)))^{-1} J_\theta f(t)^T$, is well-defined as $n\rightarrow \infty$. More specifically, under standard parametrization, $||J_\theta f(0)||=O(\sqrt{n})$. Then a necessary condition for $||H(0)||=O(1)$ is $\eta=\eta_0/n$ for some constant $\eta_0>0$ (this constant only affects the convergence time). If $\eta$ is dropped, the dynamics might explode due to divergent eigenvalues of $H(0)$. Please note that such scale is also considered in the literature on gradient flow for networks with standard parametrization, e.g., [1][2].
>
> [1] Lee, Jaehoon, et al. "Wide neural networks of any depth evolve as linear models under gradient descent." Advances in neural information processing systems 32 (2019).
>
> [2] Jin, Hui, and Guido Montúfar. "Implicit bias of gradient descent for mean squared error regression with two-layer wide neural networks." Journal of Machine Learning Research 24.137 (2023): 1-97.
>
> &nbsp;
>
> > minor comment: Considering the work only deals with mirror flow, I believe the title should be remedied to ``Implicit bias of Mirror Flow ...''.
>
> Thanks for the suggestion! We agree and have revised the title to "Implicit Bias of Mirror Flow for Shallow Neural Networks in Univariate Regression".

---

> > ### Comment · Reviewer_QTJu · 2024-11-22
> >
> > Thank you for the detailed responses and I have raised the score accordingly.

---

### Official Review · Reviewer_eX1T · 2024-11-07

**Soundness:** 4
**Presentation:** 3
**Contribution:** 2
**Rating:** 6
**Confidence:** 3

**Summary:**

The paper studies the implicit bias of mirror flow on shallow univariate neural networks in the standard parameterization. The authors show that the networks are in the lazy regime (only the second layer weights move non-negligibly), and characterize the representation norm in two different regimes. In the "unscaled" regime, the mirror map plays a minor role, only through its Taylor expansion around 0, leading to an NTK-like implicit bias similar to GD. In the "scaled" regime, where the potentials are rescaled by width, a richer implicit bias arises involving the corresponding Bregman divergence, which is no longer a RKHS norm.

**Strengths:**

The paper is well written, and the obtained implicit bias results for scaled potential is quite interesting as it involves new norms that have not been used before in the context of neural networks, to my knowledge.

**Weaknesses:**

The significance is unclear: while the obtained norms are interesting, it is unclear if mirror flow is a relevant method in the study of neural networks. The lazy training regime suggests that it may be good to reframe the paper more generally outside of neural networks (which in practice are often in non-lazy regimes), for instance mirror descent on random features models, which could be of interest to the statistical learning and kernels community more broadly. The analysis is also somewhat incremental over existing implicit bias results.

**Questions:**

* Given that the networks are in lazy regimes and similar to random feature models, is the setting related to this paper on random feature models with Lp penalties on the weights? https://arxiv.org/abs/2103.15996

* Do you have a sense of whether such mirror flow analyses extend to feature learning regimes, e.g. by switching to a mean-field parameterization? What would be the resulting implicit bias?

---

> ### Author Response · Authors · 2024-11-17
> **Initial response to Reviewer eX1T (part 1)**
>
> Thank you for taking your time to review our manuscript and for your valuable feedback. We address your comments and questions as detailed below.
> > The significance is unclear: while the obtained norms are interesting, it is unclear if mirror flow is a relevant method in the study of neural networks. The lazy training regime suggests that it may be good to reframe the paper more generally outside of neural networks (which in practice are often in non-lazy regimes), for instance ... The analysis is also somewhat incremental over existing implicit bias results.
>
> We agree that studying mirror flow for random feature model (RFM) could be an interesting topic and we believe our analysis can be extended to RFMs with appropriate adaptions. Though in this work, we consider a shallow neural network (NN) and we highlight the significance of our study below:
>
> * Notice that a NN is not a RFM and is non-linear in its parameters. Due to this non-linearity, the loss landscape of a NN is non-convex and the training dynamics of NN is potentially more complicated than those of RFM. One main contribution of our work is that we provided a close examination of the mirror flow dynamics for NN. For unscaled potentials, we proved that lazy training occurs and NN asymptotically behaves like RFM. We point out this asymptotic equivalence between NN and RFM itself is not trivial, and has not been identified by existing literature beyond gradient flow. For scaled potentials, the dynamics can be even more complex as the Hessian matrix $\nabla^2 \Phi$, which governs the dynamics, can change significantly during training. We derived sufficient conditions for the potentials so that lazy training would occur (Proposition 30 and 31), with a careful examination of how the Hessian's properties affect the dynamics. For instance, we showed that the decay rate of the smallest eigenvalue of $\nabla^2 \Phi(\theta)$ as $||\theta||_2$ increases, determines how far the parameter moves away from initialization (Proposition 27).
>
> * In addition, to analyzing the mirror flow dynamics, another important contribution of our work is that we characterized the implicit bias in the function space for NN. That is, we not only showed that mirror flow implicitly minimizes a function on the parameter, but what this means in terms of the learned function. The topic of implicit biases has been intensively studied in recent years for neural networks. Particularly, there are many works studying how different optimization choices, that generalize gradient flow, affect the implicit bias, such as large step-size [1], stochasticity [2], etc. Mirror flow is an important generalization of gradient flow that has been studied in machine learning tasks including linear regression and matrix sensing. Our work presented the first systematic investigation of the implicit bias of mirror flow on the training of NN.
>
> * Previous works on the implicit bias of gradient flow for shallow ReLU networks relied (implicitly or explicitly) on a rather restrictive assumption: data adjustment or skip connection. Specifically, previous characterizations of the implicit bias are precise only when the original data is adjusted by an unknown linear transformation, or when skip connections are added to the network architecture. We lifted this limitation and our implicit bias result applies to general data and to networks without skip connections.
> In particular, we revealed that the position of the data also has an effect on the learned network (Theorem 2 and Figure 1). This important aspect has not been captured by previous works.
>
> Beyond the aspects we mentioned above, mirror flow also has the following connections to neural network optimization:
> (i) it has been used to characterize the implicit bias of gradient flow for neural networks, as gradient flow can be expressed as a time-warped mirror flow with specific potentials [3]; (ii) mirror flow is equivalent to gradient flow under a broad class of reparametrizations [4]. Under this viewpoint, our analysis also revealed the effects of reparametrization on the implicit bias of gradient flow, which we detailed in Appendix D.
>
>
>
>
> [1] Qiao, Dan, et al. "Stable Minima Cannot Overfit in Univariate ReLU Networks: Generalization by Large Step Sizes." arXiv preprint arXiv:2406.06838 (2024).
>
> [2] Ma, Chao, and Lexing Ying. "On linear stability of sgd and input-smoothness of neural networks." Advances in Neural Information Processing Systems 34 (2021): 16805-16817.
>
> [3] Azulay, Shahar, et al. "On the implicit bias of initialization shape: Beyond infinitesimal mirror descent." International Conference on Machine Learning. PMLR, 2021.
>
> [4] Li, Zhiyuan, et al. "Implicit bias of gradient descent on reparametrized models: On equivalence to mirror descent." Advances in Neural Information Processing Systems 35 (2022): 34626-34640.

---

> ### Author Response · Authors · 2024-11-17
> **Initial response to Reviewer eX1T (part 2)**
>
> > Given that the networks are in lazy regimes and similar to random feature models, is the setting related to this paper on random feature models with Lp penalties on the weights? https://arxiv.org/abs/2103.15996
>
> Thank you for pointing out this relevant work! Please allow us to summarize their work: they considered minimal Lp-norm interpolation problem for RFM and studied when the finite-width solution approximates the infinite-width solution. Below we discuss the connection and difference between their work and ours.
>
> * While they directly studied the minimizer of the min-norm interpolation problem in a static and training-free manner (this problem is strictly convex for RFM, so the training dynamics may not be as interesting), we considered mirror flow training for NN under squared loss, and showed that, even when there is no explicit regularization in the training objective, mirror flow is implicitly biased to an interpolation solution that minimizes a functional.
>
> * Our work provided an alternative answer to the key question addressed in the referenced paper. As the tractability of solving the infinite-width min-norm interpolation problem is unclear beyond L2 norm, the authors showed that, one can solve the finite-width problem with sufficiently large width to approximate the infinite-width solution. According to our results, this is can be also achieved by training a shallow NN with sufficiently large width using mirror flow with scaled potential (Proposition 32). However, our results are asymptotic, whereas they offered non-asymptotic bounds on the distance between finite-width and infinite-width solution.
>
> * Our work and the referenced paper considered different "norm". They considered Lp norm $\phi=|x|^p$, which falls outside our assumption: when $p<2$, $\phi$ lacks second-order differentiability at the origin; when $p>2$, $\phi''(0)=0$. In both cases, the inverse Hessian of the potential is ill-defined and thus not covered by us. We point out one can modify Lp norm for $p>2$ by adding an $\epsilon x^2$ term to include it in our assumption. On the other hand, our results apply to a broad class of Bregman divergence-induced "norms", which is not included by them.
>
> * While the min-norm interpolation problem is defined on the parameter space, we translate the problem to the function space as a variational problem. This aspect is not considered in their work and we believe our results provide insights into how different choices of norms affect the solution function and offer principled guidance for norm selection.
>
>
> &nbsp;
>
> > Do you have a sense of whether such mirror flow analyses extend to feature learning regimes, e.g. by switching to a mean-field parameterization? What would be the resulting implicit bias?
>
> That is great question! In general, we think it could be quite challenging to analyze mirror flow in feature learning regime, as one needs to simultaneously consider the highly non-convex loss landscape and the effect of the mirror map. A possible first step could be considering simpler models that allow feature learning, such as deep linear networks, or imposing strong data assumptions. For well-separated data, a max-$\ell_2$-margin bias is observed in the case of gradient flow [1]. Then one might expect that mirror flow induces a max-Bregman-divergence-margin bias (such observation has been made in linear classification [2]). Regarding the mean field regime, in fact we are currently looking into this direction. Under gradient flow, the parameter evolution is described by a gradient flow in the Wasserstein space endowed with the 2-Wasserstein distance. A natural guess is that mirror flow replaces the 2-Wasserstein distance with a Bregman divergence related "distance", which relates to a mirror flow in the Wasserstein space [3]. Future works then involve showing the finite-width dynamic converges to the infinite-dimensional mirror flow and analyzing the training dynamics.
>
>
>
> [1] Lyu, Kaifeng, and Jian Li. "Gradient descent maximizes the margin of homogeneous neural networks." arXiv preprint arXiv:1906.05890 (2019).
>
> [2] Pesme, Scott, Radu-Alexandru Dragomir, and Nicolas Flammarion. "Implicit Bias of Mirror Flow on Separable Data." arXiv preprint arXiv:2406.12763 (2024).
>
> [3] Bonet, C., Uscidda, T., David, A., Aubin-Frankowski, P.-C., and Korba, A. (2024). Mirror and preconditioned gradient descent in wasserstein space. In The Thirty-eighth Annual Conference on Neural Information Processing Systems.

---

> ### Author Response · Authors · 2024-11-25
>
> Dear reviewer,
>
> As the discussion period is ending soon, we kindly ask if you could review our responses to your review. If you have further questions or comments, we will do our best to address them before the discussion period ends. If our responses have resolved your concerns, we would greatly appreciate if you could indicate this accordingly.

---

### Public Comment · ~Shuang_Liang6 · 2025-02-08
**Camera-Ready Revisions**

We sincerely thank the reviewers and the Area Chair for their valuable suggestions and feedback. Following the acceptance of the paper, we have made the following revisions in the camera-ready version addressing their final comments:

* We rephrased the interpretation of the experiments on scaled potentials (Figure 2), to better illustrate that “the parameter initialization distribution determines the strength of the curvature penalty over different locations of the input space and the potential determines the strength of the penalty over different magnitudes of the curvature”.

* We revised the discussion on representation cost in the related work section to improve accuracy and completeness.

---

### Meta-Review · Area_Chair_Q61m · 2024-12-20

**Metareview:**

This paper studies the implicit bias of mirror flow in shallow (one-hidden-layer) and wide neural networks, focusing on ReLU and absolute value activations. For a class of “unscaled” potentials, the authors show that the ReLU network under mirror flow stays in the lazy and kernel regime and (perhaps surprisingly) has an implicit bias identical to gradient flow in the infinite width limit. The paper further provides a variational characterization of the trained function in the univariate input case. The authors also consider “scaled” potentials, and show that training an absolute-value-activated NN under mirror flow now stays in the lazy regime but not necessarily in the kernel regime. For the univariate input case, the paper also offers a variational characterization of the implicit bias.

The reviewers found the results novel and interesting. The paper offers a concrete characterization on the implicit bias of mirror flow on shallow and wide networks. To me, it interesting to know that for standard unscaled potential functions the implicit bias of mirror descent ends up being the same as gradient flow in the infinite width limit, which is in contrast to other results on finite number of parameters. Also, for scaled potentials, what I find the most interesting is that the model is in the lazy regime BUT can be in a non-kernel regime—personally, this is something that I haven’t seen in the literature. The variational characterization is nice, and the experiments look very much aligned with theoretical analysis. Overall, it is deemed that the paper provides strong contributions to the implicit bias community.

Of course, there are some limitations in the analysis, as acknowledged by the authors in the conclusion section: results holding for 1) shallow and (infinitely) wide networks, 2) univariate inputs, 3) specific first layer weight initialization, and 4) under some restrictions on the potential function.

However, my overall evaluation is that the merits of this paper significantly outweigh the shortcomings, and I recommend acceptance of this paper.

**Additional Comments On Reviewer Discussion:**

- One noteworthy point on the rebuttal revision: Initially all results were stated for univariate inputs, but the authors revised the paper to include extension of some of the analyses to multivariate inputs (perhaps thanks to the question from Reviewer QTJu).
- Suggestion from AC: The authors stress their findings: “A takeaway message is that the parameter initialization distribution determines the strength of the curvature penalty over different locations of the input space and the potential determines the strength of the penalty over different magnitudes of the curvature,” but it seems like the experiments in the main text do not highlight this finding. I recommend the authors to consider including experiments highlighting this message in the main text, illustrating how different choices of initialization distribution and potential can lead to different solutions.

---

### Decision · Program_Chairs · 2025-01-22

Accept (Spotlight)